# Towards Theoretically Understanding Why SGD Generalizes Better Than ADAM in Deep Learning

**Pan Zhou**\*, **Jiashi Feng**†, **Chao Ma**‡, **Caiming Xiong**\*, **Steven HOI**\*, **Weinan E**‡

\*Salesforce Research, † National University of Singapore, ‡ Princeton University

{pzhou,shoi,cxiong}@salesforce.com  elefjia@nus.edu.sg  {chaom@, weinan@math.}princeton.edu

## Abstract

It is not clear yet why ADAM-alike adaptive gradient algorithms suffer from worse generalization performance than SGD despite their faster training speed. This work aims to provide understandings on this generalization gap by analyzing their local convergence behaviors. Specifically, we observe the heavy tails of gradient noise in these algorithms. This motivates us to analyze these algorithms through their Lévy-driven stochastic differential equations (SDEs) because of the similar convergence behaviors of an algorithm and its SDE. Then we establish the escaping time of these SDEs from a local basin. The result shows that (1) the escaping time of both SGD and ADAM depends on the Radon measure of the basin positively and the heaviness of gradient noise negatively; (2) for the same basin, SGD enjoys smaller escaping time than ADAM, mainly because (a) the geometry adaptation in ADAM via adaptively scaling each gradient coordinate well diminishes the anisotropic structure in gradient noise and results in larger Radon measure of a basin; (b) the exponential gradient average in ADAM smooths its gradient and leads to lighter gradient noise tails than SGD. So SGD is more locally unstable than ADAM at sharp minima defined as the minima whose local basins have small Radon measure, and can better escape from them to flatter ones with larger Radon measure. As flat minima here which often refer to the minima at flat or asymmetric basins/valleys often generalize better than sharp ones [1, 2], our result explains the better generalization performance of SGD over ADAM. Finally, experimental results confirm our heavy-tailed gradient noise assumption and theoretical affirmation.

## 1 Introduction

Stochastic gradient descent (SGD) [3, 4] has become one of the most popular algorithms for training deep neural networks [5–11]. In spite of its simplicity and effectiveness, SGD uses one learning rate for all gradient coordinates and could suffer from unsatisfactory convergence performance, especially for ill-conditioned problems [12]. To avoid this issue, a variety of adaptive gradient algorithms have been developed that adjust learning rate for each gradient coordinate according to the current geometry curvature of the objective function [13–16]. These algorithms, especially for ADAM, have achieved much faster convergence speed than vanilla SGD in practice.

Despite their faster convergence behaviors, these adaptive gradient algorithms usually suffer from worse generalization performance than SGD [12, 17, 18]. Specifically, adaptive gradient algorithms often show faster progress in the training phase but their performance quickly reaches a plateaus on test data. Differently, SGD usually improves model performance slowly but could achieve higher test performance. One empirical explanation [1, 19–21] for this generalization gap is that adaptive gradient algorithms tend to converge to sharp minima whose local basin has large curvature and usually generalize poorly, while SGD prefers to find flat minima and thus generalizes better. However, recent evidence [2, 22] shows that (1) for deep neural networks, the minima at the asymmetric

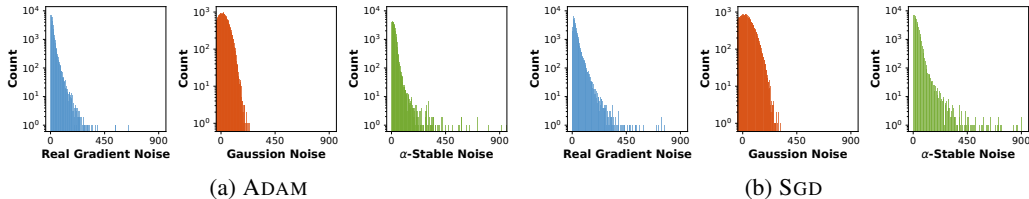

(a) ADAM                                         (b) SGD

Figure 1: Illustration of gradient noise in ADAM and SGD on AlexNet trained with CIFAR10. (b) is produced under the same setting in [23]. By comparison, one can observe (1) $\alpha$-stable noise can better characterize real gradient noise and (2) SGD has heavier gradient noise tails than ADAM.

basins/valleys where both steep and flat directions exist also generalize well though they are sharp in terms of their local curvature, and (2) SGD often converges to these minima. So the argument of the conventional "flat" and "sharp" minima defined on curvature cannot explain these new results. Thus the reason for the generalization gap between adaptive gradient methods and SGD is still unclear.

In this work, we provide a new viewpoint for understanding the generalization performance gap. We first formulate ADAM and SGD as Lévy-driven stochastic differential equations (SDEs), since the SDE of an algorithm shares similar convergence behaviors of the algorithm and can be analyzed more easily than directly analyzing the algorithm. Then we analyze the escaping behaviors of these SDEs at local minima to investigate the generalization gap between ADAM and SGD, as escaping behaviors determine which basin that an algorithm finally converges to and thus affect the generalization performance of the algorithm. By analysis, we find that compared with ADAM, SGD is more locally unstable and is more likely to converge to the minima at the flat or asymmetric basins/valleys which often have better generalization performance over other type minima. So our results can explain the better generalization performance of SGD over ADAM. Our contributions are highlighted below.

Firstly, this work is the first one that adopts Lévy-driven SDE which better characterizes the algorithm gradient noise in practice, to analyze the adaptive gradient algorithms. Specifically, Fig. 1 shows that the gradient noise in ADAM and SGD, i.e. the difference between the full and stochastic gradients, has heavy tails and can be well characterized by symmetric $\alpha$-stable ($\mathcal{S}\alpha\mathcal{S}$) distribution [24]. Based on this observation, we view ADAM and SGD as discretization of the continuous-time processes and formulate the processes as Lévy-driven SDEs to analyze their behaviors. Compared with Gaussian gradient noise assumption in SGD [25–27], $\mathcal{S}\alpha\mathcal{S}$ distribution assumption can characterize the heavy-tailed gradient noise in practice more accurately as shown in Fig. 1, and also better explains the different generalization performance of SGD and ADAM as discussed in Sec. 3. This work extends [23, 28] from SGD on the over-simplified one-dimensional problems to much more complicated adaptive algorithms on high-dimensional problems. It also differs from [29], as [29] considers escaping behaviors of SGD along several fixed directions, while this work analyzes the dynamic underlying structures in gradient noise that plays an important role in the local escaping behaviors of both ADAM and SGD.

Next, we theoretically prove that for the Lévy-driven SDEs of ADAM and SGD, their escaping time $\Gamma$ from a local basin $\boldsymbol{\Omega}$, namely the least time for escaping from the inner of $\boldsymbol{\Omega}$ to its outside, is at the order of $\mathcal{O}(\varepsilon^{-\alpha}/m(\boldsymbol{\mathcal{W}}))$, where the constant $\varepsilon \in (0, 1)$ relies on the learning rate of algorithms and $\alpha$ denotes the tail index of $\mathcal{S}\alpha\mathcal{S}$ distribution. Here $m(\boldsymbol{\mathcal{W}})$ is a non-zero Radon measure on the escaping set $\boldsymbol{\mathcal{W}}$ of ADAM and SGD at the local basin $\boldsymbol{\Omega}$ (see Sec. 4.1), and actually negatively relies on the Radon measure of $\boldsymbol{\Omega}$. So both ADAM and SGD have small escaping time at the "sharp" minima whose corresponding basins $\boldsymbol{\Omega}$ have small Radon measure. It means that ADAM and SGD are actually unstable at "sharp" minima and would escape them to "flatter" ones. Note, the Radon measure of $\boldsymbol{\Omega}$ positively depends on the volume of $\boldsymbol{\Omega}$. So these results also well explain the observations in [1, 2, 20, 21] that the minima of deep networks found by SGD often locate at the flat or asymmetric valleys, as their corresponding basins have large volumes and thus large Radon measure.

Finally, our results can answer why SGD often converges to flatter minima than ADAM in terms of Radon measure, and thus explain the generalization gap between ADAM and SGD. Firstly, our analysis shows that even for the same basin $\boldsymbol{\Omega}$, ADAM often has smaller Radon measure $m(\boldsymbol{\mathcal{W}})$ on the escaping set $\boldsymbol{\mathcal{W}}$ at $\boldsymbol{\Omega}$ than SGD, as the geometry adaptation in ADAM via adaptively scaling each gradient coordinate well diminishes underlying anisotropic structure in gradient noise and leads to smaller $m(\boldsymbol{\mathcal{W}})$. Secondly, the empirical results in Sec. 5 and Fig. 1 show that SGD often has much

smaller tail index $\alpha$ of gradient noise than ADAM for some optimization iterations and thus enjoys smaller factor $\varepsilon^{-\alpha}$. These results together show that SGD is more locally unstable and would like to converge to flatter minima with larger measure $m(\mathcal{W})$ which often refer to the minima at the flat and asymmetric basins/valleys, according with empirical evidences in [12, 17, 30, 31]. Considering the observations in [1, 19–21] that the minima at the flat and asymmetric basins/valleys often generalize better, our results well explain the generalization gap between ADAM and SGD. Besides, our results also show that SGD benefits from its anisotropic gradient noise on its escaping behaviors, while ADAM does not.

## 2  Related Work

Adaptive gradient algorithms have become the default optimization tools in deep learning because of their fast convergence speed. But they often suffer from worse generalization performance than SGD [12, 17, 30, 31]. Subsequently, most works [12, 17, 18, 30, 31] empirically analyze this issue from the argument of flat and sharp minima defined on local curvature in [19] that flat minima often generalize better than sharp ones, as they observed that SGD often converges to flatter minima than adaptive gradient algorithms, *e.g.* ADAM. However, Sagun *et al.* [22] and He *et al.* [2] observed that the minima of modern deep networks at the asymmetric valleys where both steep and flat directions exist also generalize well, and SGD often converges to these minima. So the conventional flat and sharp argument cannot explain these new results. This work theoretically shows that SGD tends to converge to the minima whose local basin has larger Radon measure. It well explains the above new observations, as the minima with larger Radon measure often locate at the flat and asymmetric basins/valleys. Moreover, based on our results, exploring invariant Radon measure to parameter scaling in networks could resolve the issue in [32] that flat minima could become sharp via parameter scaling. See more details in Appendix B. Note, ADAM could achieve better performance than SGD when gradient clipping is required [33], *e.g.* attention models with gradient exploding issue, as adaptation in ADAM provides a clipping effect. This work considers a general non-gradient-exploding setting, as it is more practical across many important tasks, *e.g.* classification.

For theoretical generalization analysis, most works [25–27, 34] only focus on analyzing SGD. They formulated SGD into Brownian motion based SDE via assuming gradient noise to be Gaussian. For instance, Jastrzkebski *et al.* [26] proved that the larger ratio of learning rate to mini-batch size in SGD leads to flatter minima and better generalization. But Simsekli *et al.* [23] empirically found that the gradient noise has heavy tails and can be characterized by $\mathcal{S}\alpha\mathcal{S}$ distribution instead of Gaussian distribution. Chaudhari *et al.* [27] also claimed that the trajectories of SGD in deep networks are not Brownian motion. Then Simsekli *et al.* [23] formulated SGD as a Lévy-driven SDE and adopted the results in [28] to show that SGD tends to converge to flat minima on one dimensional problems. Pavlyukevich *et al.* [29] extended the one-dimensional SDE in [28] and analyzed escaping behaviors of SGD along several fixed directions, differing from this work that analyzes dynamic underlying structures in gradient noise that greatly affect escaping behaviors of both ADAM and SGD.

The literature targeting theoretically understanding the generalization degeneration of adaptive gradient algorithms are limited mainly due to their more complex algorithms. Wilson *et al.* [17] constructed a binary classification problem and showed that ADAGRAD [13] tend to give undue influence to spurious features that have no effect on out-of-sample generalization. Unlike the above theoretical works that focus on analyzing SGD only or special problems, we target at revealing the different convergence behaviors of adaptive gradient algorithms and SGD and also analyzing their different generalization performance, which is of more practical interest especially in deep learning.

## 3  Lévy-driven SDEs of Algorithms in Deep Learning

In this section, we first briefly introduce SGD and ADAM, and formulate them as discretization of stochastic differential equations (SDEs) which is a popular approach to analyze algorithm behaviors. Suppose the objective function of $n$ components in deep learning models is formulated as

$$\min_{\boldsymbol{\theta} \in \mathbb{R}^d} \ \boldsymbol{F}(\boldsymbol{\theta}) := \frac{1}{n} \sum_{i=1}^{n} f_i(\boldsymbol{\theta}), \tag{1}$$

where $f_i(\boldsymbol{\theta})$ is the loss of the $i$-th sample. Subsequently, we focus on analyzing SGD and ADAM. Note our analysis technique is applicable to other adaptive algorithms with similar results as ADAM.

### 3.1  SGD and ADAM

As one of the most effective algorithms, SGD [3] solves problem (1) by sampling a data mini-batch $\mathcal{S}_t$ of size $S$ and then running one gradient descent step:

$$\boldsymbol{\theta}_{t+1} = \boldsymbol{\theta}_t - \eta \nabla f_{\mathcal{S}_t}(\boldsymbol{\theta}_t), \tag{2}$$

where $\nabla f_{\mathcal{S}_t}(\boldsymbol{\theta}_t) = \frac{1}{S}\sum_{i\in\mathcal{S}_t}\nabla f_i(\boldsymbol{\theta}_t)$ denotes the gradient on mini-batch $\mathcal{S}_t$, and $\eta$ is the learning rate. Recently, to improve the efficiency of SGD, adaptive gradient algorithms, such as ADAGRAD [13], RMSPROP [14] and ADAM [15], are developed which adjust the learning rate of each gradient coordinate according to the current geometric curvature. Among them, ADAM has become the default training algorithm in deep learning. Specifically, ADAM estimates the current gradient $\nabla \boldsymbol{F}(\boldsymbol{\theta}_t)$ as

$$\boldsymbol{m}_t = \beta_1 \boldsymbol{m}_{t-1} + (1-\beta_1)\nabla f_{\mathcal{S}_t}(\boldsymbol{\theta}_t) \quad \text{with } \boldsymbol{m}_0 = \boldsymbol{0} \text{ and } \beta_1 \in (0,1).$$

Then like natural gradient descent [35], ADAM adapts itself to the function geometry via a diagonal Fisher matrix approximation $\mathrm{diag}\,(\boldsymbol{v}_t)$ which serves as a preconditioner and is defined as

$$\boldsymbol{v}_t = \beta_2 \boldsymbol{v}_{t-1} + (1-\beta_2)[\nabla f_{\mathcal{S}_t}(\boldsymbol{\theta}_t)]^2 \quad \text{with } \boldsymbol{v}_0 = \boldsymbol{0} \text{ and } \beta_2 \in (0,1).$$

Next ADAM preconditions the problem by scaling each gradient coordinate, and updates the variable

$$\boldsymbol{\theta}_{t+1} = \boldsymbol{\theta}_t - \eta \boldsymbol{m}_t/(1-\beta_1^t)/\left(\sqrt{\boldsymbol{v}_t/(1-\beta_2^t)} + \epsilon\right) \quad \text{with a small constant } \epsilon. \tag{3}$$

### 3.2  Lévy-driven SDEs

Let $\boldsymbol{u}_t = \nabla \boldsymbol{F}(\boldsymbol{\theta}_t) - \nabla f_{\mathcal{S}_t}(\boldsymbol{\theta}_t)$ denote gradient noise. From Sec. 3.1, we can formulate SGD as follows

$$\boldsymbol{\theta}_{t+1} = \boldsymbol{\theta}_t - \eta \nabla \boldsymbol{F}(\boldsymbol{\theta}_t) + \eta \boldsymbol{u}_t.$$

To analyze behaviors of an algorithm, one effective approach is to obtain its SDE via making assumptions on $\boldsymbol{u}_t$ and then analyze its SDE. For instance, to analyze SGD, most works [25–27, 34] assume that $\boldsymbol{u}_t$ obeys a Gaussian distribution $\mathcal{N}(\boldsymbol{0},\boldsymbol{\Sigma}_t)$ with covariance matrix

$$\boldsymbol{\Sigma}_t = \frac{1}{S}\left[\frac{1}{n}\sum_{i=1}^n \nabla f_i(\boldsymbol{\theta}_t)\nabla f_i(\boldsymbol{\theta}_t)^T - \nabla \boldsymbol{F}(\boldsymbol{\theta}_t)\nabla \boldsymbol{F}(\boldsymbol{\theta}_t)^T\right].$$

However, both recent work [23] and Fig. 1 show that the gradient noise $\boldsymbol{u}_t$ has heavy tails and can be better characterized by $\mathcal{S}\alpha\mathcal{S}$ distribution [24]. Moreover, the heavy-tail assumption can also better explain the behaviors of SGD than Gaussian noise assumption. Concretely, for the SDE of SGD on the one-dimensional problems, under Gaussian noise assumption its escaping time from a simple quadratic basin respectively exponentially and polynomially depends on the height and width of the basin [36], indicating that SGD gets stuck at deeper minima as opposed to wider/flatter minima. This contradicts with the observations in [1, 19–21] that SGD often converges to flat minima. By contrast, on the same problem, for Lévy-driven SDE, both [23] and this work show that SGD tends to converge to flat minima instead of deep minima, well explaining the convergence behaviors of SGD.

Following [23], we also assume $\boldsymbol{u}_t$ obeys $\mathcal{S}\alpha\mathcal{S}$ distribution but with a time-dependent covariance matrix $\boldsymbol{\Sigma}_t$ to better characterize the underlying structure in the gradient noise $\boldsymbol{u}_t$. In this way, when the learning rate $\eta$ is small and $\varepsilon = \eta^{(\alpha-1)/\alpha}$, we can write the Lévy-driven SDE of SGD as

$$\mathsf{d}\boldsymbol{\theta}_t = -\nabla \boldsymbol{F}(\boldsymbol{\theta}_t) + \varepsilon \boldsymbol{\Sigma}_t \mathsf{d}L_t. \tag{4}$$

Here the Lévy motion $L_t \in \mathbb{R}^d$ is a random vector and its $i$-th entry $L_{t,i}$ obeys the $\mathcal{S}\alpha\mathcal{S}(1)$ distribution which is defined through the characteristic function $\mathbb{E}[\exp(i\omega x)] = \exp(-\sigma^\alpha|\omega|^\alpha)$ if $x \sim \mathcal{S}\alpha\mathcal{S}(\sigma)$. Intuitively, the $\mathcal{S}\alpha\mathcal{S}$ distribution is a heavy-tailed distribution with a decay density like $1/|x|^{1+\alpha}$. When the tail index $\alpha$ is 2, $\mathcal{S}\alpha\mathcal{S}(1)$ becomes a Gaussian distribution and thus has stronger data-fitting capacity over Gaussian distribution. In this sense, the SDE of SGD in [25–27, 34, 37] is actually a special case of the Lévy-driven SDE in this work. Moreover, Eqn. (4) extends the one-dimensional SDE of SGD in [23]. Note, Eqn. (4) differs from [29], since it considers dynamic covariance matrix $\boldsymbol{\Sigma}_t$ in gradient noise and shows great effects of its underlying structure to the escaping behaviors in both ADAM and SGD, while [29] analyzed escaping behaviors of SGD along several fixed directions.

Similarly, we can derive the SDE of ADAM. For brevity, we define $\boldsymbol{m}_t' = \beta_1 \boldsymbol{m}_{t-1}' + (1-\beta_1)\nabla \boldsymbol{F}(\boldsymbol{\theta}_t)$ with $\boldsymbol{m}_0' = \boldsymbol{0}$. Then by the definitions of $\boldsymbol{m}_t$ and $\boldsymbol{m}_t'$, we can compute

$$\boldsymbol{m}_t' - \boldsymbol{m}_t = (1-\beta_1)\sum_{i=0}^t \beta_1^{t-i}[\nabla \boldsymbol{F}(\boldsymbol{\theta}_i) - \nabla f_{\mathcal{S}_i}(\boldsymbol{\theta}_i)] = (1-\beta_1)\sum_{i=0}^t \beta_1^{t-i}\boldsymbol{u}_i.$$

As noise $u_t$ has heavy tails, their exponential average should have similar behaviors, which is also illustrated by Fig. 1. So we also assume $\frac{1}{1-\beta_1^t}(m_t' - m_t)$ obeys $\mathcal{S}\alpha\mathcal{S}(1)$ distribution with covariance matrix $\Sigma_t$. Meanwhile, we can write ADAM as

$$\theta_{t+1} = \theta_t - \eta m_t'/z_t + \eta(m_t' - m_t)/z_t \quad \text{with} \quad z_t = (1-\beta_1^t)\left(\sqrt{v_t/(1-\beta_2^t)} + \epsilon\right).$$

So we can derive the Lévy-driven SDE of ADAM:

$$\mathsf{d}\theta_t = -\mu_t Q_t^{-1} m_t + \varepsilon Q_t^{-1}\Sigma_t \mathsf{d}L_t, \;\; \mathsf{d}m_t = \beta_1(\nabla F(\theta_t) - m_t), \;\; \mathsf{d}v_t = \beta_2([\nabla f_{\mathcal{S}_t}(\theta_t)]^2 - v_t), \;\; (5)$$

where $\varepsilon = \eta^{(\alpha-1)/\alpha}$, $Q_t = \mathrm{diag}\left(\sqrt{\omega_t v_t} + \epsilon\right)$, $\mu_t = 1/(1 - e^{-\beta_1 t})$ and $\omega_t = 1/(1 - e^{-\beta_2 t})$ are two constants to correct the bias in $m_t$ and $v_t$. Note, here we replace $m_t'$ with $m_t$ for brevity. Appendix A provides more construction details, randomness discussion and shows the fitting capacity of this SDE to ADAM. Subsequently, we will analyze escaping behaviors of the SDEs in Eqns. (4) and (5).

# 4    Analysis for Escaping Local Minima

Now we analyze the local stability of ADAM-alike adaptive algorithms and SGD. Suppose the process $\theta_t$ in Eqns. (4) and (5) starts at a local basin $\Omega$ with a minimum $\theta^*$, i.e. $\theta_0 \in \Omega$. Here we are particularly interested in the first escaping time $\Gamma$ of $\theta_t$ produced by an algorithm which reveals the convergence behaviors and generalization performance of the algorithm. Formally, let $\Omega^{-\varepsilon^\gamma} = \{y \in \Omega \mid \mathsf{dis}(\partial\Omega, y) \geq \varepsilon^\gamma\}$ denote the inner part of $\Omega$. Then we give two important definitions, i.e. (1) the escaping time $\Gamma$ of the process $\theta_t$ from the local basin $\Omega$ and (2) the escaping set $\mathcal{W}$ at $\Omega$, as

$$\Gamma = \inf\{t \geq 0 \mid \theta_t \notin \Omega^{-\varepsilon^\gamma}\} \quad \text{and} \quad \mathcal{W} = \{y \in \mathbb{R}^d \mid Q_{\theta^*}^{-1}\Sigma_{\theta^*}y \notin \Omega^{-\varepsilon^\gamma}\}, \qquad (6)$$

where the constant $\gamma > 0$ satisfies $\lim_{\varepsilon \to 0} \varepsilon^\gamma = 0$, $\Sigma_{\theta^*} = \lim_{\theta_t \to \theta^*}\Sigma_t$ for both SGD and ADAM, and $Q_{\theta^*} = I$ for SGD and $Q_{\theta^*} = \lim_{\theta_t \to \theta^*}Q_t$ for ADAM. Then we define Radon measure [38].

**Definition 1.** *If a measure $m(\mathcal{V})$ defined on Hausdorff topological space $\mathcal{X}$ obeys (1) inner regular, i.e. $m(\mathcal{V}) = \sup_{\mathcal{U} \subseteq \mathcal{V}} m(\mathcal{U})$, (2) outer regular, i.e. $m(\mathcal{V}) = \inf_{\mathcal{V} \subseteq \mathcal{U}} m(\mathcal{U})$, and (3) local finiteness, i.e. every point of $\mathcal{X}$ has a neighborhood $\mathcal{U}$ with finite $m(\mathcal{U})$, then $m(\mathcal{V})$ is a Radon measure.*

Then we define non-zero Radon measure which further obeys $m(\mathcal{U}) < m(\mathcal{V})$ if $\mathcal{U} \subset \mathcal{V}$. Since larger set has larger volume, $m(\mathcal{U})$ positively depends on the volume of the set $\mathcal{U}$. Let $m(\mathcal{W})$ be a non-zero Radon measure on the set $\mathcal{W}$. Then we first introduce two mild assumptions for analysis.

**Assumption 1.** *For both ADAM and SGD, suppose the objective $F(\theta)$ is a upper-bounded non-negative loss, and is locally $\mu$-strongly convex and $\ell$-smooth in the basin $\Omega$.*

**Assumption 2.** *For ADAM, suppose its process $\theta_t$ satisfies $\int_0^\Gamma \langle \frac{\nabla F(\theta_s)}{1+F(\theta_s)}, \mu_s Q_s^{-1}m_s \rangle ds \geq 0$ almost sure, and its parameters $\beta_1$ and $\beta_2$ obey $\beta_1 \leq \beta_2 \leq 2\beta_1$. Moreover, for ADAM, we assume $\|m_t - \widehat{m}_t\| \leq \tau_m \|\int_0^{t^-}(m_s - \widehat{m}_s)ds\|$ and $\|\widehat{m}_t\| \geq \tau\|\nabla F(\widehat{\theta}_t)\|$ where $\widehat{m}_t$ and $\widehat{\theta}_t$ are obtained by Eqn. (5) with $\varepsilon = 0$. Each coordinate $v_{t,i}$ of $v_t$ in ADAM obeys $v_{\min} \leq \sqrt{v_{t,i}} \leq v_{\max}$ ($\forall i, t$).*

Assumption 1 is very standard for analyzing stochastic optimization [39–44] and network analysis [45–48]. In Assumption 2, we indeed require similar directions of gradient estimate $m_t$ and full gradient $\nabla F(\theta_t)$ in ADAM in most cases, as we assume their inner product is non-negative along the iteration trajectory. So this assumption can be satisfied in practice. To analyze the processes $\theta_t$ and $\widehat{\theta}_t$ in ADAM, we make an assumption on the distance between their corresponding gradient estimates $m_t$ and $\widehat{m}_t$ which can be easily fulfilled by their definitions. Then

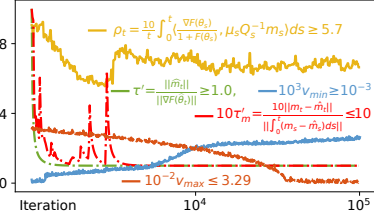

Figure 2: Empirical investigation of Assumption 2 on ADAM.

for ADAM, we mildly assume its estimated $v_t$ to be bounded. For $v_{\min}$, we indeed allow $v_{\min} = 0$ because of the small constant $\epsilon$. The relation $\beta_1 \leq \beta_2 \leq 2\beta_1$ is also satisfied under the default setting of ADAM. Actually, we also empirically investigate Assumption 2 on ADAM. In Fig. 2, we report the values of $\rho_t = \frac{10}{t}\int_0^\Gamma \langle \frac{\nabla F(\theta_s)}{1+F(\theta_s)}, \mu_s Q_s^{-1}m_s \rangle ds$, $\tau_m' = \frac{\|m_t - \widehat{m}_t\|}{\|\int_0^{t^-}(m_s - \widehat{m}_s)ds\|}$, $\tau' = \frac{\|\widehat{m}_t\|}{\|\nabla F(\widehat{\theta}_t)\|}$, $v_{\min} = \min_i \sqrt{v_{t,i}}, v_{\max} = \max_i \sqrt{v_{t,i}}$ in the SDE of ADAM on the 4-layered fully connected networks with width 20. Note that we scale some values of $\rho_t, \tau_m', \tau', v_{\min}$ and $v_{\max}$ so that we can

plot them in one figures. From Fig. 2, one can observe that $\rho_t, \tau'$ and $v_{\min}$ are well lower bounded, and $\tau'_m$ and $v_{\max}$ are well upper bounded. These results demonstrate the validity of Assumption 2.

With these two assumptions, we analyze the escaping time $\Gamma$ of process $\boldsymbol{\theta}_t$ and summarize the main results in Theorem 1. For brevity, we define a group of key constants for SGD and ADAM: $\kappa_1 = \ell$ and $\kappa_2 = 2\mu$ in SGD, $\kappa_1 = \frac{c_1 \ell}{(v_{\min}+\epsilon)|\tau_m - 1|}$ and $\kappa_2 = \frac{2\mu\tau}{\beta_1(v_{\max}+\epsilon)+\mu\tau}(\beta_1 - \frac{\beta_2}{4})$ in ADAM with a constant $c_1$.

**Theorem 1.** *Suppose Assumptions 1 and 2 hold. Let $\Theta(\varepsilon^{-1}) = \frac{2}{\alpha}\varepsilon^\alpha$, $\rho_0 = \frac{1}{16(1+c_2\kappa_1)}$ and $\ln\left(\frac{2\Delta}{\mu\varepsilon^{1/3}}\right) \leq \kappa_2 \varepsilon^{-\frac{1}{3}}$ with $\Delta = \boldsymbol{F}(\boldsymbol{\theta}_0) - \boldsymbol{F}(\boldsymbol{\theta}^*)$ and a constant $c_2$. Then for any $\boldsymbol{\theta}_0 \in \boldsymbol{\Omega}^{-2\varepsilon^\gamma}$, $u > -1$, $\varepsilon \in (0, \varepsilon_0]$, $\gamma \in (0, \gamma_0]$ and $\rho \in (0, \rho_0]$ satisfying $\varepsilon^\gamma \leq \rho_0$ and $\lim_{\varepsilon \to 0} \rho = 0$, SGD in (4) and ADAM in (5) obey*

$$\frac{1-\rho}{1+u+\rho} \leq \mathbb{E}\left[\exp\left(-um(\boldsymbol{\mathcal{W}})\Theta(\varepsilon^{-1})\Gamma\right)\right] \leq \frac{1+\rho}{1+u-\rho}.$$

See its proof in Appendix D.1. By setting $\varepsilon$ small, Theorem 1 shows that for both ADAM and SGD, the upper and lower bounds of their expected escaping time $\Gamma$ are at the order of $\mathcal{O}\left(\frac{1}{m(\boldsymbol{\mathcal{W}})\Theta(\varepsilon^{-1})}\right)$. Note, $m(\boldsymbol{\mathcal{W}})$ has different values for SGD and ADAM due to their different $\boldsymbol{Q}_{\boldsymbol{\theta}^*}$ in Eqn. (6). If the escaping time $\Gamma$ is very large, it means that the algorithm cannot well escape from the basin $\boldsymbol{\Omega}$ and would get stuck in $\boldsymbol{\Omega}$. Moreover, given the same basin $\boldsymbol{\Omega}$, if one algorithm has smaller escaping time $\Gamma$ than other algorithms, then it is more locally unstable and would faster escape from this basin to others. In the following sections, we discuss the effects of the geometry adaptation and the gradient noise structure of ADAM and SGD to the escaping time $\Gamma$ which are respectively reflected by the factors $m(\boldsymbol{\mathcal{W}})$ and $\Theta(\varepsilon^{-1})$. Our results show that SGD has smaller escaping time than ADAM and can better escape from local basins with small Radon measure to those with lager Radon measure.

## 4.1 Preference to Flat Minima

To interpret Theorem 1, we first define the *"flat" minima* in this work in terms of Radon measure.

**Definition 2.** *A minimum $\boldsymbol{\theta}^* \in \boldsymbol{\Omega}$ is said to be flat if its basin $\boldsymbol{\Omega}$ has large nonzero Radon measure.*

Due to the factor $m(\boldsymbol{\mathcal{W}})$ in Theorem 1, both ADAM and SGD have large escaping time $\Gamma$ at the "flat" minima. Specifically, if the basin $\boldsymbol{\Omega}$ has larger Radon measure, then the complementary set $\boldsymbol{\mathcal{W}}^c = \{\boldsymbol{y} \in \mathbb{R}^d \mid \boldsymbol{Q}_{\boldsymbol{\theta}^*}^{-1}\boldsymbol{\Sigma}_{\boldsymbol{\theta}^*}\boldsymbol{y} \in \boldsymbol{\Omega}^{-\varepsilon^\gamma}\}$ of $\boldsymbol{\mathcal{W}}$ also has larger Radon measure. Meanwhile, the Radon measure on $\boldsymbol{\mathcal{W}}^c \cup \boldsymbol{\mathcal{W}}$ is a constant, meaning the larger $m(\boldsymbol{\mathcal{W}}^c)$ the smaller $m(\boldsymbol{\mathcal{W}})$. So ADAM and SGD have larger escaping time at "flat" minima. Thus, they would escape "sharp" minima due to their smaller escaping time, and tend to converge to "flat" ones. Since for basin $\boldsymbol{\Omega}$, its Radon measure positively relies on its volume, $m(\boldsymbol{\mathcal{W}})$ negatively depends on the volume of $\boldsymbol{\Omega}$. So ADAM and SGD are more stable at the minima with larger basin $\boldsymbol{\Omega}$ in terms of volume. This can be intuitively understood: for the process $\boldsymbol{\theta}_t$, the volume of the basin determines the necessary jump size of the Lévy motion $L_t$ in the SDEs to escape, which means the larger the basin the harder for an algorithm to escape.

Note the "flat" minima here is defined on Radon measure, and differ from the conventional flat ones whose local basins have no large curvature (no large eigenvalues in its Hessian matrix). In most cases, the flat minima here consist of the conventional flat ones and the minma at the asymmetric basins/valleys since local basins of these minima often have large volumes and thus larger Radon measures. Accordingly, our theoretical results can well explain the phenomenons observed in many works [2, 12, 17, 22, 30, 31] that SGD often converges to the minima at the flat or asymmetric valleys which is interpreted by our theory to have larger Radon measure and attract SGD to stay at these places. In contrast, the conventional flat argument cannot explain asymmetric valleys, as asymmetric valleys means sharp minima under the conventional definition and should be avoided by SGD.

## 4.2 Analysis of Generalization Gap between ADAM and SGD

Theorem 1 can also well explain the generalization gap between ADAM-alike adaptive algorithms and SGD. That is, compared with SGD, the minima given by ADAM often suffer from worse test performance [12, 17, 18, 30, 31]. On one hand, the observations in [1, 19–21] show that the minima at the flat or asymmetric basins/valleys often enjoy better generalization performance than others. On the other hand, Theorem 1 shows that ADAM and SGD can escape sharp minima to flat ones with larger Radon measure. As aforementioned, flat minima in terms of Radon measure often refer to the minima at the flat or asymmetric basins/valleys. This implies that if one algorithm can escape the

current minima faster, it is more likely for the algorithm to find flatter minima. These results together show the benefit of faster escaping behaviors of an algorithm to its generalization performance.

According to Theorem 1, two main factors, i.e. the gradient noise and geometry adaptation respectively reflected by the factors $\Theta(\varepsilon^{-1}) = \frac{2}{\alpha}\varepsilon^{-\alpha}$ and $m(\mathcal{W})$, affects the escaping time $\Gamma$ of both ADAM and SGD. We first look at the factor $\Theta(\varepsilon^{-1})$ in the escaping time $\Gamma$. As illustrated in Fig. 3 in Sec. 5, the gradient noise in SGD enjoys very similar tail index $\alpha$ with ADAM for most optimization iterations, but it has much smaller tail index $\alpha$ than ADAM for some iterations, which means SGD has larger Lévy discontinuous jumps in these iterations and thus enjoys smaller escaping time $\Gamma$. This different tail property of gradient noise in these algorithms are caused by the following reason. SGD assumes the gradient noise $\boldsymbol{u}_t = \nabla \boldsymbol{F}(\boldsymbol{\theta}_t) - \nabla f_{\mathcal{S}_t}(\boldsymbol{\theta}_t)$ at one iteration has heavy tails, while ADAM considers the exponential gradient noise $\frac{1-\beta_1}{1-\beta_1^t}\sum_{i=0}^{t}\beta_1^{t-i}\boldsymbol{u}_i$ which indeed smooths gradient noise over the iteration trajectory and prevents large occasional gradient noise. In this way, SGD reveals heavier tails of gradient noise than ADAM and thus has smaller tail index $\alpha$ for some optimization iterations, helping escaping behaviors. Moreover, to guarantee convergence, ADAM needs to use smaller learning rate $\eta$ than SGD due to the geometry adaptation in ADAM, *e.g.* default learning rate $10^{-3}$ in ADAM and $10^{-2}$ in SGD, leading to smaller $\varepsilon = \eta^{(\alpha-1)/\alpha}$ and thus larger escaping time $\Gamma$ in ADAM. Thus, compared with ADAM, SGD is more locally unstable and will converge to flatter minima which often locate at the flat or asymmetric basins/valleys and enjoy better generalization performance [1, 19–21].

Besides, the factor $m(\mathcal{W})$ also plays an important role in the generalization degeneration phenomenon of ADAM. W.o.l.g., assume the minimizer $\boldsymbol{\theta}^* = \boldsymbol{0}$ in the basin $\boldsymbol{\Omega}$. As the local basin $\boldsymbol{\Omega}$ is often small, following [34, 49] we adopt second-order Taylor expansion to approximate $\boldsymbol{\Omega}$ as a quadratic basin with center $\boldsymbol{\theta}^*$, i.e. $\boldsymbol{\Omega} = \left\{\boldsymbol{y} \mid \boldsymbol{F}(\boldsymbol{\theta}^*) + \frac{1}{2}\boldsymbol{y}^T \boldsymbol{H}(\boldsymbol{\theta}^*)\boldsymbol{y} \leq h(\boldsymbol{\theta}^*)\right\}$ with a basin height $h(\boldsymbol{\theta}^*)$ and Hessian matrix $\boldsymbol{H}(\boldsymbol{\theta}^*)$ at $\boldsymbol{\theta}^*$. Then for SGD, since $\boldsymbol{Q}_{\boldsymbol{\theta}^*} = \boldsymbol{I}$ in Eqn. (6), its corresponding escaping set $\mathcal{W}$ is

$$\mathcal{W}_{\text{SGD}} = \left\{\boldsymbol{y} \in \mathbb{R}^d \mid \boldsymbol{y}^T \boldsymbol{\Sigma}_{\boldsymbol{\theta}^*} \boldsymbol{H}(\boldsymbol{\theta}^*)\boldsymbol{\Sigma}_{\boldsymbol{\theta}^*}\boldsymbol{y} \geq h_f^*\right\} \tag{7}$$

with $h_f^* = 2(h(\boldsymbol{\theta}^*) - \boldsymbol{F}(\boldsymbol{\theta}^*))$, while according to Eqn. (6), ADAM has escaping set

$$\mathcal{W}_{\text{ADAM}} = \{\boldsymbol{y} \in \mathbb{R}^d \mid \boldsymbol{y}^T \boldsymbol{\Sigma}_{\boldsymbol{\theta}^*} \boldsymbol{Q}_{\boldsymbol{\theta}^*}^{-1} \boldsymbol{H}(\boldsymbol{\theta}^*)\boldsymbol{Q}_{\boldsymbol{\theta}^*}^{-1}\boldsymbol{\Sigma}_{\boldsymbol{\theta}^*}\boldsymbol{y} \geq h_f^*\}. \tag{8}$$

Then we prove that for most time interval except the jump time, the current variable $\boldsymbol{\theta}_t$ is indeed close to the minimum $\boldsymbol{\theta}^*$. Specifically, we first decompose the Lévy process $L_t$ into two components $\boldsymbol{\xi}_t$ and $\boldsymbol{\zeta}_t$, i.e. $L_t = \boldsymbol{\xi}_t + \boldsymbol{\zeta}_t$, with the jump sizes $\|\boldsymbol{\xi}_t\| < \varepsilon^{-\delta}$ and $\|\boldsymbol{\zeta}_t\| \geq \varepsilon^{-\delta}$ ($\delta \in (0,1)$). In this way, the stochastic process $\boldsymbol{\xi}$ does not departure from $\boldsymbol{\theta}_t$ a lot due to its limited jump size. The process $\boldsymbol{\zeta}$ is a compound Poisson process with intensity $\Theta(\varepsilon^{-\delta}) = \int_{\|\boldsymbol{y}\| \geq \varepsilon^{-\delta}} \nu(\mathrm{d}\boldsymbol{y}) = \int_{\|\boldsymbol{y}\| \geq \varepsilon^{-\delta}} \frac{\mathrm{d}\boldsymbol{y}}{\|\boldsymbol{y}\|^{1+\alpha}} = \frac{2}{\alpha}\varepsilon^{\alpha\delta}$ and jumps distributed according to the law of $1/\Theta(\varepsilon^{-\delta})$. Specifically, let $0 = t_0 < t_1 < \cdots < t_k < \cdots$ denote the time of successive jumps of $\boldsymbol{\zeta}$. Then the inner-jump time intervals $\sigma_k = t_k - t_{k-1}$ are i.i.d. exponentially distributed random variables with mean value $\mathbb{E}(\sigma_k) = \frac{1}{\Theta(\varepsilon^{-\delta})}$ and probability function $\mathbb{P}(\sigma_k \geq x) = \exp(-x\Theta(\varepsilon^{-\delta}))$. Based on this decomposition, we state our results in Theorem 2.

**Theorem 2.** *Suppose Assumptions 1 and 2 hold. Assume the process $\widehat{\boldsymbol{\theta}}_t$ is produced by setting $\varepsilon = 0$ in the Lévy-driven SDEs of SGD and ADAM.*

*(1) $\widehat{\boldsymbol{\theta}}_t$ exponentially converges to the minimizer $\boldsymbol{\theta}^*$ in $\boldsymbol{\Omega}$. Specifically, by defining $\Delta = \boldsymbol{F}(\boldsymbol{\theta}_0) - \boldsymbol{F}(\boldsymbol{\theta}^*)$, $\kappa_3 = \frac{2\mu\tau}{\beta_1(v_{\max}+\epsilon)+\mu\tau}\left(\beta_1 - \frac{\beta_2}{4}\right)$ in ADAM and $\kappa_3 = 2\mu$ in SGD, for any $\bar{\rho} > 0$, it satisfies*

$$\|\widehat{\boldsymbol{\theta}}_t - \boldsymbol{\theta}^*\|_2^2 \leq \varepsilon^{\bar{\rho}} \quad \text{if } t \geq v_\varepsilon \triangleq \kappa_3^{-1}\ln\left(2\Delta\mu^{-1}\varepsilon^{-\bar{\rho}}\right).$$

*(2) Assume $\delta \in (0,1)$, $p = \min((\bar{\rho}(1+c_3\kappa_1))/4, \bar{p})$, $\bar{\rho} = \frac{1-\delta}{16(1+c_4\kappa_1)}$, where $\kappa_1$ (in Theorem 1), $\bar{p}$, $c_3$ and $c_4$ are four positive constants. When $\boldsymbol{\theta}_t$ and $\widehat{\boldsymbol{\theta}}_t$ have the same initialization $\boldsymbol{\theta}_0 = \widehat{\boldsymbol{\theta}}_0$, we have*

$$\sup_{\boldsymbol{\theta}_0 \in \boldsymbol{\Omega}} \mathbb{P}\left(\sup_{0 \leq t < \sigma_1} \|\boldsymbol{\theta}_t - \widehat{\boldsymbol{\theta}}_t\|_2 \geq 2\varepsilon^{\bar{\rho}}\right) \leq 2\exp\left(-\varepsilon^{-p}\right).$$

See its proof in Appendix D.2. By inspecting the first part of Theorem 2, one can observe that the gradient-noise-free processes $\widehat{\boldsymbol{\theta}}_t$ produced by setting $\varepsilon = 0$ in the Lévy-driven SDEs of SGD and ADAM locate in a very small neighborhood of the minimizer $\boldsymbol{\theta}^*$ in the local basin $\boldsymbol{\Omega}$ after a very small time interval $v_\varepsilon = \kappa_3^{-1}\ln\left(2\Delta\mu^{-1}\varepsilon^{-\bar{\rho}}\right)$. The second part of Theorem 2 shows that before the first jump time $t_1 = \sigma_1$ of the jump $\boldsymbol{\zeta}$ with size larger than $\varepsilon^{-\delta}$ in Lévy motion $L_t$, the distance

between $\boldsymbol{\theta}_t$ and $\widehat{\boldsymbol{\theta}}_t$ is very small. So these two parts together guarantee small distance between $\boldsymbol{\theta}_t$ and $\boldsymbol{\theta}^*$ for the most time interval before the first big jump in the Lévy motion $L_t$ since the mean jump time $\mathbb{E}(\sigma_1) = \frac{\alpha}{2\varepsilon^{\alpha\delta}} = \mathcal{O}(\varepsilon^{-1})$ of the first big jump is much larger than $v_\varepsilon = \mathcal{O}(\ln(\varepsilon^{-1}))$ when $\varepsilon$ is small. Next after the first big jump, if $\boldsymbol{\theta}_t$ does not escape from the local basin $\boldsymbol{\Omega}$, by using the first part of Theorem 2, after the time interval $v_\varepsilon$, $\boldsymbol{\theta}_t$ becomes close to $\boldsymbol{\theta}^*$ again. This process will continue until the algorithm escapes from the basin. So for most time interval before escaping from $\boldsymbol{\Omega}$, the stochastic process $\boldsymbol{\theta}_t$ locates in a very small neighborhood of the minimizer $\boldsymbol{\theta}^*$.

The above analysis results on Theorem 2 hold for moderately ill-conditioned local basins (ICLBs). Specifically, the analysis requires $v_\varepsilon \le \sigma_1$ to guarantee small distance of current solution $\boldsymbol{\theta}_t$ to $\boldsymbol{\theta}^*$ before each big jump. So if $\mu$ of ICLBs is larger than $\mathcal{O}(\varepsilon^{\alpha\delta})$ which is very small as $\varepsilon$ in SDE is often small to precisely mimic algorithm behaviors, The above analysis results 2 still hold. Moreover, to obtain the result (1) in Theorem 2, we assume the optimization trajectory goes along the eigenvector direction corresponding to $\mu$ which is the worse case and leads to the worst convergence speed. As the measure of one/several eigenvector directions on high dimension is 0, optimization trajectory cannot always go along the eigenvector direction corresponding to $\mu$. So $v_\varepsilon$ is actually much larger than $\mathcal{O}\big(\frac{1}{\mu}\ln(\frac{1}{\mu\varepsilon^\delta})\big)$, largely improving applicability of our theory. For extremely ICLBs ($\mu \to 0$ or $\mu = 0$), the above analysis does not hold which accords with the previous results that first-order gradient algorithms cannot escape from them provably [50]. Fortunately, $\mu \to 0$ and $\mu = 0$ give asymmetric basins which often generalize well [2, 22] and are not needed to escape.

By using the above results, we have $\boldsymbol{\theta}_t \approx \boldsymbol{\theta}^*$ before escaping and thus $\boldsymbol{v}_t = \lim_{\boldsymbol{\theta}_t \to \boldsymbol{\theta}^*}[\nabla f_{\mathcal{S}_t}(\boldsymbol{\theta}_t)]^2$. Considering the randomness of the mini-batch $\mathcal{S}_t$, $\omega_t \approx 1$ and $\epsilon \approx 0$, we can approximate

$$\mathbb{E}[\boldsymbol{Q}_{\boldsymbol{\theta}^*}] \approx \mathbb{E}\big[\lim_{\boldsymbol{\theta}_t \to \boldsymbol{\theta}^*} \mathrm{diag}\big(\sqrt{\omega_t \boldsymbol{v}_t}\big)\big] \approx \mathrm{diag}\left(\sqrt{\frac{1}{n}\sum_{i=1}^n [\nabla f_i(\boldsymbol{\theta}^*)]^2}\right).$$

Meanwhile, since $\boldsymbol{\Sigma}_{\boldsymbol{\theta}^*} = \frac{1}{S}\bar{\boldsymbol{\Sigma}}_{\boldsymbol{\theta}^*}$ because of $\lim_{\boldsymbol{\theta}_t \to \boldsymbol{\theta}^*} \boldsymbol{F}(\boldsymbol{\theta}_t) = \boldsymbol{0}$ where $\bar{\boldsymbol{\Sigma}}_{\boldsymbol{\theta}^*} = \frac{1}{n}\sum_{i=1}^n \nabla f_i(\boldsymbol{\theta}^*)\nabla f_i(\boldsymbol{\theta}^*)^T$, one can approximately compute $\mathbb{E}[\boldsymbol{\Sigma}_{\boldsymbol{\theta}^*}\boldsymbol{Q}_{\boldsymbol{\theta}^*}^{-1}] \approx \frac{1}{S}\boldsymbol{I}$. Plugging this result into the escaping set $\boldsymbol{\mathcal{W}}_{\text{ADAM}}$ yields

$$\boldsymbol{\mathcal{W}}_{\text{ADAM}} \approx \Big\{\boldsymbol{y} \in \mathbb{R}^d \,\big|\, \boldsymbol{y}^T \boldsymbol{H}(\boldsymbol{\theta}^*)\boldsymbol{y} \ge S^2 h_f^*\Big\}.$$

Now we compare the escaping sets $\boldsymbol{\mathcal{W}}_{\text{SGD}}$ of SGD and $\boldsymbol{\mathcal{W}}_{\text{ADAM}}$ of ADAM. For clarity, we re-write $\boldsymbol{\mathcal{W}}_{\text{SGD}}$ in Eqn. (7) as

$$\boldsymbol{\mathcal{W}}_{\text{SGD}} = \Big\{\boldsymbol{y} \in \mathbb{R}^d \,\big|\, \boldsymbol{y}^T \bar{\boldsymbol{\Sigma}}_{\boldsymbol{\theta}^*} \boldsymbol{H}(\boldsymbol{\theta}^*)\bar{\boldsymbol{\Sigma}}_{\boldsymbol{\theta}^*}\boldsymbol{y} \ge S^2 h_f^*\Big\}.$$

By comparison, one can observe that for ADAM, its gradient noise does not affect the escaping set $\boldsymbol{\mathcal{W}}_{\text{ADAM}}$ due to the geometry adaptation via scaling each gradient coordinate, while for SGD, its gradient noise plays an important role. Suppose $\bar{\boldsymbol{H}}(\boldsymbol{\theta}^*) = \bar{\boldsymbol{\Sigma}}_{\boldsymbol{\theta}^*} \boldsymbol{H}(\boldsymbol{\theta}^*)\boldsymbol{\Sigma}_{\boldsymbol{\theta}^*}$, and the singular values of $\boldsymbol{H}(\boldsymbol{\theta}^*)$ and $\bar{\boldsymbol{\Sigma}}_{\boldsymbol{\theta}^*}$ are respectively $\lambda_1 \ge \lambda_2 \ge \cdots \ge \lambda_d$ and $\varsigma_1 \ge \varsigma_2 \ge \cdots \ge \varsigma_d$. Zhu *et al.* [34] proved that $\bar{\boldsymbol{\Sigma}}_{\boldsymbol{\theta}^*}$ of SGD on deep neural networks well aligns the Hessian matrix $\boldsymbol{H}(\boldsymbol{\theta}^*)$, namely the top eigenvectors associated with large eigenvalues in $\boldsymbol{\Sigma}_{\boldsymbol{\theta}^*}$ have similar directions in those in $\boldsymbol{H}(\boldsymbol{\theta}^*)$. Besides, for modern over-parameterized neural networks, both Hessian $\boldsymbol{H}(\boldsymbol{\theta}^*)$ and the gradient covariance matrix $\boldsymbol{\Sigma}_{\boldsymbol{\theta}^*}$ are ill-conditioned and anisotropic near minima [22, 27]. Based on these results, we can approximate the singular values of $\bar{\boldsymbol{H}}(\boldsymbol{\theta}^*)$ as $\lambda_1\varsigma_1^2 \ge \lambda_2\varsigma_2^2 \ge \cdots \ge \lambda_d\varsigma_d^2$, implying that $\bar{\boldsymbol{H}}(\boldsymbol{\theta}^*)$ becomes much more singular than $\boldsymbol{H}(\boldsymbol{\theta}^*)$. Then the volume of the component set $\boldsymbol{\mathcal{W}}_{\text{ADAM}}^c$ of $\boldsymbol{\mathcal{W}}_{\text{ADAM}}$ is $\mathrm{V}(\boldsymbol{\mathcal{W}}_{\text{ADAM}}^c) = \zeta\prod_{i=1}^d \lambda_i$ where $\zeta = 2d^{-1}(\pi S/h_f^*)^{d/2}g^{-1}(d/2)$ with a gamma function $g$. Similarly, we can obtain the volume $\mathrm{V}(\boldsymbol{\mathcal{W}}_{\text{SGD}}^c) = \zeta\prod_{i=1}^d \lambda_i\varsigma_i^2$ of the component set $\boldsymbol{\mathcal{W}}_{\text{SGD}}^c$ of $\boldsymbol{\mathcal{W}}_{\text{SGD}}$. As aforementioned, covariance matrix $\boldsymbol{\Sigma}_{\boldsymbol{\theta}^*}$ is ill-conditioned and anisotropic near minima and has only a few larger singular values [22, 27], indicating $\prod_{i=1}^d \varsigma_i^2 \ll 1$. So $\mathrm{V}(\boldsymbol{\mathcal{W}}_{\text{SGD}}^c)$ is actually much smaller than $\mathrm{V}(\boldsymbol{\mathcal{W}}_{\text{ADAM}}^c)$. Hence $\boldsymbol{\mathcal{W}}_{\text{SGD}}$ has larger volume than $\boldsymbol{\mathcal{W}}_{\text{ADAM}}$ and thus has larger Radon measure $m(\boldsymbol{\mathcal{W}}_{\text{SGD}})$ than $m(\boldsymbol{\mathcal{W}}_{\text{ADAM}})$. Accordingly, SGD has smaller escaping time at the local basin $\boldsymbol{\Omega}$ than ADAM. Thus, SGD would escape from $\boldsymbol{\Omega}$ and converges to flat minima whose local basins have large Radon measure, while ADAM will get stuck in $\boldsymbol{\Omega}$. Since flat minima with large Radon measure usually locate at the flat or asymmetric basins/valleys and generalize better [12, 17, 30, 31, 51], SGD often enjoys better testing performance. From the above analysis, one can also observe that for SGD, the covariance matrix $\boldsymbol{\Sigma}_{\boldsymbol{\theta}^*}$ helps increase Radon measure $m(\boldsymbol{\mathcal{W}}_{\text{SGD}})$ of $\boldsymbol{\mathcal{W}}_{\text{SGD}}$. So anisotropic gradient noise helps SGD escape from the local basin but cannot help ADAM's escaping behaviors.

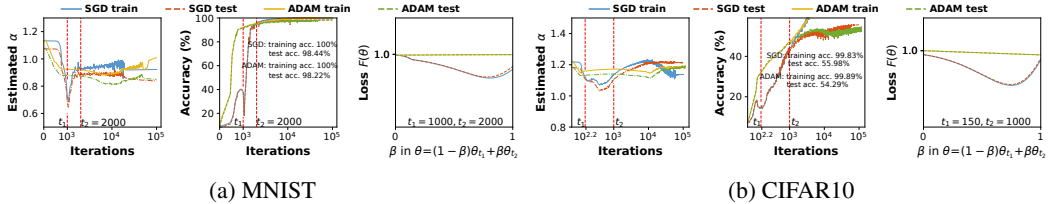

Figure 3: Behaviors illustration of SGD and ADAM on fully connected networks. In both (a) and (b), the left and middle figures respectively report the estimated tail index $\alpha$ in $\mathcal{S}\alpha\mathcal{S}$ distribution and classification accuracies; right figures show possible barriers between the solutions $\theta_{1000}$ and $\theta_{2000}$ on MNIST, and $\theta_{150}$ and $\theta_{1000}$ on CIFAR10, respectively. **Best viewed in $\times 2$ sized color pdf file.**

## 5 Experiments

In this section, we first investigate the gradient noise in ADAM and SGD, and then show their iteration-based convergence behaviors to testify the implications of our escaping theory. The code is available at https://panzhous.github.io.

**Heavy Tails of Gradient Noise.** We respectively use SGD and ADAM to train AlexNet [52] on CIFAR10, and show the statistical behaviors of gradient noise on CIFAR10. To fit the noise via $\mathcal{S}\alpha\mathcal{S}$ distribution, we consider covariance matrix $\Sigma_t$ and use the approach in [23, 53] to estimate the tail index $\alpha$. Fig. 1 in Sec. 1 and Fig. 4 in Appendix A show that the gradient noise in both SGD and ADAM usually reveal the heavy tails and can be well characterized by $\mathcal{S}\alpha\mathcal{S}$ distribution. This testifies the heavy tail assumption on the gradient noise in our theories.

**Escaping Behaviors.** We investigate the iteration-based convergence behaviors of SGD and ADAM, including their training and test accuracies and losses and tail index of their gradient noise. For MNIST [54] and CIFAR10 [55], we respectively use nine- and seven-layered fully-connected-networks. Each layer has 512 neurons and contains a linear layer and a ReLu layer. Firstly, the results in the middle figures show that SGD usually has better generalization performance than ADAM-alike adaptive algorithms which is consistent with the results in [12, 17, 18, 30].

Moreover, from the trajectories of the tail index $\alpha$ and accuracy of SGD on MNIST and CIFAR10 in Fig. 3, one can observe two distinct phases. Specifically, for the first 1000 iterations in MNIST and 150 iterations in CIFAR10, both the training and test accuracies increase tardily, while the tail index parameter $\alpha$ reduces quickly. This process continues until $\alpha$ reaches its lowest value. When considering the barrier around inflection point (*e.g.* a barrier between $\theta_{1000}$ and $\theta_{2000}$ on MNIST), it seems that the process of SGD has a sudden jump from one basin to another one which leads to a sudden accuracy drop, and then gradually converges. Accordingly, the accuracies are improved quickly. In contrast, one cannot observe similar phenomenon in ADAM. This is because as our theory suggested, SGD is more locally unstable and converges to flatter minima than ADAM, which is caused by the geometry adaptation, exponential gradient average and smaller learning rate in ADAM. All these results are consistent with our theories and also explain the well observed evidences in [12, 17, 30, 31, 51] that SGD usually converges to flat minima which often locate at the flat or asymmetric basins/valleys, while ADAM does not. Because the empirical observations [1, 19–21] show that minima at the flat or asymmetric basins/valleys often generalize better than sharp ones, our empirical and theoretical results can well explain the generalization gap between ADAM-alike algorithms and SGD.

## 6 Conclusion

In this work, we analyzed the generalization performance degeneration of ADAM-alike adaptive algorithms over SGD. By looking into the local convergence behaviors of the Lévy-driven SDEs of these algorithms through analyzing their escaping time, we prove that for the same basin, SGD has smaller escaping time than ADAM and tends to converge to flatter minima whose local basins have larger Radon measure, explaining its better generalization performance. This result is also consistent with the widely observed convergence behaviors of SGD and ADAM in many literatures. Finally our experimental results testify the heavy gradient noise assumption and implications in our theory.

## Broader Impacts

This work theoretically analyzes a fundamental problem in deep learning field, namely the generalization gap between adaptive gradient algorithms and SGD, and reveals the essential reasons for the generalization degeneration of adaptive algorithms. The established theoretical understanding of these algorithms may inspire new algorithms with both fast convergence speed and good generalization performance, which alleviate the need for computational resource and achieve state-of-the-art results. Yet it still needs more efforts to provide more insights to design practical algorithms.

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
