[Supplementary Material]

# Towards Theoretically Understanding Why SGD Generalizes Better Than ADAM in Deep Learning (Supplementary File)

**Pan Zhou**[*], **Jiashi Feng**[†], **Chao Ma**[‡], **Caiming Xiong**[*], **Steven HOI**[*], **Weinan E**[‡]
[*]Salesforce Research, [†] National University of Singapore, [‡] Princeton University
{pzhou,shoi,cxiong}@salesforce.com  elefjia@nus.edu.sg  {chaom@, weinan@math.}princeton.edu

This supplementary document contains the technical proofs of convergence results and some additional numerical results of the work entitled "Towards Theoretically Understanding Why SGD Generalizes Better Than ADAM in Deep Learning". It is structured as follows. In Appendix A, we provides more construction details of the SDE for ADAM and also conduct experiments which show very similar convergence behaviors of ADAM (SGD) and its SDE. Appendix B compares our work with the related work [1, 2] in more details. Appendix C summarizes the notations throughout this document and also provides the auxiliary theories and lemmas for subsequent analysis whose proofs are deferred to Appendix E. Then Appendix D gives the proofs of the main results in Sec. 4, including Theorem 1 which analyzes the escaping time analysis of Lévy-driven SDEs and Theorem 2 which proves the processes with and without Lévy motion are close to each other. Finally, in Appendix E we presents the proofs of auxiliary theories and lemmas in Appendix C, including Theorems $3 \sim 4$ and Lemmas $1 \sim 3$.

## A   More Discussion of SDE in ADAM

Here we provide more discussion and construction details for the SDE in ADAM. We first investigate the second order moment of the gradient noise in ADAM. Then we introduce the two types of randomness in the SDE of ADAM. Finally, we run experiments to investigate the validity of the constructed SDEs of ADAM and SGD.

### A.1   $\mathcal{S}\alpha\mathcal{S}$-distributed Gradient Noise in ADAM

In the manuscript, we have shown the gradient noise itself to be $\mathcal{S}\alpha\mathcal{S}$-distributed. Here we further investigate the second-order moment of the gradient noise. From the bottom row of Figure 4, one can observe that (1) both the second-order moment of the gradient noise also reveals heavy tails; (2) compared with Gaussian distribution, $\mathcal{S}\alpha\mathcal{S}$ distribution can better characterize this kind of second-order moment of the gradient noise. All these results demonstrate that the gradient noise in both ADAM and SGD actually satisfies the $\mathcal{S}\alpha\mathcal{S}$ distribution. So the heavy-tailed gradient noise assumptions in our manuscript is very reasonable.

### A.2   Randomness in SDE of ADAM

The SDE of ADAM approximates gradient noise $\boldsymbol{m}_t$ via the combination of full gradient and Lévy motion but does not approximate $\boldsymbol{v}_t$. This SDE should be more accurate than the one which approximates both $\boldsymbol{m}_t$ and the coefficients $\boldsymbol{v}_t$. So the randomness in the SDE of ADAM comes from the Lévy motion and also $\boldsymbol{v}_t$ caused by sampling a minibatch. But these two types of randomness actually do not depend on each other. Note that as shown in many literatures, *e.g.* [3, 4], SDE allows randomness in coefficients and also enjoys many good properties, such as stability and unique solution. This type of SDE is usually called "SDE with random coefficients", and usually appears in

(a) Gradient noise in ADAM        (b) Gradient noise in SGD

(c) Second-order moment of gradient noise in ADAM     (d) Second-order moment of gradient noise in SGD

Figure 4: Illustration of gradient noise in ADAM and SGD. The left figures in (a) and (b) are the real gradient noise computed with AlexNet on CIFAR10. Similarly, the left figures in (c) and (d) are the second-order moment of gradient noise computed with AlexNet on CIFAR10. The middle and right figures in (a) ∼ (d) are respectively the fitted Gaussian and systemic $\alpha$-stable noise. By comparison, $\alpha$-stable noise can better characterize real gradient noise in deep learning.

(a) ADAM

(b) SGD

Figure 5: Illustration of convergence trajectories of ADAM, SGD and their SDEs. One can observe that for ADAM, it convergence trajectories are very similar to its SDE, which shows the validity of the SDE construction. Similarly, we can observe the same observations on SGD and its SDE.

stochastic jump systems [5], economics and finance [6, 7], biology [8, 9], mechanics and physics [10], etc. See more details of SDE with random coefficients in [3, 4].

## A.3 Convergence Behavior Comparison between Algorithm and Its SDE

Here we conduct experiments on 784-10-10-sized networks and report the convergence behvariors of ADAM (SGD) and its SDE in Fig. 5. Note SDE actually equals to injecting heavy tailed noise into SGD and ADAM that use full gradients. We use a relatively small network since simulating high-dimensional gradient noise $u_t$ and computing the huge covariance matrix $\Sigma_t$ at each iteration are too computationally expensive to compute. From the convergence trajectories of both ADAM and its SDE in Fig. 5 (a), one can observe that they have very similar convergence behaviors. Similarly, in Fig. 5 (b) we can observe the same observations on SGD and its SDE. So injecting heavy tailed noise into SGD and ADAM that use full gradients leads to similar convergence behaviors to SGD and ADAM that use stochastic gradients. These results well demonstrate the validity of current SDE

construction. Note that here we do not observe jump behaviors, since the networks are very small and may have not very sharp minima. But these results as aforementioned can testify the validity of current SDE construction.

## B  Comparison to Related Works

Dinh et al. [1] showed flat minimum can become sharp by scaling two layers at the same time. But with this scaling, sharp minimum cannot be arbitrarily flat, as if the eigenvalues of two parameters in the same layer has large ratio, this scaling cannot change this ratio. So flat and sharp minimum are not totally equivalent. Combining the observation in many works that flat minima could achieve better generalization performance than sharper ones, one could conclude that flat minima can generalize well in most case, while sharp minima that can become flat one by linearly scaling two layers also can generalize but other sharp minima cannot. So analyzing the flat and sharp properties is still meaningful. Besides, the flatness in this work is defined on general non-zero Radon measure. If one finds an invariant measure to the scaling in [1], the flatness is also invariant, providing more insights to generalization. So it is promising to explore this invariant measure in the future.

## C  Notations and Auxiliary Lemmas

### C.1  Notations

For analyzing the uniform Lévy-driven SDEs in Eqn. (4) and (5), we first decompose the Lévyprocess $L_t$ into two components $\boldsymbol{\xi}_t$ and $\boldsymbol{\zeta}_t$, namely

$$L_t = \boldsymbol{\xi}_t + \boldsymbol{\zeta}_t \tag{7}$$

whose characteristic functions are respectively defined as

$$\mathbb{E}[e^{i\langle \boldsymbol{\lambda}, \boldsymbol{\xi}_t \rangle}] = e^{t \int_{\mathbb{R}^d \backslash \{\mathbf{0}\}} \zeta \mathbb{I}\{\|\boldsymbol{y}\| \leq \frac{1}{\varepsilon^\delta}\} \nu(\mathrm{d}\boldsymbol{y})},$$

$$\mathbb{E}[e^{i\langle \boldsymbol{\lambda}, \boldsymbol{\zeta}_t \rangle}] = e^{t \int_{\mathbb{R}^d \backslash \{\mathbf{0}\}} \zeta \mathbb{I}\{\|\boldsymbol{y}\| \geq \frac{1}{\varepsilon^\delta}\} \nu(\mathrm{d}\boldsymbol{y})},$$

where $\zeta = e^{i\langle \boldsymbol{\lambda}, \boldsymbol{y} \rangle} - 1 - i\langle \boldsymbol{\lambda}, \boldsymbol{y} \rangle \mathbb{I}\{\|\boldsymbol{y}\| \leq 1\}$, $\varepsilon$ (in Eqn. (4) and (5)) and $\delta$ are two small constants satisfying $\varepsilon^{-\delta} < 1$ and will be specified later. Define the Lévy measures $\nu$ as $\nu(\mathrm{d}\boldsymbol{y}) = \frac{1}{\|\boldsymbol{y}\|^{1+\alpha}} \mathrm{d}\boldsymbol{y}$. Accordingly, the Lévy measures $\nu$ of the stochastic processes $\boldsymbol{\xi}$ and $\boldsymbol{\zeta}$ are

$$\nu_{\boldsymbol{\xi}} = \nu\big(\boldsymbol{A} \cap \{0 < \|\boldsymbol{y}\| \leq \frac{1}{\varepsilon^\delta}\}\big), \ \nu_{\boldsymbol{\zeta}} = \nu\big(\boldsymbol{A} \cap \{\|\boldsymbol{y}\| \geq \frac{1}{\varepsilon^\delta}\}\big),$$

where $\boldsymbol{A} \in \mathcal{B}(\mathbb{R}^d)$. In this way, the stochastic process $\boldsymbol{\xi}$ has infinite Lévy measure with support $\{\boldsymbol{y} \mid 0 < \|\boldsymbol{y}\| \leq \varepsilon^{-\delta}\}$ and thus makes infinitely many jumps on any time interval. But the jump size does not exceed $\varepsilon^{-\delta}$ and thus is small which actually does not help escape the current local basin. In contrast, the Lévy measure $\nu_{\boldsymbol{\zeta}}(\cdot)$ of $\boldsymbol{\zeta}$ is finite and is computed as

$$\Theta(\varepsilon^{-\delta}) = \int_{\|\boldsymbol{y}\| \geq \varepsilon^{-\delta}} \nu(\mathrm{d}\boldsymbol{y}) = \int_{\|\boldsymbol{y}\| \geq \varepsilon^{-\delta}} \frac{\mathrm{d}\boldsymbol{y}}{\|\boldsymbol{y}\|^{1+\alpha}} = \frac{2}{\alpha} \varepsilon^{\alpha\delta}.$$

So the process $\boldsymbol{\zeta}$ is a compound Poisson process with intensity $\Theta(\varepsilon^{-\delta})$ and jumps distributed according to the law of $1/\Theta(\varepsilon^{-\delta})$. Specifically, let $0 = t_1 < t_2 \cdots < t_k < \cdots$ denote the times of successive jumps of $\boldsymbol{\zeta}$ and $J_k$ denote the jump size at the $k$-th jump. Then the inner-jump times $\sigma_k = t_k - t_{k-1}$ are i.i.d. exponentially distributed random variables with mean value $\mathbb{E}(\sigma_k) = \frac{1}{\Theta(\varepsilon^{-\delta})}$ and the probability distribution function $\mathbb{P}(\sigma_k \leq x) = 1 - \exp(-x\Theta(\varepsilon^{-\delta}))$. The probability law of $J_k$ is also known explicitly in terms of the Lévy measure $\nu$:

$$\mathbb{P}(J_k \in \boldsymbol{A}) = \frac{1}{\Theta(\varepsilon^{-\delta})} \nu\big(\boldsymbol{A} \cap \{\boldsymbol{y} \mid \|\boldsymbol{y}\| \geq \varepsilon^{-\delta}\}\big), \ \boldsymbol{A} \in \mathcal{B}(\mathbb{R}^d).$$

So the main force for escaping the local basin comes from the big jumps in the process $\boldsymbol{\zeta}$ which will be rigorous analyzed in the following sections.

Besides, for analysis, we usually need to consider affects of the Lévy motion (noise) $L_t$ to the Lévy-driven SDEs of SGD and ADAM given in Eqn. (4) and (5). So here we define two Lévy-free SDEs which respectively correspond to Eqn. (4) and (5):

$$\mathrm{d}\widehat{\boldsymbol{\theta}}_t = \nabla \boldsymbol{F}(\widehat{\boldsymbol{\theta}}_t), \tag{8}$$

and

$$\begin{cases} \mathrm{d}\widehat{\boldsymbol{\theta}}_t = & -\mu_t \widehat{\boldsymbol{Q}}_t^{-1}\widehat{\boldsymbol{m}}_t, \\ \mathrm{d}\widehat{\boldsymbol{m}}_t = & \beta_1(\nabla \boldsymbol{F}(\widehat{\boldsymbol{\theta}}_t) - \widehat{\boldsymbol{m}}_t) \\ \mathrm{d}\widehat{\boldsymbol{v}}_t = & \beta_2(\nabla f_{\mathcal{S}_t}(\widehat{\boldsymbol{\theta}}_t)^2 - \widehat{\boldsymbol{v}}_t). \end{cases} \tag{9}$$

where $\widehat{\boldsymbol{Q}}_t = \mathrm{diag}\left(\sqrt{\widehat{\boldsymbol{v}}_t} + \epsilon\right)$. Then by analyzing the distance $\|\widehat{\boldsymbol{\theta}}_t - \boldsymbol{\theta}_t\|$ between the processes $\widehat{\boldsymbol{\theta}}_t$ without Lévy motion and $\boldsymbol{\theta}_t$ with Lévy motion, we can well know the effects of the Lévy motion to the escaping behaviors.

## C.2 Auxiliary Theories and Lemmas

**Theorem 3.** *Suppose Assumptions 1 and 2 holds. Then for Lévy-driven SGD SDE (8) with $\widehat{\boldsymbol{Q}}_t = \boldsymbol{I}$ and $\beta_2 = 0$, the Lyapunov function $\mathcal{L}(t) = \boldsymbol{F}(\widehat{\boldsymbol{\theta}}_t) - \boldsymbol{F}(\boldsymbol{\theta}^*)$ obeys*

$$\mathcal{L}(t) \le \Delta \exp\left(-2\mu t\right)$$

*where $\Delta = \boldsymbol{F}(\widehat{\boldsymbol{\theta}}_0) - \boldsymbol{F}(\boldsymbol{\theta}^*)$ with the optimum solution $\boldsymbol{\theta}^*$ in the current local basin $\boldsymbol{\Omega}$. The sequence $\{\widehat{\boldsymbol{\theta}}_t\}$ produced by Eqn. (8) obeys*

$$\|\widehat{\boldsymbol{\theta}}_t - \boldsymbol{\theta}^*\|_2^2 \le \frac{2\Delta}{\mu}\exp\left(-2\mu t\right).$$

See its proof in Appendix E.1.

**Theorem 4.** *Suppose Assumptions 1 and 2 holds. Assume the sequence $\{(\widehat{\boldsymbol{\theta}}_t, \widehat{\boldsymbol{m}}_t, \widehat{\boldsymbol{v}}_t)\}$ are produced by Eqn. (9). Let $\widehat{\boldsymbol{s}}_t = \frac{h_t}{\mu_t}\left(\sqrt{\omega_t \widehat{\boldsymbol{v}}_t} + \epsilon\right)$ with $h_t = \beta_1$, $\mu_t = (1 - e^{-\beta_1 t})^{-1}$ and $\omega_t = (1 - e^{-\beta_2 t})^{-1}$. We define $\|\boldsymbol{x}\|_{\boldsymbol{y}}^2 = \sum_i \boldsymbol{y}_i \boldsymbol{x}_i^2$. Then for Lévy-driven ADAM SDEs in Eqn. (9), its Lyapunov function $\mathcal{L}(t) = \boldsymbol{F}(\widehat{\boldsymbol{\theta}}_t) - \boldsymbol{F}(\boldsymbol{\theta}^*) + \frac{1}{2}\|\widehat{\boldsymbol{m}}_t\|_{\widehat{\boldsymbol{s}}_t^{-1}}$ with the optimum solution $\boldsymbol{\theta}^*$ in the current local basin $\boldsymbol{\Omega}$ obeys*

$$\mathcal{L}(t) \le \Delta \exp\left(-\frac{2\mu\tau}{\beta_1(v_{\max} + \epsilon) + \mu\tau}\left(\beta_1 - \frac{\beta_2}{4}\right)t\right)$$

*where $\Delta = \boldsymbol{F}(\widehat{\boldsymbol{\theta}}_0) - \boldsymbol{F}(\boldsymbol{\theta}^*)$ due to $\widehat{\boldsymbol{m}}_0 = \boldsymbol{0}$. The sequence $\{\widehat{\boldsymbol{\theta}}_t\}$ produced by Eqn. (9) obeys*

$$\|\widehat{\boldsymbol{\theta}}_t - \boldsymbol{\theta}^*\|_2^2 \le \frac{2\Delta}{\mu}\exp\left(-\frac{2\mu\tau}{\beta_1(v_{\max} + \epsilon) + \mu\tau}\left(\beta_1 - \frac{\beta_2}{4}\right)t\right).$$

See its proof in Appendix E.2.

**Lemma 1.** *(1) The process $\boldsymbol{\xi}$ in Eqn. (7) can be decomposed into two processes $\widehat{\boldsymbol{\xi}}$ and linear drift, namely,*

$$\boldsymbol{\xi}_t = \widehat{\boldsymbol{\xi}}_t + \mu_\varepsilon t, \tag{10}$$

*where $\widehat{\boldsymbol{\xi}}$ is a zero mean Lévymartingale with bounded jumps.*
*(2) Let $\delta \in (0, 1)$, $\mu_\varepsilon = \mathbb{E}[\boldsymbol{\xi}_1]$ and $T_\varepsilon = \varepsilon^{-\theta}$ for some $\theta > 0$, $\rho_0 = \rho_0(\delta) = \frac{1-\delta}{4} > 0$ and $\theta_0 = \theta_0(\delta) = \frac{1-\delta}{3} > 0$. Suppose $\varepsilon$ is sufficient small such that such that $\Theta(1) \le \varepsilon^{-\frac{1-\delta}{6}}$ and $\varepsilon^{-\rho} - 2(C + \Theta(1))\varepsilon^{\frac{7}{6}(1-\delta) + \frac{\rho}{2}} \ge 1$ with a constant $C = \left|\int_{0 < u \le 1} u^2 \mathrm{d}\Theta(u)\right| \in (0, +\infty)$. Then for all $\delta \in (0, \delta_0)$, $\theta \in (0, \theta_0)$ there are $p_0 = p_0(\delta) = \frac{\delta}{2}$ and $\varepsilon_0 = \varepsilon_0(\delta, \rho)$ such that the estimates*

$$\|\varepsilon \boldsymbol{\xi}_{T_\varepsilon}\| = \varepsilon \|\mu_\varepsilon\| T_\varepsilon < \varepsilon^{2\rho} \quad \text{and} \quad \mathbb{P}\left([\varepsilon\boldsymbol{\xi}]_{T_\varepsilon}^d \ge \varepsilon^\rho\right) \le \exp(-\varepsilon^{-p})$$

*hold for all $p \in (0, p_0]$ and $\varepsilon \in (0, \varepsilon_0]$.*

See its proof in Appendix E.3.

**Lemma 2.** *Let $\delta \in (0,1)$ and $g_{t \geq 0}^t$ be a bounded adapted càdlàg stochastic process with values in $\mathbb{R}^d$, $T_\varepsilon = \varepsilon^{-\theta}$, $\theta > 0$. Suppose $\sup_{t \geq 0} \|g^t\|$ is well bounded. Assume $\rho_0 = \rho_0(\delta) = \frac{1-\delta}{16} > 0$, $\theta_0 = \theta_0(\delta) = \frac{1-\delta}{3} > 0$, $p_0 = \frac{\rho}{2}$. For $\widehat{\xi}_t$ in Eqn. (10), there is $\delta_0 = \delta_0(\delta) > 0$ such that for all $\rho \in (0, \rho_0)$ and $\theta \in (0, \theta_0)$, it holds*

$$\mathbb{P}\left( \sup_{0 \leq t \leq T_\varepsilon} \epsilon \left| \sum_{i=1}^{d} \int_0^t g_{s-}^i \, d\widehat{\xi}_s^i \right| \geq \varepsilon^\rho \right) \leq 2 \exp\left( -\varepsilon^{-p} \right)$$

*for all $p \in (0, p_0]$ and $0 < \varepsilon \leq \varepsilon_0$ with $\epsilon_0 = \varepsilon_0(\rho)$, where $\widehat{\xi}_s^i$ denotes the $i$-th entry in $\widehat{\xi}_s$.*

See its proof of Appendix E.4.

**Lemma 3.** *Suppose Assumptions 1 and 2 holds. Assume $\delta \in (0,1)$, $\rho_0 = \rho_0(\delta) = \frac{1-\delta}{16(1+c_1\kappa_1)} > 0$, $\theta_0 = \theta_0(\delta) = \frac{1-\delta}{3} > 0$, $p_0 = \min(\frac{\bar\rho(1+c_1\kappa_1)}{2}, p)$, $\frac{1}{c_2} \ln\left( \frac{2\Delta}{\mu \varepsilon^{\bar\rho}} \right) \leq \varepsilon^{-\theta_0}$ where $\kappa_1 = \ell$ and $c_2 = 2\mu$ in SGD, $\kappa_1 = \frac{c_2 \ell}{(v_{\min}+\epsilon)|\tau_m - 1|}$ and $c_3 = \frac{2\mu\tau}{\beta_1(v_{\max}+\epsilon)+\mu\tau}\left( \beta_1 - \frac{\beta_2}{4} \right)$ in ADAM. Here $c_1 \sim c_3$ are positive constants. For all $\bar\rho \in (0, \rho_0)$, $p \in (0, p_0]$, $0 < \varepsilon \leq \varepsilon_0$ with $\varepsilon_0 = \varepsilon_0(\bar\rho)$, and $\theta_0 = \widehat\theta_0$, we have*

$$\sup_{\theta_0 \in \Omega} \mathbb{P}\left( \sup_{0 \leq t < \sigma_1} \|\theta_t - \widehat\theta_t\| \geq 2\varepsilon^{\bar\rho} \right) \leq 2\exp(-\varepsilon^{-p/2}), \tag{11}$$

*where the sequences $\theta_t$ and $\widehat\theta_t$ are respectively produced by Eqn. (5) and (9) in Adam or Eqn. (4) and (8) in RMSPROP and SGD.*

See its proof in Appendix E.5.

## D Proof of Results in Sec. 4

### D.1 Proof of Theorem 1

*Proof.* Here we first briefly introduce our proof idea. As we proved in Lemma 3, for any $\delta \in (0,1)$, there exist $\rho_0$, $p_0$ and $\varepsilon_0$ such that for all $\bar\rho \in (0, \rho_0)$, $p \in (0, p_0]$ and $0 < \varepsilon \leq \varepsilon_0$, we have

$$\sup_{\theta_0 \in \Omega} \mathbb{P}\left( \sup_{0 \leq t < \sigma_1} \|\theta_t - \widehat\theta_t\| \geq 2\varepsilon^{\bar\rho} \right) \leq 2\exp(-\varepsilon^{-p/2}), \tag{12}$$

where the sequences $\theta_t$ and $\widehat\theta_t$ share the same initialization $\theta_0 = \widehat\theta_0$. Such a result holds for both SGD and Adam. Besides, from Theorems 3 and 4, we know that the sequence $\{\widehat\theta_t\}$ produced by Eqn. (8) or (9) (namely, the dynamic systems of SGD and Adam) exponentially converges to the minimum $\theta^*$ of the current local basin $\Omega$. To escape the local basin $\Omega$, there are two possible choices, the small jumps in the process $\xi$ and the big jumps $J_k$ in the process $\zeta$. As the small jumps in the process $\xi$ is well bounded, it is not very likely that these small jumps can help escape the local basin $\Omega$ which is verified by Eqn. (12). We well prove this more rigorously latter. For the big jumps $J$, since the expectation jump time $\mathbb{E}(\sigma_1)$ is $1/\Theta(\varepsilon^{-\delta})$, such as $\mathbb{E}(\sigma_1) = \frac{2}{\alpha}\varepsilon^{\alpha\delta}$ in the $\alpha$-stable ($S\alpha S$) distribution, $\mathbb{E}(\sigma_1)$ is usually much larger than the necessary time $t = \mathcal{O}(\ln(1/\varepsilon))$ to achieve $\|\widehat\theta_t - \theta^*\| \leq \varepsilon^\delta$. This means that before the jump time $\sigma_1$ the sequence $\widehat\theta_t$ is very close to the optimum of $\Omega$ and thus $\theta_t$ is very close to the minimum $\theta^*$. In this way, the escaping time $\Gamma$ of the sequence $\{\theta_t\}$ most likely occurs at the time $\sigma_1$ if the big jump $\varepsilon J_1$ in the process $\zeta$ is large. If the jump $\varepsilon J_1$ is small and $\theta_{\sigma_1}$ does not escape $\Omega$, then $\theta_t$ will converge to the minimum $\theta^*$ exponentially and stay in the small neighborhood of $\theta^*$. Accordingly, before the second jump time $t_2 = t_1 + \sigma_2$, $\theta_{t_2}$ will jump. This process will continue during the time interval $[0, t]$. Since for each jump time $t_k-$, $\theta_{t_k-}$ is very close to the optimum $\theta^*$, the big jump size $\varepsilon Q_{t_k}^{-1} \Sigma_{t_k} J_{t_k} \approx \varepsilon Q_{\theta^*}^{-1} \Sigma_{\theta^*} J_{t_k}$. So we can use $\varepsilon Q_t^{-1} \Sigma_t J_{t_k} \approx \varepsilon Q_{\theta^*}^{-1} \Sigma_{\theta^*} J_{t_k} \notin \Omega$ to judge whether at time $t_k$, $\theta_{t_k}$ escapes the local basin $\Omega$. The events $\{\varepsilon J_1 \notin \mathcal{W}\} = \{\varepsilon Q_{\theta^*}^{-1} \Sigma_{\theta^*} J_{t_k} \notin \Omega\}, \cdots, \{\varepsilon J_{k-1} \notin \mathcal{W}\} = \{\varepsilon Q_{\theta^*}^{-1} \Sigma_{\theta^*} J_{t_{k-}} \notin \Omega\}$, $\{\varepsilon J_k \notin \mathcal{W}\} = \{\varepsilon Q_{\theta^*}^{-1} \Sigma_{\theta^*} J_{t_k} \notin \Omega\}$ are independent.

Now we prove the desired results from two aspects, namely establishing upper bound and lower bound of $\mathbb{E}\left[\exp\left(-um(\mathcal{W})\Theta(\varepsilon^{-1})\Gamma\right)\right]$ for any $u > -1$. Before that, we first establish basic inequalities for lower and upper bounds.

**Basic inequalities for lower and upper bounds.** Since $\sigma_1$ is exponentially distributed with the parameter $\Theta(\epsilon^{-\delta})$, we compute the Laplace transform of $m(\mathcal{W})\Theta(\varepsilon^{-1})\sigma_1$ as follows:

$$\mathbb{E}\left[e^{-um(\mathcal{W})\Theta(\varepsilon^{-1})\sigma_1}\right] = \mathbb{E}\left[\int_0^{+\infty} e^{-um(\mathcal{W})\Theta(\varepsilon^{-1})\sigma_1}\cdot\Theta(\varepsilon^{-\delta})e^{-\Theta(\varepsilon^{-\delta})\sigma_1}\mathsf{d}\sigma_1\right]$$

$$=\frac{\Theta(\varepsilon^{-\delta})}{\Theta(\varepsilon^{-\delta})+um(\mathcal{W})\Theta(\varepsilon^{-1})} = \frac{1}{1+ua_\varepsilon},$$

where $a_\varepsilon = m(\mathcal{W})\frac{\Theta(\varepsilon^{-1})}{\Theta(-\varepsilon^\delta)}$ and $\Theta(\varepsilon^{-\delta}) = \Theta(-\varepsilon^\delta)$. Besides, for the probability law of the big jump we have

$$\mathbb{P}\left(Q_{\theta^*}^{-1}\Sigma_{\theta^*}\varepsilon J_1 \notin \Omega\right) = \mathbb{P}\left(\varepsilon J_1 \in \mathcal{W}\right) = \frac{\nu(\mathcal{W}/\varepsilon)}{\Theta(\varepsilon^{-\delta})}.$$

Since for the Lévy measure, we have $m(\mathcal{W}) = \lim_{u\to+\infty}\frac{\nu(u\mathcal{W})}{\Theta(u)}$ according to [11]. So for any $\delta'$, there always exists $\varepsilon$ such that it holds

$$a_\varepsilon(1-\delta') \leq \frac{\nu(\mathcal{W}/\varepsilon)}{\Theta(\varepsilon^{-\delta})} = \frac{\nu(\mathcal{W}/\varepsilon)}{\Theta(\varepsilon^{-1})}\frac{\Theta(\varepsilon^{-1})}{\Theta(\varepsilon^{-\delta})} \overset{\textcircled{1}}{\approx} m(\mathcal{W})\frac{\Theta(\varepsilon^{-1})}{\Theta(\varepsilon^{-\delta})} = m(\mathcal{W})\frac{\Theta(\varepsilon^{-1})}{\Theta(\varepsilon^{-\delta})} \leq a_\varepsilon(1+\delta'). \tag{13}$$

where $\textcircled{1}$ holds since $\varepsilon$ is enough small. Then with the help of the continuity of the function $(\theta, z) \to Q_\theta^{-1}\Sigma_\theta z$ both in $\theta$ and $z$. Indeed, for any $\delta'$ we can choose $R > 0$ enough large such that for small $\varepsilon$ we have

$$\mathbb{P}\left(\|\varepsilon J_1\| > R\right) \leq \frac{\delta'}{4}\frac{\Theta(\varepsilon^{-1})}{\Theta(\varepsilon^{-\delta})}.$$

Further, the function $(\theta, z) \to Q_\theta^{-1}\Sigma_\theta z$ is uniformly continuous in $z$ in the ball $\|z\| \leq R$ and is continuous in $\theta$ at the optimum $\theta^*$. Following [11], by using the scaling property of the jump measure $\nu$ and the fact that the limiting measure $m$ has no atoms we show that uniformly over $\|\theta - \theta^*\| \leq \varepsilon^\gamma$:

$$\begin{cases} \left|\mathbb{P}\left(Q_\theta^{-1}\Sigma_\theta\varepsilon J_k \notin \Omega^{\pm\varepsilon^\gamma},\ \|\varepsilon J_k\| \leq R\right) - \mathbb{P}\left(Q_{\theta^*}^{-1}\Sigma_{\theta^*}\varepsilon J_k \notin \Omega,\ \|\varepsilon J_k\| \leq R\right)\right| \leq \frac{\delta'}{4}\frac{\Theta(\varepsilon^{-1})}{\Theta(\varepsilon^{-\delta})}, \\ \left|\mathbb{P}\left(Q_\theta^{-1}\Sigma_\theta\varepsilon J_k \notin \Omega,\ \|\varepsilon J_k\| \leq R\right) - \mathbb{P}\left(Q_{\theta^*}^{-1}\Sigma_{\theta^*}\varepsilon J_k \notin \Omega,\ \|\varepsilon J_k\| \leq R\right)\right| \leq \frac{\delta'}{4}\frac{\Theta(\varepsilon^{-1})}{\Theta(\varepsilon^{-\delta})}, \end{cases} \tag{14}$$

At the same time, we also can establish

$$\mathbb{P}\left(Q_{\theta^*}^{-1}\Sigma_{\theta^*}\varepsilon J_k \notin \Omega\right) - \mathbb{P}\left(Q_{\theta^*}^{-1}\Sigma_{\theta^*}\varepsilon J_k \notin \Omega,\ \|\varepsilon J_k\| \leq R\right)$$

$$=\mathbb{P}\left(Q_{\theta^*}^{-1}\Sigma_{\theta^*}\varepsilon J_k \notin \Omega\right) - \mathbb{P}\left(\|\varepsilon J_k\| \leq R \mid Q_{\theta^*}^{-1}\Sigma_{\theta^*}\varepsilon J_k \notin \Omega\right)\mathbb{P}\left(Q_{\theta^*}^{-1}\Sigma_{\theta^*}\varepsilon J_k \notin \Omega\right)$$

$$=\mathbb{P}\left(Q_{\theta^*}^{-1}\Sigma_{\theta^*}\varepsilon J_k \notin \Omega\right)\left(1 - \mathbb{P}\left(\|\varepsilon J_k\| \leq R \mid Q_{\theta^*}^{-1}\Sigma_{\theta^*}\varepsilon J_k \notin \Omega\right)\right)$$

$$=\mathbb{P}\left(Q_{\theta^*}^{-1}\Sigma_{\theta^*}\varepsilon J_k \notin \Omega\right)\mathbb{P}\left(\|\varepsilon J_k\| > R \mid Q_{\theta^*}^{-1}\Sigma_{\theta^*}\varepsilon J_k \notin \Omega\right) \leq \mathbb{P}\left(\|\varepsilon J_k\| > R\right) \leq \frac{\delta'}{4}\frac{\Theta(\varepsilon^{-1})}{\Theta(\varepsilon^{-\delta})}. \tag{15}$$

**Upper bound of** $\mathbb{E}\left[\exp\left(-um(\mathcal{W})\Theta(\varepsilon^{-1})\Gamma\right)\right]$**.** In this part, we consider both the big jumps in the process $\zeta$ and the small jumps in the process $\xi$ which may escape the local minimum $\theta^*$. Instead of estimate the escaping time $\Gamma$ from $\Omega$, we first estimate the escaping time $\widetilde{\Xi}$ from $\Omega^{-\bar{\rho}}$. Here we define the inner part of $\Omega$ as $\Omega^{-\bar{\rho}} = \{y \in \Omega \mid \mathsf{dis}(\partial\Omega, y) \geq \rho\}$ and the outer $\rho$-neighborhood of $\Omega$ as $\Omega^{+\bar{\rho}} = \{y \mid \mathsf{dis}(\partial\Omega, y) \geq \bar{\rho}\}$. Then by setting $\bar{\rho} \downarrow 0$, we can use $\widetilde{\Xi}$ to estimate $\Gamma$ well. Let $\bar{\rho} = \varepsilon^\gamma$ where $\gamma$ is a constant such that the results of Lemmas 1$\sim$3 holds. Here we suppose the initial point $\theta_0 \in \Omega^{-2\varepsilon^\gamma}$.

**Step 1.** In this step we give the formulation of the upper bound of $\mathbb{E}\left[e^{-um(\mathcal{W})\Theta(\varepsilon^{-1})\Gamma}\right]$. For any $u > -1$, we can compute the formula of the total probability as follows

$$\mathbb{E}\left[e^{-um(\mathcal{W})\Theta(\varepsilon^{-1})\widetilde{\Xi}}\right] \leq \sum_{k=1}^{+\infty}\mathbb{E}\left[e^{-um(\mathcal{W})\Theta(\varepsilon^{-1})t_k}\mathbb{I}\left\{\widetilde{\Xi} = t_k\right\} + \mathrm{Res}_k\right],$$

where

$$
\mathrm{Res}_k \leq \begin{cases} \mathbb{E}\left[e^{-um(\boldsymbol{\mathcal{W}})\Theta(\varepsilon^{-1})t_k}\mathbb{I}\left\{\widetilde{\Xi}\in(t_{k-1},t_k)\right\}\right], & \text{if } u\in(-1,0] \\ \mathbb{E}\left[e^{-um(\boldsymbol{\mathcal{W}})\Theta(\varepsilon^{-1})t_{k-1}}\mathbb{I}\left\{\widetilde{\Xi}\in(t_{k-1},t_k)\right\}\right], & \text{if } u\in(0,+\infty). \end{cases}
$$

**Step 2.** In this step we specifically upper bounds the first term $\sum_{k=1}^{+\infty}\mathbb{E}\left[e^{-um(\boldsymbol{\mathcal{W}})\Theta(\varepsilon^{-1})t_k}\mathbb{I}\left\{\widetilde{\Xi}=t_k\right\}\right]$. For $k\geq 1$, we can use the strong Markov property and obtain

$$
\mathbb{E}\left[e^{-um(\boldsymbol{\mathcal{W}})\Theta(\varepsilon^{-1})t_k}\mathbb{I}\left\{\widetilde{\Xi}=t_k\right\}\right]=\mathbb{E}\left[e^{-um(\boldsymbol{\mathcal{W}})\Theta(\varepsilon^{-1})t_k}\mathbb{I}\left\{\boldsymbol{\theta}_t\in\boldsymbol{\Omega}^{-\varepsilon^\gamma},t\in[0,t_k),\boldsymbol{\theta}_{t_k}\notin\boldsymbol{\Omega}^{-\varepsilon^\gamma}\right\}\right]
$$

$$
=\mathbb{E}\left[e^{-um(\boldsymbol{\mathcal{W}})\Theta(\varepsilon^{-1})\sigma_k}\mathbb{I}\left\{\boldsymbol{\theta}_{t+t_{k-1}}\in\boldsymbol{\Omega}^{-\varepsilon^\gamma},t\in[0,\sigma_k)\right\}\mathbb{I}\left\{\boldsymbol{\theta}_{t_k}\notin\boldsymbol{\Omega}^{-\varepsilon^\gamma}\right\}\right.
$$

$$
\left.\cdot\prod_{i=1}^{k-1}e^{-um(\boldsymbol{\mathcal{W}})\Theta(\varepsilon^{-1})\sigma_i}\mathbb{I}\left\{\boldsymbol{\theta}_{t+t_{i-1}}\in\boldsymbol{\Omega}^{-\varepsilon^\gamma},t\in[0,\sigma_k]\right\}\right]
$$

$$
\leq\sup_{\boldsymbol{\theta}_0\in\boldsymbol{\Omega}^{-2\varepsilon^\gamma}}\mathbb{E}\left[e^{-um(\boldsymbol{\mathcal{W}})\Theta(\varepsilon^{-1})\sigma_1}\mathbb{I}\left\{\boldsymbol{\theta}_t\in\boldsymbol{\Omega}^{-\varepsilon^\gamma},t\in[0,\sigma_1)\right\}\mathbb{I}\left\{\boldsymbol{\theta}_{\sigma_1}\notin\boldsymbol{\Omega}^{-\varepsilon^\gamma}\right\}\right]
$$

$$
\cdot\sup_{\boldsymbol{\theta}_0\in\boldsymbol{\Omega}^{-2\varepsilon^\gamma}}\mathbb{E}\left[e^{-um(\boldsymbol{\mathcal{W}})\Theta(\varepsilon^{-1})\sigma_1}\mathbb{I}\left\{\boldsymbol{\theta}_t\in\boldsymbol{\Omega}^{-\varepsilon^\gamma},t\in[0,\sigma_1]\right\}\right]^{k-1}.
$$

Recall $\bar{\rho}=\varepsilon^\gamma$ where $\gamma$ is a constant such that the results of Lemmas 1$\sim$3 holds. The escaping from the basin $\boldsymbol{\Omega}^{-\varepsilon^\gamma}$ with a big jump $\varepsilon J_1$ occurs when $\boldsymbol{Q}_{\sigma_1-}^{-1}\boldsymbol{\Sigma}_{\sigma_1-}\varepsilon J_1\in\boldsymbol{\Omega}^{-\varepsilon^\gamma}$. Furthermore, $\sup_{0\leq t<\sigma_1}\|\boldsymbol{\theta}_t-\widehat{\boldsymbol{\theta}}_t\|\leq\frac{1}{2}\varepsilon^\gamma$ with probability exponentially close to 1 (verified by Lemma 3). Meanwhile $\sigma_1=\frac{2}{\alpha}\varepsilon^{\alpha\delta}$ in the $\alpha$-stable ($\mathcal{S}\alpha\mathcal{S}$) distribution is much larger than $v_\varepsilon=\mathcal{O}(\ln(1/\varepsilon))$ with sufficient small $\varepsilon$, $\widehat{\boldsymbol{\theta}}_t$ reaches a $\frac{1}{2}\varepsilon^\gamma$-neighborhood of the optimum $\boldsymbol{\theta}^*$ which only requires time $v_\varepsilon$. So this actually means $\sup_{0\leq t<\sigma_1}\|\boldsymbol{\theta}_t-\boldsymbol{\theta}^*\|\leq\varepsilon^\gamma$. In this way, to obtain the final upper bound results, we only need to estimate the escaping probability $\mathbb{P}\left(\boldsymbol{Q}_{\boldsymbol{\theta}}^{-1}\boldsymbol{\Sigma}_{\boldsymbol{\theta}}\varepsilon J_1\in\boldsymbol{\Omega}^{-\varepsilon^\gamma}\right)$ and $\mathbb{P}\left(\boldsymbol{Q}_{\boldsymbol{\theta}}^{-1}\boldsymbol{\Sigma}_{\boldsymbol{\theta}}\varepsilon J_1\notin\boldsymbol{\Omega}^{-\varepsilon^\gamma}\right)$ uniformly over $\|\boldsymbol{\theta}-\boldsymbol{\theta}^*\|\leq\varepsilon^\gamma$. Then we first give two important inequalities which will used to bound each component later:

$$
\sup_{\|\boldsymbol{\theta}-\boldsymbol{\theta}^*\|\leq\varepsilon^\gamma}\mathbb{P}\left(\boldsymbol{Q}_{\boldsymbol{\theta}}^{-1}\boldsymbol{\Sigma}_{\boldsymbol{\theta}}\varepsilon J_k\notin\boldsymbol{\Omega}^{-\varepsilon^\gamma}\right)
$$

$$
=\sup_{\|\boldsymbol{\theta}-\boldsymbol{\theta}^*\|\leq\varepsilon^\gamma}\mathbb{P}\left(\boldsymbol{Q}_{\boldsymbol{\theta}}^{-1}\boldsymbol{\Sigma}_{\boldsymbol{\theta}}\varepsilon J_k\notin\boldsymbol{\Omega}^{-\varepsilon^\gamma},\|\varepsilon J_k\|\leq R\right)+\mathbb{P}\left(\boldsymbol{Q}_{\boldsymbol{\theta}}^{-1}\boldsymbol{\Sigma}_{\boldsymbol{\theta}}\varepsilon J_k\notin\boldsymbol{\Omega}^{-\varepsilon^\gamma},\|\varepsilon J_k\|>R\right)
$$

$$
\overset{\text{①}}{\geq}\mathbb{P}\left(\boldsymbol{Q}_{\boldsymbol{\theta}^*}^{-1}\boldsymbol{\Sigma}_{\boldsymbol{\theta}^*}\varepsilon J_k\notin\boldsymbol{\Omega},\|\varepsilon J_k\|\leq R\right)-\frac{\delta'}{4}\frac{\Theta(\varepsilon^{-1})}{\Theta(\varepsilon^{-\delta})}+\mathbb{P}\left(\boldsymbol{Q}_{\boldsymbol{\theta}}^{-1}\boldsymbol{\Sigma}_{\boldsymbol{\theta}}\varepsilon J_k\notin\boldsymbol{\Omega}^{-\varepsilon^\gamma},\|\varepsilon J_k\|>R\right)
$$

$$
\geq\mathbb{P}\left(\boldsymbol{Q}_{\boldsymbol{\theta}^*}^{-1}\boldsymbol{\Sigma}_{\boldsymbol{\theta}^*}\varepsilon J_k\notin\boldsymbol{\Omega},\|\varepsilon J_k\|\leq R\right)-\frac{\delta'}{4}\frac{\Theta(\varepsilon^{-1})}{\Theta(\varepsilon^{-\delta})}
$$

$$
\overset{\text{②}}{\geq}\mathbb{P}\left(\boldsymbol{Q}_{\boldsymbol{\theta}^*}^{-1}\boldsymbol{\Sigma}_{\boldsymbol{\theta}^*}\varepsilon J_k\notin\boldsymbol{\Omega}\right)-\frac{\delta'}{2}\frac{\Theta(\varepsilon^{-1})}{\Theta(\varepsilon^{-\delta})}
$$

$$
\overset{\text{③}}{\geq}m(\boldsymbol{\mathcal{W}})\left(1-\delta'-\frac{\delta'}{2m(\boldsymbol{\mathcal{W}})}\right)\frac{\Theta(\varepsilon^{-1})}{\Theta(\varepsilon^{-\delta})}\overset{\text{④}}{\geq}m(\boldsymbol{\mathcal{W}})(1-\rho)\frac{\Theta(\varepsilon^{-1})}{\Theta(\varepsilon^{-\delta})},
$$

where ① uses the result in Eqn. (14), ② uses Eqn. (15), ③ uses Eqn. (13), and in ④ we set $\delta'$ enough small such that $\rho\geq\delta'+\frac{\delta'}{2m(\boldsymbol{\mathcal{W}})}$. So in this way, for any $\rho$ we choose $\delta'>0$ small enough to lower bound $\sup_{\|\boldsymbol{\theta}-\boldsymbol{\theta}^*\|\leq\varepsilon^{-\gamma}}\mathbb{P}\left(\boldsymbol{Q}_{\boldsymbol{\theta}}^{-1}\boldsymbol{\Sigma}_{\boldsymbol{\theta}}\varepsilon J_k\in\boldsymbol{\Omega}^{-\varepsilon^\gamma}\right)$ as follows:

$$
\sup_{\|\boldsymbol{\theta}-\boldsymbol{\theta}^*\|\leq\varepsilon^\gamma}\mathbb{P}\left(\boldsymbol{Q}_{\boldsymbol{\theta}}^{-1}\boldsymbol{\Sigma}_{\boldsymbol{\theta}}\varepsilon J_k\in\boldsymbol{\Omega}^{-\varepsilon^\gamma}\right)=1-\sup_{\|\boldsymbol{\theta}-\boldsymbol{\theta}^*\|\leq\varepsilon^\gamma}\mathbb{P}\left(\boldsymbol{Q}_{\boldsymbol{\theta}}^{-1}\boldsymbol{\Sigma}_{\boldsymbol{\theta}}\varepsilon J_k\notin\boldsymbol{\Omega}^{-\varepsilon^\gamma}\right)\geq 1-a_\varepsilon(1-\rho).
$$

Similarly, we only need to upper bound the remaining term $\sup_{\|\boldsymbol{\theta}-\boldsymbol{\theta}^*\|\leq\varepsilon^\gamma}\mathbb{P}\left(\boldsymbol{Q}_{\boldsymbol{\theta}}^{-1}\boldsymbol{\Sigma}_{\boldsymbol{\theta}}\varepsilon J_k\notin\boldsymbol{\Omega}\right)$ as follows:

$$\sup_{\|\boldsymbol{\theta}-\boldsymbol{\theta}^*\|\leq\varepsilon^\gamma}\mathbb{P}\left(\boldsymbol{Q}_{\boldsymbol{\theta}}^{-1}\boldsymbol{\Sigma}_{\boldsymbol{\theta}}\varepsilon J_k\notin\boldsymbol{\Omega}^{-\varepsilon^\gamma}\right)$$

$$=\sup_{\|\boldsymbol{\theta}-\boldsymbol{\theta}^*\|\leq\varepsilon^\gamma}\mathbb{P}\left(\boldsymbol{Q}_{\boldsymbol{\theta}}^{-1}\boldsymbol{\Sigma}_{\boldsymbol{\theta}}\varepsilon J_k\notin\boldsymbol{\Omega}^{-\varepsilon^\gamma},\|\varepsilon J_k\|\leq R\right)+\mathbb{P}\left(\boldsymbol{Q}_{\boldsymbol{\theta}}^{-1}\boldsymbol{\Sigma}_{\boldsymbol{\theta}}\varepsilon J_k\notin\boldsymbol{\Omega}^{-\varepsilon^\gamma},\|\varepsilon J_k\|>R\right)$$

$$\overset{①}{\leq}\sup_{\|\boldsymbol{\theta}-\boldsymbol{\theta}^*\|\leq\varepsilon^\gamma}\mathbb{P}\left(\boldsymbol{Q}_{\boldsymbol{\theta}}^{-1}\boldsymbol{\Sigma}_{\boldsymbol{\theta}}\varepsilon J_k\notin\boldsymbol{\Omega}^{-\varepsilon^\gamma},\|\varepsilon J_k\|\leq R\right)+\frac{\delta'}{4}\frac{\Theta(\varepsilon^{-1})}{\Theta(\varepsilon^{-\delta})}$$

$$\overset{②}{\leq}\sup_{\|\boldsymbol{\theta}-\boldsymbol{\theta}^*\|\leq\varepsilon^\gamma}\mathbb{P}\left(\boldsymbol{Q}_{\boldsymbol{\theta}^*}^{-1}\boldsymbol{\Sigma}_{\boldsymbol{\theta}^*}\varepsilon J_k\notin\boldsymbol{\Omega},\ \|\varepsilon J_k\|\leq R\right)+\frac{\delta'}{2}\frac{\Theta(\varepsilon^{-1})}{\Theta(\varepsilon^{-\delta})}$$

$$\leq\sup_{\|\boldsymbol{\theta}-\boldsymbol{\theta}^*\|\leq\varepsilon^\gamma}\mathbb{P}\left(\boldsymbol{Q}_{\boldsymbol{\theta}^*}^{-1}\boldsymbol{\Sigma}_{\boldsymbol{\theta}^*}\varepsilon J_k\notin\boldsymbol{\Omega}\right)+\frac{\delta'}{2}\frac{\Theta(\varepsilon^{-1})}{\Theta(\varepsilon^{-\delta})}$$

$$\overset{③}{\leq}m(\boldsymbol{\mathcal{W}})\left(1+\delta'+\frac{\delta'}{2m(\boldsymbol{\mathcal{W}})}\right)\frac{\Theta(\varepsilon^{-1})}{\Theta(\varepsilon^{-\delta})}\leq m(\boldsymbol{\mathcal{W}})(1+\rho/3)\frac{\Theta(\varepsilon^{-1})}{\varepsilon^{-\delta}}=a_\varepsilon(1+\rho/3),$$

where ① uses $\mathbb{P}\left(\boldsymbol{Q}_{\boldsymbol{\theta}}^{-1}\boldsymbol{\Sigma}_{\boldsymbol{\theta}}\varepsilon J_k\notin\boldsymbol{\Omega}^{-\varepsilon^\gamma},\|\varepsilon J_k\|>R\right)\leq\mathbb{P}\left(\|\varepsilon J_k\|>R\right)\leq\frac{\delta'}{4}\frac{\Theta(\varepsilon^{-1})}{\Theta(\varepsilon^{-\delta})}$, ② uses the result in Eqn. (14), ② uses Eqn. (15), and ③ uses Eqn. (13).

Next, for any $\rho>0$ and $\varepsilon$ we can obtain the Laplace transforms for any $u>-1$ as follows:

$$\sup_{\boldsymbol{\theta}_0\in\boldsymbol{\Omega}^{-2\varepsilon^\gamma}}\mathbb{E}\left[e^{-um(\boldsymbol{\mathcal{W}})\Theta(\varepsilon^{-1})\sigma_1}\mathbb{I}\left\{\boldsymbol{\theta}_t\in\boldsymbol{\Omega}^{-\varepsilon^\gamma},\ t\in[0,\sigma_1]\right\}\right]$$

$$\leq[1-a_\varepsilon(1+\rho)]\mathbb{E}\left[\int_0^{+\infty}e^{-um(\boldsymbol{\mathcal{W}})\Theta(\varepsilon^{-1})\sigma_1}\cdot\Theta(\varepsilon^{-\delta})e^{-\Theta(\varepsilon^{-\delta})\sigma_1}\mathsf{d}\sigma_1\right] \qquad (16)$$

$$=\frac{1-a_\varepsilon(1-\rho)}{1+ua_\varepsilon}.$$

and

$$\sup_{\boldsymbol{\theta}_0\in\boldsymbol{\Omega}^{-2\varepsilon^\gamma}}\mathbb{E}\left[e^{-um(\boldsymbol{\mathcal{W}})\Theta(\varepsilon^{-1})\sigma_1}\mathbb{I}\left\{\boldsymbol{\theta}_t\in\boldsymbol{\Omega}^{-\varepsilon^\gamma},t\in[0,\sigma_1)\right\}\mathbb{I}\left\{\boldsymbol{\theta}_{\sigma_1}\notin\boldsymbol{\Omega}^{-\varepsilon^\gamma}\right\}\right]$$

$$\leq a_\varepsilon\left(1+\frac{\rho}{3}\right)\mathbb{E}\left[\int_0^{+\infty}e^{-um(\boldsymbol{\mathcal{W}})\Theta(\varepsilon^{-1})\sigma_1}\cdot\Theta(\varepsilon^{-\delta})e^{-\Theta(\varepsilon^{-\delta})\sigma_1}\mathsf{d}\sigma_1\right]=\frac{a_\varepsilon(1-\rho/3)}{1+ua_\varepsilon}.$$

Here we summarize the above results such that we can upper bound the first term $\sum_{k=1}^{+\infty}\mathbb{E}\left[e^{-um(\boldsymbol{\mathcal{W}})\Theta(\varepsilon^{-1})t_k}\mathbb{I}\{\Gamma=t_k\}\right]$:

$$\mathcal{R}_1=\sum_{k=1}^{+\infty}\mathbb{E}\left[e^{-um(\boldsymbol{\mathcal{W}})\Theta(\varepsilon^{-1})t_k}\mathbb{I}\{\Gamma=t_k\}\right]\leq\frac{a_\varepsilon(1+\rho/3)}{1+ua_\varepsilon}\sum_{k=1}^{+\infty}\left(\frac{1-a_\varepsilon(1-\rho)}{1+ua_\varepsilon}\right)^{k-1}$$

$$\leq\frac{a_\varepsilon(1+\rho/3)}{1+ua_\varepsilon}\sum_{k=0}^{+\infty}\left(\frac{1-a_\varepsilon(1-\rho)}{1+ua_\varepsilon}\right)^{k-1}=\frac{1+\rho/3}{1+u-\rho}.$$

**Step 3.** In this step we specifically upper bounds the second term $\sum_{k=1}^{+\infty}\mathbb{E}\left[\text{Res}_k\right]$. Specifically, we establish upper bound for each $\mathbb{E}\left[\text{Res}_k\right]$ as follows. We first consider the case where $k=1$:

$$\text{Res}_1\leq\begin{cases}\mathbb{E}\left[e^{-um(\boldsymbol{\mathcal{W}})\Theta(\varepsilon^{-1}))t_1}\mathbb{I}\{\Gamma\in(0,t_1)\}\right], & \text{if }u\in(-1,0]\\ \mathbb{E}\left[\mathbb{I}\{\Gamma\in(0,t_1)\}\right], & \text{if }u\in(0,+\infty).\end{cases}$$

$$=\begin{cases}\mathbb{E}\left[e^{-um(\boldsymbol{\mathcal{W}})\Theta(\varepsilon^{-1}))\sigma_1}\mathbb{I}\{\exists t\in(0,\sigma_1):\ \boldsymbol{\theta}_t\notin\boldsymbol{\Omega}^{-\varepsilon^\gamma}\}\right], & \text{if }u\in(-1,0]\\ \mathbb{E}\left[\mathbb{I}\{\exists t\in(0,\sigma_1):\ \boldsymbol{\theta}_t\notin\boldsymbol{\Omega}^{-\varepsilon^\gamma}\}\right], & \text{if }u\in(0,+\infty).\end{cases}$$

$$\leq\begin{cases}\mathbb{E}\left[e^{-um(\boldsymbol{\mathcal{W}})\Theta(\varepsilon^{-1}))\sigma_1}\sup_{\boldsymbol{\theta}_0\in\boldsymbol{\Omega}^{-2\varepsilon^\gamma}}\mathbb{I}\{\exists t\in(0,\sigma_1):\ \boldsymbol{\theta}_t\notin\boldsymbol{\Omega}^{-\varepsilon^\gamma}\}\right], & \text{if }u\in(-1,0]\\ \mathbb{E}\left[\sup_{\boldsymbol{\theta}_0\in\boldsymbol{\Omega}^{-2\varepsilon^\gamma}}\mathbb{I}\{\exists t\in(0,\sigma_1):\ \boldsymbol{\theta}_t\notin\boldsymbol{\Omega}^{-\varepsilon^\gamma}\}\right], & \text{if }u\in(0,+\infty).\end{cases}$$

For $k \geq 2$, it needs more efforts to be upper bounded:

$\text{Res}_k$

$$\leq \begin{cases} \mathbb{E}\left[e^{-um(\boldsymbol{W})\Theta(\varepsilon^{-1}))t_k}\mathbb{I}\left\{\Gamma \in (t_{k-1}, t_k)\right\}\right], & \text{if } u \in (-1, 0] \\ \mathbb{E}\left[e^{-um(\boldsymbol{W})\Theta(\varepsilon^{-1}))t_{k-1}}\mathbb{I}\left\{\Gamma \in (t_{k-1}, t_k)\right\}\right], & \text{if } u \in (0, +\infty). \end{cases}$$

$$= \begin{cases} \mathbb{E}\left[e^{-um(\boldsymbol{W})\Theta(\varepsilon^{-1}))t_k}\mathbb{I}\left\{t \in [0, t_{k-1}]: \boldsymbol{\theta}_t \in \boldsymbol{\Omega}^{-\varepsilon^{\gamma}}\right\}\mathbb{I}\left\{\exists t \in (t_{k-1}, t_k): \boldsymbol{\theta}_t \notin \boldsymbol{\Omega}^{-\varepsilon^{\gamma}}\right\}\right], & \text{if } u \in (-1, 0] \\ \mathbb{E}\left[e^{-um(\boldsymbol{W})\Theta(\varepsilon^{-1}))t_{k-1}}\mathbb{I}\left\{t \in [0, t_{k-1}]: \boldsymbol{\theta}_t \in \boldsymbol{\Omega}^{-\varepsilon^{\gamma}}\right\}\mathbb{I}\left\{\exists t \in (t_{k-1}, t_k): \boldsymbol{\theta}_t \notin \boldsymbol{\Omega}^{-\varepsilon^{\gamma}}\right\}\right], & \text{if } u \in (0, +\infty). \end{cases}$$

In this case, for all $u > 0$ we can upper bound $\text{Res}_k$ as

$$\text{Res}_k \leq \left[\mathbb{E}\left[x \sup_{\boldsymbol{\theta}_0 \in \boldsymbol{\Omega}^{-2\varepsilon^{\gamma}}} \mathbb{I}\left\{t \in [0, \sigma_1]: \boldsymbol{\theta}_t \in \boldsymbol{\Omega}^{-\varepsilon^{\gamma}}\right\}\right]\right]^{k-2}$$

$$\mathbb{E}\left[x \sup_{\boldsymbol{\theta}_0 \in \boldsymbol{\Omega}^{-2\varepsilon^{\gamma}}} \mathbb{I}\left\{t \in [0, \sigma_1]: \boldsymbol{\theta}_t \in \boldsymbol{\Omega}^{-\varepsilon^{\gamma}}\right\}\mathbb{I}\left\{\exists t \in (0, \sigma_1): \boldsymbol{\theta}_t \notin \boldsymbol{\Omega}^{-\varepsilon^{\gamma}}\right\}\right].$$

where $x = e^{-um(\boldsymbol{W})\Theta(\varepsilon^{-1}))\sigma_1}$. Let the event $E = \{\sup_{0 \leq t < \sigma_1} \|\boldsymbol{\theta}_t - \widehat{\boldsymbol{\theta}}_t\| \leq \varepsilon^{\gamma}\}$. Now we bound each term in the above inequalities:

$$\mathbb{E}\left[e^{-um(\boldsymbol{W})\Theta(\varepsilon^{-1}))\sigma_1} \sup_{\boldsymbol{\theta}_0 \in \boldsymbol{\Omega}^{-2\varepsilon^{\gamma}}} \mathbb{I}\left\{\exists t \in (0, \sigma_1): \boldsymbol{\theta}_t \notin \boldsymbol{\Omega}^{-\varepsilon^{\gamma}}\right\}\right]$$

$$= \mathbb{E}\left[e^{-um(\boldsymbol{W})\Theta(\varepsilon^{-1}))\sigma_1} \sup_{\boldsymbol{\theta}_0 \in \boldsymbol{\Omega}^{-2\varepsilon^{\gamma}}} \mathbb{I}\left\{\exists t \in (0, \sigma_1): \boldsymbol{\theta}_t \notin \boldsymbol{\Omega}^{-\varepsilon^{\gamma}}\right\}(\mathbb{I}\{E\} + \mathbb{I}\{E^c\})\right]$$

$$\leq \mathbb{E}\left[e^{-um(\boldsymbol{W})\Theta(\varepsilon^{-1}))\sigma_1} \sup_{\boldsymbol{\theta}_0 \in \boldsymbol{\Omega}^{-2\varepsilon^{\gamma}}} \mathbb{I}\left\{\exists t \in (0, \sigma_1): \boldsymbol{\theta}_t \notin \boldsymbol{\Omega}^{-\varepsilon^{\gamma}}\right\}\mathbb{I}\{E^c\}\right] \qquad (17)$$

$$\overset{①}{\leq} \mathbb{E}\left[e^{-um(\boldsymbol{W})\Theta(\varepsilon^{-1}))\sigma_1} \exp(-\varepsilon^{-p})\right] = \frac{\Theta(\varepsilon^{-\delta})}{\Theta(\varepsilon^{-\delta}) + um(\boldsymbol{W})\Theta(\varepsilon^{-1})} \cdot 2\exp(-\varepsilon^{-p})$$

$$= \frac{1}{1 + ua_{\varepsilon}}\exp(-\varepsilon^{-p}) \overset{②}{\leq} \frac{\rho/3}{1 + u - \rho},$$

where ① uses the fact that $\sup_{\boldsymbol{\theta}_0 \in \boldsymbol{\Omega}^{-2\varepsilon^{\gamma}}} \mathbb{I}\{\exists t \in (0, \sigma_1): \boldsymbol{\theta}_t \notin \boldsymbol{\Omega}\} \leq 1$ and the sequence $\widehat{\boldsymbol{\theta}}_t$ obeys $\boldsymbol{\Omega}^{-2\varepsilon^{\gamma}}$ due to $\boldsymbol{\theta}_0 \in \boldsymbol{\Omega}^{-2\varepsilon^{\gamma}}$ and the results in Lemma 3:

$$\sup_{\boldsymbol{\theta}_0 \in \boldsymbol{\Omega}} \mathbb{P}\left(\sup_{0 \leq t < \sigma_1} \|\boldsymbol{\theta}_t - \widehat{\boldsymbol{\theta}}_t\| \geq \varepsilon^{\gamma}\right) \leq 2\exp(-\varepsilon^{-p}),$$

where the sequences $\boldsymbol{\theta}_t$ and $\widehat{\boldsymbol{\theta}}_t$ share the same initialization $\boldsymbol{\theta}_0 = \widehat{\boldsymbol{\theta}}_0$. In ② we set $\varepsilon$ small enough such that $2\exp(-\varepsilon^{-p}) \leq \frac{\rho/3}{1+u-\rho}$. Similarly, we can upper bound

$$\mathbb{E}\left[\sup_{\boldsymbol{\theta}_0 \in \boldsymbol{\Omega}^{-2\varepsilon^{\gamma}}} \mathbb{I}\left\{\exists t \in (0, t_1): \boldsymbol{\theta}_t \notin \boldsymbol{\Omega}^{-\varepsilon^{\gamma}}\right\}\right] \leq \mathbb{E}\left[\sup_{\boldsymbol{\theta}_0 \in \boldsymbol{\Omega}^{-2\varepsilon^{\gamma}}} \mathbb{I}\left\{\exists t \in (0, t_1): \boldsymbol{\theta}_t \notin \boldsymbol{\Omega}^{-\varepsilon^{\gamma}}\right\}(\mathbb{I}\{E\} + \mathbb{I}\{E^c\})\right]$$

$$\leq \exp(-\varepsilon^{-p}) \leq \frac{\rho/3}{1 + u - \rho}. \qquad (18)$$

Since $p$ is much smaller than 1, then we have for $k = 2, \cdots, k$

$$\text{Res}_k \leq \left[\mathbb{E}\left[e^{-um(\boldsymbol{W})\Theta(\epsilon^{-1}))\sigma_1} \sup_{\boldsymbol{\theta}_0 \in \boldsymbol{\Omega}^{-2\varepsilon^{\gamma}}} \mathbb{I}\left\{t \in [0, \sigma_1]: \boldsymbol{\theta}_t \in \boldsymbol{\Omega}^{-\varepsilon^{\gamma}}\right\}\right]\right]^{k-2} \mathbb{E}\left[e^{-um(\boldsymbol{W})\Theta(\epsilon^{-1}))\sigma_1}\right.$$

$$\sup_{\boldsymbol{\theta}_0 \in \boldsymbol{\Omega}^{-2\varepsilon^{\gamma}}} \mathbb{I}\left\{t \in [0, \sigma_1]: \boldsymbol{\theta}_t \in \boldsymbol{\Omega}^{-\varepsilon^{\gamma}}\right\}\mathbb{I}\left\{\exists t \in (0, \sigma_1): \boldsymbol{\theta}_t \notin \boldsymbol{\Omega}^{-\varepsilon^{\gamma}}\right\}\right] \leq \left[\frac{1 - a_{\varepsilon}(1 - \rho)}{1 + ua_{\varepsilon}}\right]^{k-2} \frac{a_{\varepsilon}(1 + \rho/3)}{1 + ua_{\varepsilon}}.$$

where we use the above results, namely, $\sup_{\boldsymbol{\theta}_0 \in \boldsymbol{\Omega}^{-2\varepsilon^\gamma}} \mathbb{E}\left[e^{-um(\boldsymbol{W})\Theta(\varepsilon^{-1})\sigma_1}\mathbb{I}\left\{\boldsymbol{\theta}_t \in \boldsymbol{\Omega}^{-\varepsilon^\gamma}, \ t \in [0, \sigma_1]\right\}\right] \leq$ $\frac{1-a_\varepsilon(1-\rho)}{1+ua_\varepsilon}$ and $\sup_{\boldsymbol{\theta}_0 \in \boldsymbol{\Omega}^{-2\varepsilon^\gamma}} \mathbb{E}\left[e^{-um(\boldsymbol{W})\Theta(\varepsilon^{-1})\sigma_1}\mathbb{I}\left\{\boldsymbol{\theta}_t \in \boldsymbol{\Omega}^{-\varepsilon^\gamma}, \ t \in [0, \sigma_1]\right\}\mathbb{I}\left\{\boldsymbol{\theta}_{\sigma_1} \notin \boldsymbol{\Omega}^{-\varepsilon^\gamma}\right\}\right] \leq$ $\frac{a_\varepsilon(1+\rho/3)}{1+ua_\varepsilon}$. So in this case, we have

$$\mathcal{R}_2 = \sum_{k=1}^{+\infty} \mathbb{E}\left[\text{Res}_k\right] \leq \frac{\rho/3}{1+u-\rho} + \sum_{k=2}^{+\infty}\left[\frac{1-a_\varepsilon(1-\rho)}{1+ua_\varepsilon}\right]^{k-2}\frac{a_\varepsilon(1+\rho/3)}{1+ua_\varepsilon} = \frac{1+2\rho/3}{1+u-\rho}$$

Therefore, for any $\boldsymbol{\theta}_0 \in \boldsymbol{\Omega}^{-2\varepsilon^\gamma}$ we can upper bound

$$\mathbb{E}\left[e^{-um(\boldsymbol{W})\Theta(\varepsilon^{-1}))\Gamma}\right] \leq \mathcal{R}_1 + \mathcal{R}_2 \leq \frac{1+\rho}{1+u-\rho},$$

where $\rho \downarrow 0$ as $\varepsilon \downarrow 0$.

**Lower bound of** $\mathbb{E}\left[\exp\left(-um(\boldsymbol{W})\Theta(\varepsilon^{-1})\Gamma\right)\right]$**.** In this part, we only consider the big jumps in the process $\boldsymbol{\zeta}$ which may escape the local minimum $\boldsymbol{\theta}^*$, and ignore the possibility of the small jumps in the process $\boldsymbol{\xi}$ which may also help escape local minimum $\boldsymbol{\theta}^*$. Here we consider the result under $\boldsymbol{\theta}_0 \in \boldsymbol{\Omega}^{-\varepsilon^\gamma}$ which is stronger than the results under $\boldsymbol{\theta}_0 \in \boldsymbol{\Omega}^{-2\varepsilon^\gamma}$ due to $\boldsymbol{\Omega}^{-2\varepsilon^\gamma} \subset \boldsymbol{\Omega}^{-\varepsilon^\gamma}$.

**Step 1.** In this step we give the formulation of the lower bound of $\mathbb{E}\left[e^{-um(\boldsymbol{W})\Theta(\varepsilon^{-1})\Gamma}\right]$. For any $u > -1$, we can compute the formula of the total probability as follows

$$\mathbb{E}\left[e^{-um(\boldsymbol{W})\Theta(\varepsilon^{-1})\Gamma}\right] \geq \sum_{k=1}^{+\infty}\mathbb{E}\left[e^{-um(\boldsymbol{W})\Theta(\varepsilon^{-1})t_k}\mathbb{I}\left\{\Gamma = t_k\right\}\right].$$

This inequality holds, since we ignore the small jumps in the process $\boldsymbol{\xi}$ which may also help escape local minimum $\boldsymbol{\theta}^*$.

For any small $\bar{\rho} > 0$, we define the inner part of $\boldsymbol{\Omega}$ as $\boldsymbol{\Omega}^{-\bar{\rho}} = \{\boldsymbol{y} \in \boldsymbol{\Omega} \mid \text{dis}(\partial\boldsymbol{\Omega}, \boldsymbol{y}) \geq \rho\}$ and the outer $\rho$-neighborhood of $\boldsymbol{\Omega}$ as $\boldsymbol{\Omega}^{+\bar{\rho}} = \{\boldsymbol{y} \mid \text{dis}(\partial\boldsymbol{\Omega}, \boldsymbol{y}) \geq \bar{\rho}\}$. For $k \geq 1$, we can use the strong Markov property and obtain

$$\mathbb{E}\left[e^{-um(\boldsymbol{W})\Theta(\varepsilon^{-1})t_k}\mathbb{I}\left\{\Gamma = t_k\right\}\right] = \mathbb{E}\left[e^{-um(\boldsymbol{W})\Theta(\varepsilon^{-1})t_k}\mathbb{I}\left\{\boldsymbol{\theta}_t \in \boldsymbol{\Omega}, t \in [0, t_k), \boldsymbol{\theta}_{t_k} \notin \boldsymbol{\Omega}\right\}\right]$$

$$= \mathbb{E}\left[e^{-um(\boldsymbol{W})\Theta(\varepsilon^{-1})\sigma_k}\mathbb{I}\left\{\boldsymbol{\theta}_{t+t_{k-1}} \in \boldsymbol{\Omega}, t \in [0, \sigma_k)\right\}\mathbb{I}\left\{\boldsymbol{\theta}_{t_k} \notin \boldsymbol{\Omega}\right\}\right.$$

$$\left. \cdot \prod_{i=1}^{k-1}e^{-um(\boldsymbol{W})\Theta(\varepsilon^{-1})\sigma_i}\mathbb{I}\left\{\boldsymbol{\theta}_{t+t_{i-1}} \in \boldsymbol{\Omega}, t \in [0, \sigma_i]\right\}\right]$$

$$\geq \inf_{\boldsymbol{\theta}_0 \in \boldsymbol{\Omega}^{-\bar{\rho}}}\mathbb{E}\left[e^{-um(\boldsymbol{W})\Theta(\varepsilon^{-1})\sigma_1}\mathbb{I}\left\{\boldsymbol{\theta}_t \in \boldsymbol{\Omega}^{-\bar{\rho}}, t \in [0, \sigma_1)\right\}\mathbb{I}\left\{\boldsymbol{\theta}_{\sigma_1} \notin \boldsymbol{\Omega}\right\}\right]$$

$$\cdot \inf_{\boldsymbol{\theta}_0 \in \boldsymbol{\Omega}^{-\bar{\rho}}}\mathbb{E}\left[e^{-um(\boldsymbol{W})\Theta(\varepsilon^{-1})\sigma_1}\mathbb{I}\left\{\boldsymbol{\theta}_t \in \boldsymbol{\Omega}^{-\rho}, t \in [0, \sigma_1]\right\}\right]^{k-1}.$$
(19)

**Step 2.** In this step we specifically lower bounds each terms in the lower bound of $\mathbb{E}\left[e^{-um(\boldsymbol{W})\Theta(\varepsilon^{-1})\Gamma}\right]$. Recall $\bar{\rho} = \varepsilon^\gamma$ where $\gamma$ is a constant such that the results of Lemmas 1∼ 3 holds. The escaping from the basin $\boldsymbol{\Omega}$ with a big jump $\varepsilon J_1$ occurs when $\boldsymbol{Q}_{\sigma_1-}^{-1}\boldsymbol{\Sigma}_{\sigma_1-}\varepsilon J_1 \in \boldsymbol{\Omega}$. Furthermore, $\sup_{0 \leq t < \sigma_1}\|\boldsymbol{\theta}_t - \widehat{\boldsymbol{\theta}}_t\| \leq \frac{1}{2}\varepsilon^\gamma$ with probability exponentially close to 1 (verified by Lemma 3). Meanwhile $\sigma_1 = \frac{2}{\alpha}\varepsilon^{\alpha\delta}$ in the $\alpha$-stable ($\mathcal{S}\alpha\mathcal{S}$) distribution is much larger than $v_\varepsilon = \mathcal{O}(\ln(1/\varepsilon))$ with sufficient small $\varepsilon$, $\widehat{\boldsymbol{\theta}}_t$ reaches a $\frac{1}{2}\varepsilon^\gamma$-neighborhood of the optimum $\boldsymbol{\theta}^*$ which only requires time $v_\varepsilon$. So this actually means $\sup_{0 \leq t < \sigma_1}\|\boldsymbol{\theta}_t - \boldsymbol{\theta}^*\| \leq \varepsilon^\gamma$. In this way, to obtain the final lower bound results, we only need to estimate the escaping probability $\mathbb{P}\left(\boldsymbol{Q}_{\boldsymbol{\theta}}^{-1}\boldsymbol{\Sigma}_{\boldsymbol{\theta}}\varepsilon J_1 \in \boldsymbol{\Omega}^{-\varepsilon^\gamma}\right)$ and $\mathbb{P}\left(\boldsymbol{Q}_{\boldsymbol{\theta}}^{-1}\boldsymbol{\Sigma}_{\boldsymbol{\theta}}\varepsilon J_1 \notin \boldsymbol{\Omega}\right)$ uniformly over $\|\boldsymbol{\theta} - \boldsymbol{\theta}^*\| \leq \varepsilon^\gamma$.

Based on the results in Eqn. (14) and (15) which provides the upper bound of $\mathbb{P}\left(\boldsymbol{Q}_{\boldsymbol{\theta}^*}^{-1}\boldsymbol{\Sigma}_{\boldsymbol{\theta}^*}\varepsilon J_1 \notin \boldsymbol{\Omega}\right)$ and some important inequalities, we first upper bound the term

$\inf_{\|\boldsymbol{\theta}-\boldsymbol{\theta}^*\|\leq\varepsilon^{-\gamma}}\mathbb{P}\left(\boldsymbol{Q}_{\boldsymbol{\theta}}^{-1}\boldsymbol{\Sigma}_{\boldsymbol{\theta}}\varepsilon J_k\notin\boldsymbol{\Omega}^{-\varepsilon^{-\gamma}}\right)$ as follows:

$$\inf_{\|\boldsymbol{\theta}-\boldsymbol{\theta}^*\|\leq\varepsilon^\gamma}\mathbb{P}\left(\boldsymbol{Q}_{\boldsymbol{\theta}}^{-1}\boldsymbol{\Sigma}_{\boldsymbol{\theta}}\varepsilon J_k\notin\boldsymbol{\Omega}^{-\varepsilon^\gamma}\right)$$

$$=\inf_{\|\boldsymbol{\theta}-\boldsymbol{\theta}^*\|\leq\varepsilon^\gamma}\mathbb{P}\left(\boldsymbol{Q}_{\boldsymbol{\theta}}^{-1}\boldsymbol{\Sigma}_{\boldsymbol{\theta}}\varepsilon J_k\notin\boldsymbol{\Omega}^{-\varepsilon^\gamma},\|\varepsilon J_k\|\leq R\right)+\mathbb{P}\left(\boldsymbol{Q}_{\boldsymbol{\theta}}^{-1}\boldsymbol{\Sigma}_{\boldsymbol{\theta}}\varepsilon J_k\notin\boldsymbol{\Omega}^{-\varepsilon^\gamma},\|\varepsilon J_k\|>R\right)$$

$$\overset{①}{\leq}\mathbb{P}\left(\boldsymbol{Q}_{\boldsymbol{\theta}^*}^{-1}\boldsymbol{\Sigma}_{\boldsymbol{\theta}^*}\varepsilon J_k\notin\boldsymbol{\Omega},\|\varepsilon J_k\|\leq R\right)+\frac{\delta'}{4}\frac{\Theta(\varepsilon^{-1})}{\Theta(\varepsilon^{-\delta})}+\mathbb{P}\left(\boldsymbol{Q}_{\boldsymbol{\theta}}^{-1}\boldsymbol{\Sigma}_{\boldsymbol{\theta}}\varepsilon J_k\notin\boldsymbol{\Omega}^{-\varepsilon^{-\gamma}},\|\varepsilon J_k\|>R\right)$$

$$\leq\mathbb{P}\left(\boldsymbol{Q}_{\boldsymbol{\theta}^*}^{-1}\boldsymbol{\Sigma}_{\boldsymbol{\theta}^*}\varepsilon J_k\notin\boldsymbol{\Omega}\right)+\frac{\delta'}{4}\frac{\Theta(\varepsilon^{-1})}{\Theta(\varepsilon^{-\delta})}+\mathbb{P}\left(\|\varepsilon J_k\|>R\right)$$

$$\overset{②}{\leq}m(\boldsymbol{\mathcal{W}})(1+\delta')\frac{\Theta(\varepsilon^{-1})}{\Theta(\varepsilon^{-\delta})}+\frac{\delta'}{4}\frac{\Theta(\varepsilon^{-1})}{\Theta(\varepsilon^{-\delta})}+\frac{\delta'}{4}\frac{\Theta(\varepsilon^{-1})}{\Theta(\varepsilon^{-\delta})}$$

$$=m(\boldsymbol{\mathcal{W}})(1+\delta'+\frac{\delta'}{2m(\boldsymbol{\mathcal{W}})})\frac{\Theta(\varepsilon^{-1})}{\Theta(\varepsilon^{-\delta})}\overset{③}{\leq}m(\boldsymbol{\mathcal{W}})(1+\rho)\frac{\Theta(\varepsilon^{-1})}{\Theta(\varepsilon^{-\delta})},$$

where ① uses the result in Eqn. (14), ② uses Eqn. (13), and in ③ we set $\delta'$ enough small via setting small $\varepsilon$ such that $\rho\geq\delta'+\frac{\delta'}{2m(\boldsymbol{\mathcal{W}})}$. So for any $\rho$ we choose $\delta'>0$ small enough to upper bound

$$\inf_{\|\boldsymbol{\theta}-\boldsymbol{\theta}^*\|\leq\varepsilon^\gamma}\mathbb{P}\left(\boldsymbol{Q}_{\boldsymbol{\theta}}^{-1}\boldsymbol{\Sigma}_{\boldsymbol{\theta}}\varepsilon J_k\in\boldsymbol{\Omega}^{-\varepsilon^\gamma}\right)=1-\inf_{\|\boldsymbol{\theta}-\boldsymbol{\theta}^*\|\leq\varepsilon^\gamma}\mathbb{P}\left(\boldsymbol{Q}_{\boldsymbol{\theta}}^{-1}\boldsymbol{\Sigma}_{\boldsymbol{\theta}}\varepsilon J_k\notin\boldsymbol{\Omega}^{-\varepsilon^\gamma}\right)\geq1-a_\varepsilon(1+\rho).$$

Similarly, we only need to lower bound the remaining term $\inf_{\|\boldsymbol{\theta}-\boldsymbol{\theta}^*\|\leq\varepsilon^\gamma}\mathbb{P}\left(\boldsymbol{Q}_{\boldsymbol{\theta}}^{-1}\boldsymbol{\Sigma}_{\boldsymbol{\theta}}\varepsilon J_k\notin\boldsymbol{\Omega}\right)$ as follows:

$$\inf_{\|\boldsymbol{\theta}-\boldsymbol{\theta}^*\|\leq\varepsilon^\gamma}\mathbb{P}\left(\boldsymbol{Q}_{\boldsymbol{\theta}}^{-1}\boldsymbol{\Sigma}_{\boldsymbol{\theta}}\varepsilon J_k\notin\boldsymbol{\Omega}\right)$$

$$=\inf_{\|\boldsymbol{\theta}-\boldsymbol{\theta}^*\|\leq\varepsilon^\gamma}\mathbb{P}\left(\boldsymbol{Q}_{\boldsymbol{\theta}}^{-1}\boldsymbol{\Sigma}_{\boldsymbol{\theta}}\varepsilon J_k\notin\boldsymbol{\Omega},\|\varepsilon J_k\|\leq R\right)+\mathbb{P}\left(\boldsymbol{Q}_{\boldsymbol{\theta}}^{-1}\boldsymbol{\Sigma}_{\boldsymbol{\theta}}\varepsilon J_k\notin\boldsymbol{\Omega},\|\varepsilon J_k\|>R\right)$$

$$\overset{①}{\geq}\inf_{\|\boldsymbol{\theta}-\boldsymbol{\theta}^*\|\leq\varepsilon^\gamma}\mathbb{P}\left(\boldsymbol{Q}_{\boldsymbol{\theta}^*}^{-1}\boldsymbol{\Sigma}_{\boldsymbol{\theta}^*}\varepsilon J_k\notin\boldsymbol{\Omega}^{-\varepsilon^\gamma},\|\varepsilon J_k\|\leq R\right)-\frac{\delta'}{4}\frac{\Theta(\varepsilon^{-1})}{\Theta(\varepsilon^{-\delta})}+\mathbb{P}\left(\boldsymbol{Q}_{\boldsymbol{\theta}}^{-1}\boldsymbol{\Sigma}_{\boldsymbol{\theta}}\varepsilon J_k\in\boldsymbol{\Omega},\|\varepsilon J_k\|>R\right)$$

$$\geq\inf_{\|\boldsymbol{\theta}-\boldsymbol{\theta}^*\|\leq\varepsilon^\gamma}\mathbb{P}\left(\boldsymbol{Q}_{\boldsymbol{\theta}^*}^{-1}\boldsymbol{\Sigma}_{\boldsymbol{\theta}^*}\varepsilon J_k\notin\boldsymbol{\Omega},\|\varepsilon J_k\|\leq R\right)-\frac{\delta'}{4}\frac{\Theta(\varepsilon^{-1})}{\Theta(\varepsilon^{-\delta})}+\mathbb{P}\left(\boldsymbol{Q}_{\boldsymbol{\theta}}^{-1}\boldsymbol{\Sigma}_{\boldsymbol{\theta}}\varepsilon J_k\in\boldsymbol{\Omega},\|\varepsilon J_k\|>R\right)$$

$$\geq\mathbb{P}\left(\boldsymbol{Q}_{\boldsymbol{\theta}^*}^{-1}\boldsymbol{\Sigma}_{\boldsymbol{\theta}^*}\varepsilon J_k\notin\boldsymbol{\Omega},\|\varepsilon J_k\|\leq R\right)-\frac{\delta'}{4}\frac{\Theta(\varepsilon^{-1})}{\Theta(\varepsilon^{-\delta})}$$

$$\overset{②}{\geq}\mathbb{P}\left(\boldsymbol{Q}_{\boldsymbol{\theta}^*}^{-1}\boldsymbol{\Sigma}_{\boldsymbol{\theta}^*}\varepsilon J_k\notin\boldsymbol{\Omega}\right)-\frac{\delta'}{2}\frac{\Theta(\varepsilon^{-1})}{\Theta(\varepsilon^{-\delta})}$$

$$\overset{③}{\geq}m(\boldsymbol{\mathcal{W}})\left(1-\delta'-\frac{\delta'}{2m(\boldsymbol{\mathcal{W}})}\right)\frac{\Theta(\varepsilon^{-1})}{\Theta(\varepsilon^{-\delta})}\geq m(\boldsymbol{\mathcal{W}})(1-\rho)\frac{\Theta(\varepsilon^{-1})}{\varepsilon^{-\delta}}=a_\varepsilon(1-\rho),$$

where ① uses the result in Eqn. (14), ② uses Eqn. (15), and ③ uses Eqn. (13).

Next, for any $\rho>0$ and $\varepsilon$ we can obtain Laplace transforms for any $u>-1$ as follows:

$$\inf_{\boldsymbol{\theta}_0\in\boldsymbol{\Omega}^{-\varepsilon^\gamma}}\mathbb{E}\left[e^{-um(\boldsymbol{\mathcal{W}})\Theta(\varepsilon^{-1})\sigma_1}\mathbb{I}\left\{\boldsymbol{\theta}_t\in\boldsymbol{\Omega}^{-\varepsilon^\gamma},t\in[0,\sigma_1]\right\}\right]$$

$$\geq[1-a_\varepsilon(1+\rho)]\mathbb{E}\left[\int_0^{+\infty}e^{-um(\boldsymbol{\mathcal{W}})\Theta(\varepsilon^{-1})\sigma_1}\cdot\Theta(\varepsilon^{-\delta})e^{-\Theta(\varepsilon^{-\delta})\sigma_1}\mathsf{d}\sigma_1\right] \qquad (20)$$

$$=\frac{1-a_\varepsilon(1+\rho)}{1+ua_\varepsilon},$$

and

$$\inf_{\boldsymbol{\theta}_0\in\boldsymbol{\Omega}^{\varepsilon^{-\gamma}}}\mathbb{E}\left[e^{-um(\boldsymbol{\mathcal{W}})\Theta(\varepsilon^{-1})\sigma_1}\mathbb{I}\left\{\boldsymbol{\theta}_t\in\boldsymbol{\Omega}^{-\varepsilon^\gamma},t\in[0,\sigma_1]\right\}\mathbb{I}\left\{\boldsymbol{\theta}_{\sigma_1}\notin\boldsymbol{\Omega}\right\}\right]$$

$$\geq[1-a_\varepsilon(1+\rho)]\mathbb{E}\left[\int_0^{+\infty}e^{-um(\boldsymbol{\mathcal{W}})\Theta(\varepsilon^{-1})\sigma_1}\cdot\Theta(\varepsilon^{-\delta})e^{-\Theta(\varepsilon^{-\delta})\sigma_1}\mathsf{d}\sigma_1\right]=\frac{a_\varepsilon(1-\rho)}{1+ua_\varepsilon}.$$

**Step 3.** Here we summarize the results in Steps 1 and 2 such that we can lower bound the desired results $\mathbb{E}\left[e^{-um(\boldsymbol{W})\Theta(\varepsilon^{-1})\Gamma}\right]$. Specifically, from Eqn. (19), for any $\boldsymbol{\theta}_0 \in \Omega^{-\varepsilon^\gamma}$ we can lower bound

$$\mathbb{E}\left[e^{-um(\boldsymbol{W})\Theta(\varepsilon^{-1})\Gamma}\right] \geq \frac{a_\varepsilon(1-\rho)}{1+ua_\varepsilon} \sum_{k=1}^{+\infty}\left(\frac{1-a_\varepsilon(1+\rho)}{1+ua_\varepsilon}\right)^{k-1} = \frac{1-\rho}{1+u+\rho},$$

where $\rho \downarrow 0$ as $\varepsilon \downarrow 0$. The proof is completed. $\qquad\square$

### D.2 Proof of Theorem 2

*Proof.* In this step we prove the sequence $\{\widehat{\boldsymbol{\theta}}_t\}$ produced by Eqn. (8) or (9) locates in a very small neighborhood of the optimum solution $\boldsymbol{\theta}^*$ of the local basin $\boldsymbol{\Omega}$ after a very small time interval.

**Step 1.** In this step, we prove the first part of Theorem 2. Since we assume the function is locally strongly convex, by using Lemmas 3 and 4, we know that the sequence $\{\widehat{\boldsymbol{\theta}}_t\}$ produced by Eqn. (8) or (9) exponentially converges to the minimum $\boldsymbol{\theta}^*$ at the current local basin $\boldsymbol{\Omega}$. So for any initialization $\boldsymbol{\theta}_0 \in \boldsymbol{\Omega}$, we have

$$\|\widehat{\boldsymbol{\theta}}_t - \boldsymbol{\theta}^*\|_2^2 \leq c_1 \exp\left(-c_2 t\right),$$

where $c_1 = \frac{2\Delta}{\mu}$ and $c_2 = \frac{2\mu\tau}{\beta_1(v_{\max}+\epsilon)+\mu\tau}\left(\beta_1 - \frac{\beta_2}{4}\right)$ in Adam, $c_1 = \frac{2\Delta}{\mu}$ and $c_2 = 2\mu$ in SGD. Therefore, for any initialization $\boldsymbol{\theta}_0 \in \boldsymbol{\Omega}$ and sufficient small $\varepsilon$, we can obtain

$$\|\widehat{\boldsymbol{\theta}}_t - \boldsymbol{\theta}^*\|_2^2 \leq \varepsilon^{\bar{\rho}} \text{ if } t \geq v_\varepsilon = \frac{1}{c_3} \ln\left(\frac{2\Delta}{\mu\varepsilon^{\bar{\rho}}}\right).$$

where $c_3 = \frac{2\mu\tau}{\beta_1(v_{\max}+\epsilon)+\mu\tau}\left(\beta_1 - \frac{\beta_2}{4}\right)$ in ADAM, $c_3 = 2\mu$ in SGD, and $\Delta = \boldsymbol{F}(\boldsymbol{\theta}_0) - \boldsymbol{F}(\boldsymbol{\theta}^*)$.

**Step 2.** In this step, we prove the second part of Theorem 2. By replacing $p$ with $p/2$ in Lemma E.5, we can directly obtain the results. $\qquad\square$

## E  Proofs of Auxiliary Theories and Lemmas in Appendix C

Before analysis, we first introduce two useful lemmas which will be used in subsequent analysis.

**Lemma 4** (Grönwall's Lemma [12]). *Suppose $g(s) : [0, t_0]$ is a non-negative continues function. If for almost $s \in [0, t_0]$*

$$g'(s) \leq q(s)g(s)$$

*where $q(s)$ is a continuous function, then we have*

$$g(t) \leq g(0) \exp\left(\int_0^t q(s) ds\right).$$

**Lemma 5** (Theorem 5.3 in [13]). *Consider a set $\boldsymbol{A} \in \mathcal{B}(\mathbb{R} \backslash 0)$ with $0 \in \bar{\boldsymbol{A}}$ and a function $f : \mathbb{R} \to \mathbb{R}$ with Borel measurable and finite on $\boldsymbol{A}$. Then we have*
*(1) The process $(\int_0^t \int_{\boldsymbol{A}} f(x)\nu(ds, dx))_{0 \leq t \leq T}$ is a compound Poisson process with characteristic function*

$$\mathbb{E}\left(\exp\left(i\lambda \int_0^t \int_{\boldsymbol{A}} f(x)\mu^L(ds, dx)\right)\right) = \exp\left(t \int_{\boldsymbol{A}} (e^{i\lambda f(x)} - 1)\nu(dx)\right).$$

*(2) If $f \in L^1(\boldsymbol{A})$, then*

$$\mathbb{E}\left(\int_0^t \int_{\boldsymbol{A}} f(x)\mu^L(ds, dx)\right) = t \int_{\boldsymbol{A}} f(x)\nu(dx).$$

### E.1 Proof of Theorem 3 for the Linear Convergence of Lévy-driven SGD SDE (8)

*Proof.* **Step 1.** In this step, we upper bound the gradient norm of the Lyapunov function $\mathcal{L}(t) = \boldsymbol{F}(\widehat{\boldsymbol{\theta}}_t) - \boldsymbol{F}(\boldsymbol{\theta}^*)$ of (8) with $\widehat{\boldsymbol{Q}}_t = \boldsymbol{I}$ and $\beta_2 = 0$. More specifically, we can upper bound $\mathsf{d}\mathcal{L}(t)$ as follows:

$$\mathsf{d}\mathcal{L}(t) = \langle \nabla \boldsymbol{F}(\widehat{\boldsymbol{\theta}}_t), \mathsf{d}\widehat{\boldsymbol{\theta}}_t \rangle = \left\langle \nabla \boldsymbol{F}(\widehat{\boldsymbol{\theta}}_t), -\nabla \boldsymbol{F}(\widehat{\boldsymbol{\theta}}_t) \right\rangle = -\|\nabla \boldsymbol{F}(\widehat{\boldsymbol{\theta}}_t)\|_2^2. \tag{21}$$

**Step 2.** Here we prove the linear convergence behavior of $\mathcal{L}(t) = \boldsymbol{F}(\widehat{\boldsymbol{\theta}}_t) - \boldsymbol{F}(\boldsymbol{\theta}^*)$ by using the results in Step 1. Since $\boldsymbol{F}(\boldsymbol{\theta})$ is locally $\mu$-strongly convex, then we have

$$\boldsymbol{F}(\boldsymbol{y}) \geq \boldsymbol{F}(\boldsymbol{\theta}) + \langle \nabla \boldsymbol{F}(\boldsymbol{\theta}), \boldsymbol{y} - \boldsymbol{\theta} \rangle + \frac{\mu}{2}\|\boldsymbol{y} - \boldsymbol{\theta}\|_2^2.$$

Next, by minimizing $\boldsymbol{y}$ on both side ($\boldsymbol{y} = \boldsymbol{\theta}_*$ for the left side and $\boldsymbol{y} = \boldsymbol{\theta} - \frac{1}{\mu}\nabla \boldsymbol{F}(\boldsymbol{\theta})$ for the right side), it yields

$$\|\nabla \boldsymbol{F}(\boldsymbol{\theta})\|_2^2 \geq 2\mu(\boldsymbol{F}(\boldsymbol{\theta}) - \boldsymbol{F}(\boldsymbol{\theta}_*)). \tag{22}$$

Hence, plugging the above equation into Eqn. (21) gives

$$\mathsf{d}\mathcal{L}(t) \leq -2\mu(\boldsymbol{F}(\boldsymbol{\theta}) - \boldsymbol{F}(\boldsymbol{\theta}_*)) = -2\mu\mathcal{L}(t).$$

In this way, by using the result in Lemma 4, we can easily obtain

$$\mathcal{L}(t) \leq \mathcal{L}(0) \exp\left(-\int_0^t 2\mu \mathsf{d}s\right) \leq \Delta \exp\left(-2\mu t\right),$$

where we use $\mathcal{L}(0) = \boldsymbol{F}(\widehat{\boldsymbol{\theta}}_0) - \boldsymbol{F}(\boldsymbol{\theta}^*) = \Delta$ where $\boldsymbol{\theta}^*$ is the optimum of the current basin.

**Step 3.** Finally, we explore the local strong-convexity of $\boldsymbol{F}(\boldsymbol{\theta})$ to show the linear convergence of $\|\widehat{\boldsymbol{\theta}}_t - \boldsymbol{\theta}^*\|_2^2$. Specifically, by using the strongly convex property of $\boldsymbol{F}(\boldsymbol{\theta})$, we can obtain

$$\boldsymbol{F}(\boldsymbol{\theta}) - \boldsymbol{F}(\boldsymbol{\theta}^*) \geq \frac{\mu}{2}\|\boldsymbol{\theta} - \boldsymbol{\theta}^*\|_2^2.$$

So this gives

$$\|\widehat{\boldsymbol{\theta}}_t - \boldsymbol{\theta}^*\|_2^2 \leq \frac{2\Delta}{\mu} \exp\left(-2\mu t\right).$$

The proof is completed. $\qquad\qquad\qquad\qquad\qquad\qquad\qquad\qquad\qquad\qquad\qquad\qquad\qquad$ □

### E.2 Proof of Theorem 4 for the Linear Convergence of Lévy-driven ADAM SDE (9)

*Proof.* **Step 1.** In this step, we upper bound the gradient norm of the Lyapunov function of (9) defined as

$$\mathcal{L}(t) = \boldsymbol{F}(\widehat{\boldsymbol{\theta}}_t) - \boldsymbol{F}(\boldsymbol{\theta}^*) + \frac{1}{2}\|\widehat{\boldsymbol{m}}_t\|_{\widehat{\boldsymbol{s}}_t^{-1}}^2, \tag{23}$$

where $\widehat{\boldsymbol{s}}_t = \frac{h_t}{\mu_t}\left(\sqrt{\omega_t \widehat{\boldsymbol{v}}_t} + \epsilon\right)$ with $h_t = \beta_1$, $\mu_t = (1 - e^{-\beta_1 t})^{-1}$ and $\omega_t = (1 - e^{-\beta_2 t})^{-1}$. Here we define $\|\boldsymbol{x}\|_{\boldsymbol{y}}^2 = \sum_i y_i x_i^2$. Then we can compute the derivative of Lyapunov function as

$$\mathsf{d}\mathcal{L}(t) = \underbrace{\langle \nabla \boldsymbol{F}(\widehat{\boldsymbol{\theta}}_t), \mathsf{d}\widehat{\boldsymbol{\theta}}_t \rangle + \sum_{i=1}^d \frac{1}{\widehat{\boldsymbol{s}}_{t,i}} \widehat{\boldsymbol{m}}_{t,i} \mathsf{d}\widehat{\boldsymbol{m}}_{t,i} - \sum_{i=1}^d \frac{1}{2\widehat{\boldsymbol{s}}_{t,i}^2} \widehat{\boldsymbol{m}}_{t,i}^2 \nabla_{\widehat{\boldsymbol{v}}_t} \widehat{\boldsymbol{s}}_{t,i} \mathsf{d}\widehat{\boldsymbol{v}}_{t,i}}_{P_1} - \underbrace{\sum_{i=1}^d \frac{1}{2\widehat{\boldsymbol{s}}_{t,i}^2} \widehat{\boldsymbol{m}}_{t,i}^2 \nabla_t \widehat{\boldsymbol{s}}_{t,i}}_{P_2},$$

$$\tag{24}$$

where $\widehat{\boldsymbol{m}}_{t,i}$, $\widehat{\boldsymbol{v}}_{t,i}$ and $\widehat{\boldsymbol{s}}_{t,i}$ respectively denote the $i$-th entries of $\widehat{\boldsymbol{m}}_t$, $\widehat{\boldsymbol{v}}_t$ and $\widehat{\boldsymbol{s}}_t$.

We first consider Adam in which $h_t = \beta_1$, $\mu_t = (1 - e^{-\beta_1 t})^{-1}$, and $\omega_t = (1 - e^{-\beta_2 t})^{-1}$. We also assume $\beta_1 \leq \beta_2 \leq 2\beta_1$ which is consistent with the practical setting where $\beta_1 = 0.9$ and $\beta_2 = 0.999$.

Let $[\nabla f_{\mathcal{S}_t}(\widehat{\boldsymbol{\theta}}_t)^2]_i$ denotes the $i$-th entry of the vector $\nabla f_{\mathcal{S}_t}(\widehat{\boldsymbol{\theta}}_t)^2$. Under this setting, we can first upper bound the first term $P_1$ as follows:

$$P_1$$

$$= \langle \nabla \boldsymbol{F}(\widehat{\boldsymbol{\theta}}_t), \mathrm{d}\widehat{\boldsymbol{\theta}}_t \rangle + \sum_{i=1}^{d} \frac{1}{\widehat{\boldsymbol{s}}_{t,i}} \widehat{\boldsymbol{m}}_{t,i} \mathrm{d}\widehat{\boldsymbol{m}}_{t,i} - \sum_{i=1}^{d} \frac{1}{2\widehat{\boldsymbol{s}}_{t,i}^2} \widehat{\boldsymbol{m}}_{t,i}^2 \nabla_{\widehat{\boldsymbol{v}}_t} \widehat{\boldsymbol{s}}_{t,i} \mathrm{d}\widehat{\boldsymbol{v}}_{t,i}$$

$$= \langle \nabla \boldsymbol{F}(\widehat{\boldsymbol{\theta}}_t), -\frac{\mu_t \widehat{\boldsymbol{m}}_t}{\sqrt{\omega_t \widehat{\boldsymbol{v}}_t} + \epsilon} \rangle + \beta_1 \sum_{i=1}^{d} \frac{1}{\widehat{\boldsymbol{s}}_{t,i}} \widehat{\boldsymbol{m}}_{t,i}(\nabla \boldsymbol{F}_i(\boldsymbol{\theta}_t) - \widehat{\boldsymbol{m}}_{t,i}) - \sum_{i=1}^{d} \frac{\beta_2}{2\widehat{\boldsymbol{s}}_{t,i}^2} \widehat{\boldsymbol{m}}_{t,i}^2([\nabla f_{\mathcal{S}_t}(\widehat{\boldsymbol{\theta}}_t)^2]_i - \widehat{\boldsymbol{v}}_{t,i}) \nabla_{\widehat{\boldsymbol{v}}_t} \widehat{\boldsymbol{s}}_{t,i}$$

$$= -\beta_1 \left\langle \nabla \boldsymbol{F}(\widehat{\boldsymbol{\theta}}_t), \frac{\widehat{\boldsymbol{m}}_t}{\widehat{\boldsymbol{s}}_t} \right\rangle + \beta_1 \left\langle \nabla \boldsymbol{F}(\widehat{\boldsymbol{\theta}}_t), \frac{\widehat{\boldsymbol{m}}_t}{\widehat{\boldsymbol{s}}_t} \right\rangle - \beta_1 \sum_{i=1}^{d} \frac{1}{\widehat{\boldsymbol{s}}_{t,i}} \widehat{\boldsymbol{m}}_{t,i}^2 - \beta_2 \sum_{i=1}^{d} \frac{1}{2\widehat{\boldsymbol{s}}_{t,i}^2} \widehat{\boldsymbol{m}}_{t,i}^2([\nabla f_{\mathcal{S}_t}(\widehat{\boldsymbol{\theta}}_t)^2]_i - \widehat{\boldsymbol{v}}_{t,i}) \nabla_{\widehat{\boldsymbol{v}}_t} \widehat{\boldsymbol{s}}_{t,i}$$

$$= -\beta_1 \sum_{i=1}^{d} \frac{1}{\widehat{\boldsymbol{s}}_{t,i}} \widehat{\boldsymbol{m}}_{t,i}^2 - \beta_2 \sum_{i=1}^{d} \frac{1}{2\widehat{\boldsymbol{s}}_{t,i}^2} \widehat{\boldsymbol{m}}_{t,i}^2([\nabla f_{\mathcal{S}_t}(\widehat{\boldsymbol{\theta}}_t)^2]_i - \widehat{\boldsymbol{v}}_{t,i}) \nabla_{\widehat{\boldsymbol{v}}_t} \widehat{\boldsymbol{s}}_{t,i}.$$

Next, we plug the specific formulation of $\nabla_{\widehat{\boldsymbol{v}}_t} \widehat{\boldsymbol{s}}_{t,i} = \frac{\beta_1 \sqrt{\omega_t}}{2\mu_t \sqrt{\widehat{\boldsymbol{v}}_{t,i}}}$ into the above equation and obtain:

$$P_1 = -\beta_1 \sum_{i=1}^{d} \frac{1}{\widehat{\boldsymbol{s}}_{t,i}} \widehat{\boldsymbol{m}}_{t,i}^2 - \beta_2 \sum_{i=1}^{d} \frac{1}{2\widehat{\boldsymbol{s}}_{t,i}^2} \widehat{\boldsymbol{m}}_{t,i}^2([\nabla f_{\mathcal{S}_t}(\widehat{\boldsymbol{\theta}}_t)^2]_i - \widehat{\boldsymbol{v}}_{t,i}) \frac{\beta_1 \sqrt{\omega_t}}{2\mu_t \sqrt{\widehat{\boldsymbol{v}}_{t,i}}}$$

$$= -\beta_1^2 \sum_{i=1}^{d} \frac{\widehat{\boldsymbol{m}}_{t,i}^2}{\mu_t \widehat{\boldsymbol{s}}_{t,i}^2} \left( \epsilon + (1 - \frac{\beta_2}{4\beta_1})\sqrt{\omega_t \widehat{\boldsymbol{v}}_{t,i}} + \frac{\beta_2}{4\beta_1} \frac{[\nabla f_{\mathcal{S}_t}(\widehat{\boldsymbol{\theta}}_t)^2]_i \sqrt{\omega_t}}{\sqrt{\widehat{\boldsymbol{v}}_{t,i}}} \right)$$

$$= -\beta_1 \sum_{i=1}^{d} \frac{\widehat{\boldsymbol{m}}_{t,i}^2}{\widehat{\boldsymbol{s}}_{t,i}} \left( 1 - \frac{\beta_2}{4\beta_1} + \frac{\beta_2 \epsilon}{4\beta_1(\epsilon + \sqrt{\omega_t \widehat{\boldsymbol{v}}_{t,i}})} + \frac{\beta_2}{4\beta_1} \frac{[\nabla f_{\mathcal{S}_t}(\widehat{\boldsymbol{\theta}}_t)^2]_i \sqrt{\omega_t}}{\sqrt{\widehat{\boldsymbol{v}}_{t,i}}(\epsilon + \sqrt{\omega_t \widehat{\boldsymbol{v}}_{t,i}})} \right)$$

$$\leq -\left( \beta_1 - \frac{\beta_2}{4} \right) \sum_{i=1}^{d} \frac{\widehat{\boldsymbol{m}}_{t,i}^2}{\widehat{\boldsymbol{s}}_{t,i}} = -\left( \beta_1 - \frac{\beta_2}{4} \right) \|\widehat{\boldsymbol{m}}_t\|_{\widehat{\boldsymbol{s}}_t^{-1}}^2.$$

Then we consider the second term $P_2$ under the setting $\widehat{\boldsymbol{s}}_t = \frac{\beta_1}{\mu_t} \left( \sqrt{\omega_t \widehat{\boldsymbol{v}}_t} + \epsilon \right)$ with $\mu_t = (1 - e^{-\beta_1 t})^{-1}$ and $\omega_t = (1 - e^{-\beta_2 t})^{-1}$. Similarly, we can upper bound $P_2$ as

$$P_2 = -\sum_{i=1}^{d} \frac{1}{2\widehat{\boldsymbol{s}}_{t,i}^2} \widehat{\boldsymbol{m}}_{t,i}^2 \nabla_t \widehat{\boldsymbol{s}}_{t,i}$$

$$= -\beta_1 \sum_{i=1}^{d} \frac{1}{2\widehat{\boldsymbol{s}}_{t,i}^2} \widehat{\boldsymbol{m}}_{t,i}^2 \left( \beta_1 e^{-\beta_1 t} \left( \epsilon + \sqrt{\frac{\widehat{\boldsymbol{v}}_{t,i}}{1 - e^{-\beta_2 t}}} \right) - \frac{1}{2}\beta_2 e^{-\beta_2 t} \frac{1 - e^{-\beta_1 t}}{1 - e^{-\beta_2 t}} \sqrt{\frac{\widehat{\boldsymbol{v}}_{t,i}}{1 - e^{-\beta_2 t}}} \right)$$

$$= -\frac{\beta_1^2}{2} \sum_{i=1}^{d} \frac{\widehat{\boldsymbol{m}}_{t,i}^2}{\mu_t \widehat{\boldsymbol{s}}_{t,i}^2} \frac{e^{-\beta_1 t}}{1 - e^{-\beta_1 t}} \left( \epsilon + \left( 1 - \frac{\beta_2 e^{-\beta_2 t}(1 - e^{-\beta_1 t})}{2\beta_1 e^{-\beta_1 t}(1 - e^{-\beta_2 t})} \right) \sqrt{\frac{\widehat{\boldsymbol{v}}_{t,i}}{1 - e^{-\beta_2 t}}} \right)$$

$$\overset{\text{①}}{\leq} -\frac{\beta_1^2}{2} \sum_{i=1}^{d} \frac{\widehat{\boldsymbol{m}}_{t,i}^2}{\mu_t \widehat{\boldsymbol{s}}_{t,i}^2} \frac{e^{-\beta_1 t}}{1 - e^{-\beta_1 t}} \left( \epsilon + \left( 1 - \frac{\beta_2}{2\beta_1} \right) \sqrt{\frac{\widehat{\boldsymbol{v}}_{t,i}}{1 - e^{-\beta_2 t}}} \right)$$

$$= -\frac{\beta_1}{2} \sum_{i=1}^{d} \frac{\widehat{\boldsymbol{m}}_{t,i}^2}{\widehat{\boldsymbol{s}}_{t,i}} \frac{e^{-\beta_1 t}}{1 - e^{-\beta_1 t}} \left( 1 - \frac{\beta_2}{2\beta_1} + \frac{\beta_2 \epsilon}{2\beta_1(\epsilon + \sqrt{\omega_t \widehat{\boldsymbol{v}}_{t,i}})} \right)$$

$$\leq -\frac{1}{2}\left( \beta_1 - \frac{\beta_2}{2} \right) \frac{e^{-\beta_1 t}}{1 - e^{-\beta_1 t}} \sum_{i=1}^{d} \frac{\widehat{\boldsymbol{m}}_{t,i}^2}{\widehat{\boldsymbol{s}}_{t,i}} = -\frac{1}{2}\left( \beta_1 - \frac{\beta_2}{2} \right) \frac{e^{-\beta_1 t}}{1 - e^{-\beta_1 t}} \|\widehat{\boldsymbol{m}}_t\|_{\widehat{\boldsymbol{s}}_t^{-1}}^2 \overset{\text{②}}{\leq} 0,$$

where ① uses $\frac{\beta_2 e^{-\beta_2 t}(1 - e^{-\beta_1 t})}{2\beta_1 e^{-\beta_1 t}(1 - e^{-\beta_2 t})} \leq \frac{\beta_2}{2\beta_1}$ since $\beta_2 \geq \beta_1$; in ② we assume $\beta_1 - \frac{\beta_2}{2} > 0$. Therefore, by combining the upper bounds of $P_1$ and $P_2$ we can upper bound

$$\mathrm{d}\mathcal{L}(t) \leq -\left[ \beta_1 - \frac{\beta_2}{4} \right] \|\widehat{\boldsymbol{m}}_t\|_{\widehat{\boldsymbol{s}}_t^{-1}}^2. \tag{25}$$

On the other hand, noting $h_t = \beta_1$, $\mu_t = (1 - e^{-\beta_1 t})^{-1}$ and $\omega_t = (1 - e^{-\beta_2 t})^{-1}$, we have

$$\widehat{s}_{t,i} = \frac{h_t}{\mu_t}\left(\epsilon + \sqrt{\omega_t \widehat{v}_{t,i}}\right) = \beta_1(1 - e^{-\beta_1 t})\left(\epsilon + \sqrt{\frac{\widehat{v}_{t,i}}{1 - e^{-\beta_2 t}}}\right) \leq \beta_1\left(\epsilon + \frac{1 - e^{-\beta_1 t}}{\sqrt{1 - e^{-\beta_2 t}}}\sqrt{\widehat{v}_{t,i}}\right)$$

$$\overset{①}{\leq} \beta_1\left(\epsilon + \frac{1 - e^{-\beta_1 t}}{1 - e^{-\beta_2 t/2}}\sqrt{\widehat{v}_{t,i}}\right) \overset{②}{\leq} \beta_1\left(\epsilon + v_{\max}\right),$$

where ① uses $\sqrt{1 - x} \geq 1 - \sqrt{x}$ for $0 \leq x \leq 1$ and ② holds since $\sqrt{\widehat{v}_{t,i}} \leq v_{\max}$. By using the assumption $\|\widehat{m}_t\|^2 \geq \tau \|\nabla F(\widehat{\theta}_t)\|^2$, we can establish

$$\|\widehat{m}_t\|_{\widehat{s}_t^{-1}}^2 \geq \frac{1}{\beta_1\left(\epsilon + v_{\max}\right)}\|\widehat{m}_t\|_2^2 \geq \frac{\tau}{\beta_1\left(\epsilon + v_{\max}\right)}\|\nabla F(\widehat{\theta}_t)\|_2^2. \tag{26}$$

Then from the locally $\mu$-strongly convex property Eqn. (22):

$$\|\nabla F(\theta)\|_2^2 \geq 2\mu(F(\theta) - F(\theta_*)).$$

then we plug the above inequality into Eqn. (26) and establish

$$\|\widehat{m}_t\|_{\widehat{s}_t^{-1}}^2 \geq \frac{1}{\beta_1\left(\epsilon + v_{\max}\right)}\|\widehat{m}_t\|_2^2 \geq \frac{2\mu\tau}{\beta_1\left(\epsilon + v_{\max}\right)}(F(\widehat{\theta}_t) - F(\theta_*)).$$

Finally, we can write Eqn. (25) as

$$\mathsf{d}\mathcal{L}(t) \leq -\frac{2\mu\tau}{\beta_1\left(\epsilon + v_{\max}\right) + \mu\tau}\left[\beta_1 - \frac{\beta_2}{4}\right]\left(\frac{1}{2} + \frac{\beta_1\left(\epsilon + v_{\max}\right)}{2\mu\tau}\right)\|\widehat{m}_t\|_{\widehat{s}_t^{-1}}^2$$

$$\leq -\frac{2\mu\tau}{\beta_1\left(\epsilon + v_{\max}\right) + \mu\tau}\left[\beta_1 - \frac{\beta_2}{4}\right]\left(F(\widehat{\theta}_t) - F(\theta^*) + \frac{1}{2}\|\widehat{m}_t\|_{\widehat{s}_t^{-1}}^2\right)$$

$$= -c_1\mathcal{L}(t),$$

where $c_1 = \frac{2\mu\tau}{\beta_1(\epsilon + v_{\max}) + \mu\tau}\left[\beta_1 - \frac{\beta_2}{4}\right]$.

**Step 2.** Here we prove the linear convergence behavior of $\mathcal{L}(t) = F(\widehat{\theta}_t) - F(\theta^*)$ by using the results in Step 1. More specifically, by using the result in Lemma 4, we can easily obtain

$$\mathcal{L}(t) \leq \mathcal{L}(0)\exp\left(\int_0^t c_1 \mathsf{d}s\right) = \mathcal{L}(0)\exp\left(-\frac{2\mu\tau}{\beta_1\left(\epsilon + v_{\max}\right) + \mu\tau}\left(\beta_1 - \frac{\beta_2}{4}\right)t\right)$$

$$\overset{①}{\leq} (F(\widehat{\theta}_0) - F(\theta^*))\exp\left(-\frac{2\mu\tau}{\beta_1\left(\epsilon + v_{\max}\right) + \mu\tau}\left(\beta_1 - \frac{\beta_2}{4}\right)t\right),$$

where ① uses $\mathcal{L}(0) = F(\widehat{\theta}_0) - F(\theta^*) = \Delta$ due to $\widehat{m}_0 = 0$.

**Step 3.** Finally, we explore the local strong-convexity of $F(\theta)$ to show the linear convergence of $\|\widehat{\theta}_t - \theta^*\|_2^2$. Specifically, by using the strongly convex property of $F(\theta)$, we can obtain

$$F(\theta) - F(\theta^*) \geq \frac{\mu}{2}\|\theta - \theta^*\|_2^2.$$

So this gives

$$\|\widehat{\theta}_t - \theta^*\|_2^2 \leq \frac{2\Delta}{\mu}\exp\left(-\frac{2\mu\tau}{\beta_1\left(\epsilon + v_{\max}\right) + \mu\tau}\left(\beta_1 - \frac{\beta_2}{4}\right)t\right).$$

The proof is completed. $\qquad\qquad\qquad\qquad\qquad\qquad\qquad\qquad\qquad\qquad\qquad\qquad\square$

### E.3 Proof of Lemma 1

*Proof.* To begin with, the process $\boldsymbol{\xi}$ is defined as $\boldsymbol{\xi}_t = \sum_{s \leq t}\Delta L_s\mathbb{I}\{\|L_s\| \leq \varepsilon^{-\delta}\}$. Then by setting the set $\boldsymbol{A} = \{\boldsymbol{y} \mid \|\boldsymbol{y}\| \leq \varepsilon^{-\delta}\}$ in Lemma 5 and noting $f(x) = x \in L^1(\boldsymbol{A})$, one can find

$\mathbb{E}[\boldsymbol{\xi}_t] = t \int_A f(x)\nu(\mathrm{d}x)$. Therefore, we can decompose the process $\boldsymbol{\xi}$ into two processes $\widehat{\boldsymbol{\xi}}$ and linear drift, namely,

$$\boldsymbol{\xi}_t = \widehat{\boldsymbol{\xi}}_t + \mu_\varepsilon t,$$

where $\widehat{\boldsymbol{\xi}}$ is a zero mean Lévymartingale with bounded jumps. Then we prove our results in two steps.

**Step 1.** We first estimate the value of $\mu_\varepsilon$. Since $\boldsymbol{\xi}$ is a Lévyprocess, by Lévy-Itô decomposition theory [13, Theorem 6.1] its characteristic function is of form

$$\mathbb{E}[e^{i\langle\boldsymbol{\lambda},\boldsymbol{\xi}_t\rangle}] = \exp\left(t\int_{\mathbb{R}^d\setminus\{\mathbf{0}\}}\left(e^{i\langle\boldsymbol{\lambda},\boldsymbol{y}\rangle} - 1 - i\langle\boldsymbol{\lambda},\boldsymbol{y}\rangle\mathbb{I}\{\|\boldsymbol{y}\| \le 1\}\right)\mathbb{I}\{\|\boldsymbol{y}\| \le \varepsilon^{-\delta}\}\,\mathrm{d}\boldsymbol{y}\right),$$

which can be further split into two Lévyprocesses $\boldsymbol{\xi}_{(1)}$ and $\boldsymbol{\xi}_{(2)}$ with characteristic functions

$$\mathbb{E}[e^{i\langle\boldsymbol{\lambda},\boldsymbol{\xi}_{(1),t}\rangle}] = \exp\left(t\int_{0<\|\boldsymbol{y}\|<1}\left(e^{i\langle\boldsymbol{\lambda},\boldsymbol{y}\rangle} - 1 - i\langle\boldsymbol{\lambda},\boldsymbol{y}\rangle\right)\mathrm{d}\boldsymbol{y}\right)$$

and

$$\mathbb{E}[e^{i\langle\boldsymbol{\lambda},\boldsymbol{\xi}_{(2),t}\rangle}] = \exp\left(t\int_{1\le\|\boldsymbol{y}\|\le\varepsilon^{-\delta}}\left(e^{i\langle\boldsymbol{\lambda},\boldsymbol{y}\rangle} - 1\right)\mathrm{d}\boldsymbol{y}\right).$$

Let we consider $\boldsymbol{\xi}$ on the set $\{\boldsymbol{y} \mid 0 < \|\boldsymbol{y}\| \le 1\}$. We construct a compensated compound Poisson process

$$L'_t = \sum_{s\le t}\Delta L'_s\mathbb{I}\{1 > \|\Delta L_s\| > \epsilon'\} - t\int_{1>\|\boldsymbol{y}\|>\epsilon'}\boldsymbol{y}\nu(\mathrm{d}\boldsymbol{y}) = \int_0^t\int_{1>\|\boldsymbol{y}\|>\epsilon'}\boldsymbol{y}\mu^L(\mathrm{d}\boldsymbol{y},\mathrm{d}s) - t\int_{1>\|\boldsymbol{y}\|>\epsilon'}\boldsymbol{y}\nu(\mathrm{d}\boldsymbol{y}),$$

where $\epsilon'$ is a very small constant. By applying Lemma 5 on $\sum_{s\le t}\Delta L'_s\mathbb{I}\{1 > \|\Delta L_s\| > \epsilon'\}$, the characteristic function of $L'_t$ is

$$\mathbb{E}[e^{i\langle\boldsymbol{\lambda},L'_t\rangle}] = \exp\left(t\int_{\epsilon'<\|\boldsymbol{y}\|<1}\left(e^{i\langle\boldsymbol{\lambda},\boldsymbol{y}\rangle} - 1 - i\langle\boldsymbol{\lambda},\boldsymbol{y}\rangle\right)\mathrm{d}\boldsymbol{y}\right).$$

This means that there exists a Lévyprocess $L'$ which is a square integral martingale such that $L' \to \boldsymbol{\xi}_{(1)}$ as $\epsilon' \to 0$. As $L'$ is a square integral martingale, we have $\mathbb{E}(\boldsymbol{\xi}_{(1)}) = \mathbb{E}(L') = \mathbf{0}$, which means that $\mu_\varepsilon$ is only related to $\boldsymbol{\xi}_{(2)}$. Therefore, we have

$$\mu_\varepsilon^i = \mathbb{E}[\boldsymbol{\xi}_{(2)}^i] = \int_{1\le\|\boldsymbol{y}\|\le\varepsilon^{-\delta}}\boldsymbol{y}_i\nu(\mathrm{d}\boldsymbol{y}), \quad (i = 1,\cdots,d)$$

$$\|\mu_\varepsilon\|^2 = \int_{1\le\|\boldsymbol{y}\|\le\varepsilon^{-\delta}}\|\boldsymbol{y}\|^2\nu(\mathrm{d}\boldsymbol{y}) = -\int_1^{\varepsilon^{-\delta}}u^2\mathrm{d}\Theta(u) = -u^2\Theta(u)\big|_1^{\varepsilon^{-\delta}} + 2\int_1^{\varepsilon^{-\delta}}u\Theta(u)\mathrm{d} \le \varepsilon^{-2\delta}\Theta(1).$$

Thus, we can bound $\|\mu_\varepsilon\| \le \varepsilon^{-\delta}\sqrt{\Theta(1)}$. Finally, by setting $\theta_0 = (1-\delta)/3$ and $\rho_0 = (1-\delta)/4$ we can obtain $\varepsilon\|\mu_\varepsilon\|T_\varepsilon = \varepsilon^{1-\delta-\theta}\sqrt{\Theta(1)} \le \varepsilon^{2\rho}$ by setting $\varepsilon$ sufficient small such that $\Theta(1) \le \frac{1}{\varepsilon^{1-2\rho-\delta-\theta}}$.

**Step 2.** Since the increment is non-negative, the quadratic variation process $[\varepsilon\widehat{\boldsymbol{\xi}}]_t^d$ is a Lévysubordinator, namely,

$$[\varepsilon\widehat{\boldsymbol{\xi}}]_t^d = \varepsilon^2\sum_{s\le t}\|\Delta\widehat{\boldsymbol{\xi}}_s\|^2 = \varepsilon^2\int_o^t\int_{0<\|\boldsymbol{y}\|\le\varepsilon^{-\delta}}\|\boldsymbol{y}\|^2 N(\mathrm{d}\boldsymbol{y},\mathrm{d}s),$$

where $\Delta\widehat{\boldsymbol{\xi}}_s = \widehat{\boldsymbol{\xi}}_s - \widehat{\boldsymbol{\xi}}_{s-}$ where $\widehat{\boldsymbol{\xi}}_{s-} = \lim_{t\uparrow s}\widehat{\boldsymbol{\xi}}_t$.

Since the jumps of $[\varepsilon\widehat{\boldsymbol{\xi}}]^d$ are bounded, its Laplace transform is well-defined for all $\lambda \in \mathbb{R}$:

$$\mathbb{E}e^{\lambda[\varepsilon\widehat{\boldsymbol{\xi}}]_t^d} = \exp\left(t\int_{0<\|\boldsymbol{y}\|\le\varepsilon^{-\delta}}(e^{\lambda\varepsilon^2\|\boldsymbol{y}\|^2} - 1)\nu(\mathrm{d}\boldsymbol{y})\right) = \exp\left(-t\int_{0<u\le\varepsilon^{-\delta}}(e^{\lambda\varepsilon^2 u^2} - 1)\mathrm{d}\Theta(u)\right).$$

For any $\lambda > 0$, the exponential Chebyshev inequality indicates

$$
\begin{aligned}
\mathbb{P}\left([\varepsilon\widehat{\boldsymbol{\xi}}]^d_{T_\varepsilon} > \varepsilon^\rho\right) &= \mathbb{P}\left(e^{\lambda[\varepsilon\widehat{\boldsymbol{\xi}}]^d_{T_\varepsilon}} > e^{\lambda\varepsilon^\rho}\right) \leq e^{-\lambda\varepsilon^\rho}\mathbb{E}[e^{\lambda[\varepsilon\widehat{\boldsymbol{\xi}}]^d_{T_\varepsilon}}] \\
&= \exp\left(-\lambda\varepsilon^\rho - T_\varepsilon \int_{0<u\leq\varepsilon^{-\delta}} (e^{\lambda\varepsilon^2 u^2} - 1)\mathsf{d}\Theta(u)\right).
\end{aligned}
\tag{27}
$$

For $\lambda = \lambda_\varepsilon = \varepsilon^{-2\rho}$ with $0 < \rho < \rho_0 = (1-\delta)/4$ we have $\max_{0\leq u\leq\varepsilon^{-\delta}} \lambda\varepsilon^2 u^2 \leq \lambda_\varepsilon\varepsilon^{2(1-\delta)} \leq \varepsilon^{\frac{3}{2}(1-\delta)} \downarrow 0$ as $\varepsilon \downarrow 0$. With help of the elementary inequality $e^x - 1 \leq 2x$ for small positive $x$ the second summand appearing in the exponent in right-hand side of (27) can be now established as

$$
\begin{aligned}
\left|T_\varepsilon \int_{0<u\leq\varepsilon^{-\delta}} (e^{\lambda_\varepsilon\varepsilon^2 u^2} - 1)\mathsf{d}\Theta(u)\right| &\leq \left|2T_\varepsilon\lambda_\varepsilon\varepsilon^2 \left(\int_{0<u\leq1} + \int_{1<u\leq\varepsilon^{-\delta}}\right) u^2\mathsf{d}\Theta(u)\right| \\
&\leq 2T_\varepsilon\lambda_\varepsilon\varepsilon^2\left|\int_{0<u\leq1} u^2\mathsf{d}\Theta(u)\right| + 2T_\varepsilon\lambda_\varepsilon\varepsilon^{2(1-\delta)}\left|\int_{1<u\leq\varepsilon^{-\delta}} \mathsf{d}\Theta(u)\right| \\
&\leq 2CT_\varepsilon\lambda_\varepsilon\epsilon^2 + 2\Theta(1)T_\varepsilon\lambda_\varepsilon\varepsilon^{2(1-\delta)}
\end{aligned}
$$

where $C = \left|\int_{0<u\leq1} u^2\mathsf{d}\Theta(u)\right| \in (0, +\infty)$ is a constant. Consequently, for all $0 < \rho \leq \rho_0$ and $0 < \theta < \theta_0$ we see that the exponential inequality

$$
\mathbb{P}\left([\varepsilon\widehat{\boldsymbol{\xi}}]^d_{T_\varepsilon} > \varepsilon^\rho\right) \leq \exp\left(-\lambda_\varepsilon\varepsilon^\rho + 2CT_\varepsilon\lambda_\varepsilon\varepsilon^2 + 2\Theta(1)T_\varepsilon\lambda_\varepsilon\varepsilon^{2(1-\delta)}\right) \leq \exp(-\varepsilon^{-\rho/2})
$$

holds for small enough $\varepsilon$ with $p \in (0, \rho/2)$. This is because

$$
-\lambda_\varepsilon\varepsilon^\rho + 2CT_\varepsilon\lambda_\varepsilon\varepsilon^2 + 2\Theta(1)T_\varepsilon\lambda_\varepsilon\varepsilon^{2(1-\delta)} = -\varepsilon^{-\rho} + 2C\varepsilon^{2-\frac{1-\delta}{3}-\frac{1-\delta}{2}} + 2\Theta(1)\varepsilon^{2(1-\delta)-\frac{1-\delta}{3}-\frac{1-\delta}{2}}
$$

$$
\leq -\varepsilon^{-\rho} + 2(C+\Theta(1))\varepsilon^{2(1-\delta)-\frac{1-\delta}{3}-\frac{1-\delta}{2}} \leq -\varepsilon^{-\rho} + 2(C+\Theta(1))\varepsilon^{\frac{7}{6}(1-\delta)} \overset{①}{\leq} -\varepsilon^{-\rho/2},
$$

where ① holds by setting $\varepsilon$ enough small such that $(\varepsilon^{-\rho} - 2(C+\Theta(1))\varepsilon^{\frac{7}{6}(1-\delta)})/\varepsilon^{-\rho/2} \geq \varepsilon^{-\rho/2} - 2(C+\Theta(1))\varepsilon^{\frac{7}{6}(1-\delta)+\frac{\rho}{2}} \geq 1$. The proof is completed. $\qquad\square$

### E.4  Proof of Lemma 2

*Proof.* **Step 1.** Suppose $\sup_{t\geq0}\|\boldsymbol{g}^t\| \leq c_g$ for some constant $c_g > 0$. Then we consider the one-dimensional martingale

$$
M_t = \sum_{i=1}^{d} \int_0^t \boldsymbol{g}^i_{s-}\mathsf{d}\widehat{\boldsymbol{\xi}}^i_s.
$$

We estimate the probability of a deviation of the size $\varepsilon^\rho$ of $\varepsilon M_t$ from zero with help of the exponential inequality for martingales, see Theorem 26.17 (i) in [14]. Indeed for any $\rho > 0$ and $\theta > 0$, we have

$$
\mathbb{P}\left(\sup_{t\leq T_\varepsilon}|\varepsilon M_t| \geq \varepsilon^\rho\right) \leq \mathbb{P}\left(\sup_{t\leq T_\varepsilon}|\varepsilon M_t| \geq \varepsilon^\rho \,\big|\, [\varepsilon M]_{T_\varepsilon} \leq \varepsilon^{4\rho}\right) + \mathbb{P}\left([\varepsilon M]_{T_\varepsilon} > \varepsilon^{4\rho}\right).
$$

Inspecting the proofs of Lemma 26.19 and Theorem 26.17 (i) in [14] we get that for any $\lambda > 0$

$$
\mathbb{P}\left(\sup_{t\leq T_\varepsilon}|\varepsilon M_t| \geq \varepsilon^\rho \,\big|\, [\varepsilon M]_{T_\varepsilon} \leq \varepsilon^{4\rho}\right) \leq \exp\left(-\lambda\varepsilon^\rho + \lambda^2 h(\lambda c_g\varepsilon^{1-\delta})\varepsilon^{4\rho}\right),
$$

where $h(x) = -(x + \ln(1-x)_+)x^{-2}$. For any $0 < \rho < \rho_1 = (1-\delta)/2$ we set $\lambda = \lambda_\varepsilon = \varepsilon^{-2\rho}$ so that $h(\lambda_\varepsilon c_g\varepsilon^{1-\delta}) \to 1/2$ as $\varepsilon \to 0$ by using LHopital's rule. Hence we obtain the estimate

$$
\mathbb{P}\left(\sup_{t\leq T_\varepsilon}|\varepsilon M_t| \geq \varepsilon^\rho \,\big|\, [\varepsilon M]_{T_\varepsilon} \leq \varepsilon^{4\rho}\right) \leq \exp\left(-\varepsilon^{-\rho} + \frac{1}{2}\right) \overset{①}{\leq} \exp\left(-\varepsilon^{-\rho/2}\right) \leq \exp\left(-\varepsilon^{-p}\right),
$$

which holds for small enough $\varepsilon$ and $p \in (0, \rho/2]$. In ①, we set $\varepsilon$ enough small such that $0 < \varepsilon^{-\rho/2} - \varepsilon^{\rho/2} \leq 1$.

**Step 2.** Since $\|\boldsymbol{g}^t\| \leq c_g$ is well bounded, then there is a constant $c_1$ with

$$
[\varepsilon M]_t = \int_0^t \boldsymbol{g}^2_{s-}\mathsf{d}[\varepsilon\widehat{\boldsymbol{\xi}}]^d_s \leq c_1[\varepsilon\widehat{\boldsymbol{\xi}}]^d_t.
$$

Then we can use Lemma 1 to upper bound:

$$\mathbb{P}\left([\varepsilon M]_{T_\varepsilon} \geq \varepsilon^{4\rho}\right) \leq \mathbb{P}\left(c_1[\varepsilon\widehat{\boldsymbol{\xi}}]_t^d \geq \varepsilon^{4\rho}\right) \overset{①}{\leq} \exp\left(-p\right),$$

where ① uses $\rho < \rho_2 < \frac{\rho_0}{4}$ with $\rho_0 = \frac{1-\delta}{4}$ in Lemma 1 and sets $\varepsilon$ sufficient small such that $\varepsilon^{\rho_0 - 4\rho} \leq c_1$. This is because if $\varepsilon^{\rho_0} \leq \frac{\varepsilon^{4\rho}}{c_1}$, then it yields $\mathbb{P}\left([\varepsilon\widehat{\boldsymbol{\xi}}]_t^d \geq \varepsilon^{4\rho}/c_1\right) \leq \exp\left(-p\right)$ due to $\mathbb{P}\left([\varepsilon\widehat{\boldsymbol{\xi}}]_t^d \geq \varepsilon^{\rho_0}\right) \leq \exp\left(-p\right)$. So the result in this lemma holds with $\rho_0 = \min(\rho_0 = \frac{1-\delta}{4}, \rho_1, \rho_2) = \frac{1-\delta}{16}$, $p_0 = \min(p_0 = \frac{\rho}{2}, p_1) = \frac{\delta}{2}$. The parameters $\rho_0$ and $p_0$ in the operator $(\cdot)$ are from Lemma 1 as the results here is based on Lemma 1. Under this setting, we have

$$\mathbb{P}\left(\sup_{0 \leq t \leq T_\varepsilon} \varepsilon \left|\sum_{i=1}^{d} \int_0^t \boldsymbol{g}_{s-}^i \mathsf{d}\widetilde{\boldsymbol{\xi}}_s^i\right| \geq \varepsilon^\rho\right) \leq 2\exp\left(-\varepsilon^{-p}\right)$$

The proof is completed. □

### E.5  Proof of Lemma 3

*Proof.* **Step 1.** In this step we prove the sequence $\{\widehat{\boldsymbol{\theta}}_t\}$ produced by Eqn. (8) or (9) locates in a very small neighborhood of the optimum solution $\boldsymbol{\theta}^*$ of the local basin $\boldsymbol{\Omega}$ after a very small time interval. Since we assume the function is locally strongly convex, by using Theorems 3 and 4, we know that the sequence $\{\widehat{\boldsymbol{\theta}}_t\}$ produced by Eqn. (8) or (9) exponentially converges to the minimum $\boldsymbol{\theta}^*$ at the current local basin $\boldsymbol{\Omega}$. So for any initialization $\boldsymbol{\theta}_0 \in \boldsymbol{\Omega}$, we have

$$\|\widehat{\boldsymbol{\theta}}_t - \boldsymbol{\theta}^*\|_2^2 \leq c_1 \exp\left(-c_2 t\right),$$

where $c_1 = \frac{2\Delta}{\mu}$ and $c_2 = \frac{2\mu\tau}{\beta_1(v_{\max} + \epsilon) + \mu\tau}\left(\beta_1 - \frac{\beta_2}{4}\right)$ in ADAM, $c_1 = \frac{2\Delta}{\mu}$ and $c_2 = 2\mu$ in SGD. Therefore, for any initialization $\boldsymbol{\theta}_0 \in \boldsymbol{\Omega}$ and sufficient small $\varepsilon$, we can obtain

$$\|\widehat{\boldsymbol{\theta}}_t - \boldsymbol{\theta}^*\|_2^2 \leq \varepsilon^{\bar\rho} \text{ when } t \geq v_\varepsilon = \frac{1}{c_2}\ln\left(\frac{c_1}{\varepsilon^{\bar\rho}}\right).$$

**Step 2.** Here we prove that for the time $t \in [0, v_\varepsilon]$, the sequence $\{\boldsymbol{\theta}_t\}$ is always very close to the sequence $\{\widehat{\boldsymbol{\theta}}_t\}$ when they are with the same initialization $\boldsymbol{\theta}_0$ in the absence of the big jumps $J_k$ in the stochastic process $L$.

To begin with, according to the updating rule in SGD, we have

$$\|\boldsymbol{\theta}_{t \wedge v_\varepsilon \wedge \sigma_1 -} - \widehat{\boldsymbol{\theta}}_{t \wedge v_\varepsilon \wedge \sigma_1 -}\| = \left|\int_0^{t \wedge v_\varepsilon \wedge \sigma_1 -}\left(-\nabla \boldsymbol{F}(\boldsymbol{\theta}_s) + \nabla \boldsymbol{F}(\widehat{\boldsymbol{\theta}}_s)\right)\mathsf{d}s + \int_0^{t \wedge v_\varepsilon \wedge \sigma_1 -} \varepsilon\boldsymbol{\Sigma}_s \mathsf{d}L_s\right|$$

$$\overset{①}{\leq} \ell \int_0^{t \wedge v_\varepsilon \wedge \sigma_1 -} \|\boldsymbol{\theta}_s - \widehat{\boldsymbol{\theta}}_s\|\mathsf{d}s + \varepsilon \left\|\int_0^{t \wedge v_\varepsilon \wedge \sigma_1 -} \boldsymbol{\Sigma}_s \mathsf{d}L_s\right\|,$$

(28)

where in ①, $\boldsymbol{F}(\boldsymbol{\theta})$ is $\ell$-smooth, namely $\|\nabla \boldsymbol{F}(\boldsymbol{\theta}_1) - \nabla \boldsymbol{F}(\boldsymbol{\theta}_2)\| \leq \ell\|\boldsymbol{\theta}_1 - \boldsymbol{\theta}_2\|$ for any $\boldsymbol{\theta}_1$ and $\boldsymbol{\theta}_2$ in the local basin $\boldsymbol{\Omega}$.

Then we consider ADAM which needs more efforts. According to the dynamic system of ADAM, we can first establish

$$\boldsymbol{m}_t - \widehat{\boldsymbol{m}}_t = \int_0^t (\nabla \boldsymbol{F}(\boldsymbol{\theta}_s) - \nabla \boldsymbol{F}(\widehat{\boldsymbol{\theta}}_s))\mathsf{d}s - \int_0^t (\boldsymbol{m}_s - \widehat{\boldsymbol{m}}_s)\mathsf{d}s.$$

Therefore, with the assumption $\|\boldsymbol{m}_t - \widehat{\boldsymbol{m}}_t\| \leq \tau_m \|\int_0^t (\boldsymbol{m}_s - \widehat{\boldsymbol{m}}_s)\mathsf{d}s\|$, it yields

$$|1 - \tau_m| \cdot \left\|\int_0^t (\boldsymbol{m}_s - \widehat{\boldsymbol{m}}_s)\mathsf{d}s\right\| \leq \left\|\boldsymbol{m}_t - \widehat{\boldsymbol{m}}_t + \int_0^t (\boldsymbol{m}_s - \widehat{\boldsymbol{m}}_s)\mathsf{d}s\right\| = \left\|\int_0^t (\nabla \boldsymbol{F}(\boldsymbol{\theta}_s) - \nabla \boldsymbol{F}(\widehat{\boldsymbol{\theta}}_s))\mathsf{d}s\right\|$$

$$\leq \ell \int_0^t \left\|\boldsymbol{\theta}_s - \widehat{\boldsymbol{\theta}}_s\right\|\mathsf{d}s.$$

Moreover, we can upper bound $\frac{\mu_s}{\sqrt{\omega_s \boldsymbol{v}_s}+\epsilon} = \frac{\sqrt{1-e^{-\beta_2 t}}}{1-e^{-\beta_1 t}} \cdot \frac{1}{1+\varepsilon\sqrt{1-e^{-\beta_2 t}}}$. Then let $q(x) = \frac{\sqrt{1-e^{-\beta_2 t}}}{1-e^{-\beta_1 t}} \leq c_4 = \min(q(0), q(+\infty), q(t^*))$, where $t^*$ is a time such that $q'(t^*) = 0$. Since $q(0) = \frac{\beta_2}{2\beta_1}$ by LHopital's rule, $q(+\infty) = 1$ and $q(t^*) < \infty$ is a constant, $c_4 < \infty$ is a constant. So there exists a constant $c_5$ such that $\frac{\mu_s}{\sqrt{\omega_s \boldsymbol{v}_s}+\epsilon} \leq \frac{c_5}{v_{\min}+\epsilon}$. Then similarly, in ADAM, we also can establish

$$\|\boldsymbol{\theta}_{t\wedge v_\varepsilon \wedge \sigma_1 -} - \widehat{\boldsymbol{\theta}}_{t\wedge v_\varepsilon \wedge \sigma_1 -}\| = \left|\int_0^{t\wedge v_\varepsilon \wedge \sigma_1 -} \left(-\frac{\mu_s \boldsymbol{m}_s}{\sqrt{\omega_s \boldsymbol{v}_s}+\epsilon} + \frac{\mu_s \widehat{\boldsymbol{m}}_s}{\sqrt{\omega_s \widehat{\boldsymbol{v}}_s}+\epsilon}\right) \mathrm{d}s + \int_0^{t\wedge v_\varepsilon \wedge \sigma_1 -} \varepsilon \boldsymbol{Q}_s^{-1}\boldsymbol{\Sigma}_s \mathrm{d}L_s\right|$$

$$\overset{\text{①}}{\leq} \frac{c_5 \ell}{(v_{\min}+\epsilon)|\tau_m - 1|}\int_0^{t\wedge v_\varepsilon \wedge \sigma_1 -}\|\boldsymbol{\theta}_s - \widehat{\boldsymbol{\theta}}_s\|\mathrm{d}s + \varepsilon\left\|\int_0^{t\wedge v_\varepsilon \wedge \sigma_1 -} \boldsymbol{Q}_s^{-1}\boldsymbol{\Sigma}_s \mathrm{d}L_s\right\|.$$

Next, we can employ Gronwall's to estimate

$$\sup_{0\leq t\leq \sigma_1 \wedge v_\varepsilon} \|\boldsymbol{\theta}_t - \widehat{\boldsymbol{\theta}}_t\| \leq \exp(\kappa_1 v_\varepsilon)\sup_{0\leq t\leq v_\varepsilon} \varepsilon\left\|\int_0^t \boldsymbol{Q}_s^{-1}\boldsymbol{\Sigma}_s \mathrm{d}\boldsymbol{\xi}_s\right\|,$$

where $\kappa_1 = \ell$ in SGD, and $\kappa_1 = \frac{c_5 \ell}{(v_{\min}+\epsilon)|\tau_m - 1|}$ in ADAM. Since when $\varepsilon$ is small enough, $v_\varepsilon = \frac{1}{c_2}\ln\left(\frac{c_1}{\varepsilon^{\bar{\rho}}}\right)$ is much smaller than $T_\varepsilon = \varepsilon^{-\theta}$ when $\varepsilon$ is sufficient small. It yields

$$\mathbb{P}\left(\sup_{0\leq t\leq \sigma_1 \wedge v_\varepsilon}\|\boldsymbol{\theta}_t - \widehat{\boldsymbol{\theta}}_t\| \geq \varepsilon^{\bar{\rho}}\right) \leq \mathbb{P}\left(\exp(\kappa_1 v_\varepsilon)\sup_{0\leq t\leq v_\varepsilon} \varepsilon\left\|\int_0^t \boldsymbol{Q}_s^{-1}\boldsymbol{\Sigma}_s \mathrm{d}\boldsymbol{\xi}_s\right\| \geq \varepsilon^{\bar{\rho}}\right)$$

$$\overset{\text{①}}{\leq}\mathbb{P}\left(\sup_{0\leq t\leq v_\varepsilon} \varepsilon\left\|\int_0^t \boldsymbol{Q}_s^{-1}\boldsymbol{\Sigma}_s \mathrm{d}\widehat{\boldsymbol{\xi}}_s\right\| + \varepsilon\|\mu_\varepsilon\|T_\varepsilon \geq \varepsilon^{\bar{\rho}+c_3\kappa_1\bar{\rho}}\right)$$

$$=\mathbb{P}\left(\sup_{0\leq t\leq v_\varepsilon} \varepsilon\left\|\int_0^t \boldsymbol{Q}_s^{-1}\boldsymbol{\Sigma}_s \mathrm{d}\widehat{\boldsymbol{\xi}}_s\right\| \geq \varepsilon^\rho(\varepsilon^{\bar{\rho}(1+c_3\kappa_1)-\rho} - \varepsilon^\rho)\right)$$

$$\overset{\text{②}}{\leq} \exp(-p),$$

where ① uses Lemma 1: (1) the process $\boldsymbol{\xi}$ can be decomposed into two processes $\widehat{\boldsymbol{\xi}}$ and linear drift, namely, $\boldsymbol{\xi}_t = \widehat{\boldsymbol{\xi}}_t + \mu_\varepsilon t$, where $\widehat{\boldsymbol{\xi}}$ is a zero mean Lévymartingale with bounded jumps; (2) $\|\varepsilon\boldsymbol{\xi}_{T_\varepsilon}\| = \varepsilon\|\mu_\varepsilon\|T_\varepsilon < \varepsilon^{2\rho}$. In ②, (1) we set $\bar{\rho}(1+c_3\kappa_1) < \rho$ and also set $\varepsilon$ sufficient small such that $\varepsilon^{\bar{\rho}(1+c_3\kappa_1)-\rho} - \varepsilon^\rho \geq 1$; (2) by assume $\rho_0 = \rho_0(\delta) = \frac{1-\delta}{16} > 0$, $\theta_0 = \theta_0(\delta) = \frac{1-\delta}{3} > 0$ and $p_0 = p_0(\rho) = \frac{\rho}{2}$, we use Lemma 2 by setting $\boldsymbol{g}^t = \boldsymbol{Q}_t^{-1}\boldsymbol{\Sigma}_t$ and obtain $\mathbb{P}\left(\sup_{0\leq t\leq T_\varepsilon} \varepsilon\left|\sum_{i=1}^d \int_0^t \boldsymbol{g}_{s-}^i \mathrm{d}\widehat{\boldsymbol{\xi}}_s^i\right| \geq \varepsilon^\rho\right) \leq 2\exp(-\varepsilon^{-p})$ for all $p \in (0, p_0]$ and $0 < \varepsilon \leq \varepsilon_0$ with $\varepsilon_0 = \varepsilon_0(\rho)$.

**Step 3.** In the first step, we have analyzed that the sequence $\{\widehat{\boldsymbol{\theta}}_t\}$ will converge to the optimum $\boldsymbol{\theta}^*$ of the basin $\boldsymbol{\Omega}$. Moreover, in the second step, we prove that $\boldsymbol{\theta}_t$ is very close to $\widehat{\boldsymbol{\theta}}_t$. In this step, we show that in absence of the big jumps of the driving process $L$ the sequence $\boldsymbol{\theta}_t$ is close to $\boldsymbol{\theta}^*$. For brevity, we set $\boldsymbol{\theta}^* = \boldsymbol{0}$. Then we define a function $h(\boldsymbol{\theta}) = \ln(1+\boldsymbol{F}(\boldsymbol{\theta})) \geq 0$. Since for a small local convex basin $\boldsymbol{\Omega}$, the function $\boldsymbol{F}(\boldsymbol{\theta})$ can be well approximated by a quadratic function. In this way, for small $\boldsymbol{\theta}$ one can always estimate $c_6\|\boldsymbol{\theta}\|^2 \leq h(\boldsymbol{\theta}) \leq c_7\|\boldsymbol{\theta}\|^2$ for some positive constants $c_6$ and $c_7$. Furthermore, the derivatives $\partial_i h(\boldsymbol{\theta}) = \frac{\partial_i \boldsymbol{F}(\boldsymbol{\theta})}{1+\boldsymbol{F}(\boldsymbol{\theta})}$ and $\partial_i\partial_j h(\boldsymbol{\theta}) = \frac{\partial_{ij}\boldsymbol{F}(\boldsymbol{\theta})(1+\boldsymbol{F}(\boldsymbol{\theta}))-\partial_i \boldsymbol{F}(\boldsymbol{\theta})\partial_j \boldsymbol{F}(\boldsymbol{\theta})}{(1+\boldsymbol{F}(\boldsymbol{\theta}))^2}$ are bounded since the assumptions on the function $\boldsymbol{F}(\boldsymbol{\theta})$, namely $\boldsymbol{F}(\boldsymbol{\theta})$ being upper bounded, $\ell$-smooth. Next we can apply the Itô formulation to the process $h(\boldsymbol{\theta}_t)$:

$$0 \leq h(\boldsymbol{\theta}_{t\wedge T_\varepsilon \wedge \sigma_1 -}) = h(\boldsymbol{\theta}) + \sum_{i=1}^d \int_0^{t\wedge T_\varepsilon \wedge \sigma_1 -} \partial_i h(\boldsymbol{\theta}_{s-})\mathrm{d}\boldsymbol{\theta}_{s-}^i + \frac{1}{2}\sum_{i,j=1}^d \int_0^{t\wedge T_\varepsilon \wedge \sigma_1 -} \partial_i\partial_j h(\boldsymbol{\theta}_{s-})\mathrm{d}[\boldsymbol{\theta}^i, \boldsymbol{\theta}^j]_s^c$$

$$+ \sum_{s<t\wedge T_\varepsilon \wedge \sigma_1}\left(h(\boldsymbol{\theta}_s) - h(\boldsymbol{\theta}_{s-}) - \sum_{i=1}^d \partial_i h(\boldsymbol{\theta}_{s-})\Delta\boldsymbol{\theta}_s^i\right)$$

$$\overset{\text{①}}{\leq} h(\boldsymbol{\theta}) - \int_0^{t\wedge T_\varepsilon \wedge \sigma_1 -}\left\langle\frac{\nabla\boldsymbol{F}(\boldsymbol{\theta}_{s-})}{1+\boldsymbol{F}(\boldsymbol{\theta}_{s-})}, \frac{\mu_t \boldsymbol{m}_{s-}}{\epsilon+\sqrt{\omega_{s-}\boldsymbol{v}_{s-}}}\right\rangle\mathrm{d}s + \int_0^{t\wedge T_\varepsilon \wedge \sigma_1 -}\frac{\varepsilon(\nabla\boldsymbol{F}(\boldsymbol{\theta}_{s-}))^T\boldsymbol{Q}_{s-}^{-1}\boldsymbol{\Sigma}_{s-}}{1+\boldsymbol{F}(\boldsymbol{\theta}_{s-})}\mathrm{d}L_s$$

$$+ \sum_{s<t\wedge T_\varepsilon \wedge \sigma_1}\left(h(\boldsymbol{\theta}_s) - h(\boldsymbol{\theta}_{s-}) - \sum_{i=1}^d \partial_i h(\boldsymbol{\theta}_{s-})\Delta\boldsymbol{\theta}_s^i\right),$$

where ① uses $\mathrm{d}\boldsymbol{\theta}_s = -\frac{\mu_s \boldsymbol{m}_s}{\epsilon + \sqrt{\omega_s \boldsymbol{v}_s}} + \varepsilon \boldsymbol{Q}_s^{-1} \boldsymbol{\Sigma}_s \mathrm{d}L_s$ and the path-by-path continuous part $[\boldsymbol{\theta}^i, \boldsymbol{\theta}^j]_s^c = 0$ of the quadratic covariation of $\boldsymbol{\theta}^i$ and $\boldsymbol{\theta}^j$. Since in Adam by assumption $\int_0^{t \wedge T_\varepsilon \wedge \sigma_1 -} \left\langle \frac{\nabla \boldsymbol{F}(\boldsymbol{\theta}_{s-})}{1 + \boldsymbol{F}(\boldsymbol{\theta}_{s-})}, \frac{\mu_t \boldsymbol{m}_{s-}}{\epsilon + \sqrt{\omega_{s-} \boldsymbol{v}_{s-}}} \right\rangle \mathrm{d}s \geq 0$, the second term is non-negative due to $\boldsymbol{F}(\boldsymbol{\theta}) \geq 0$. Note in SGD, $\boldsymbol{m}_s = \nabla \boldsymbol{F}(\boldsymbol{\theta}_s)$. So in SGD we do not make the assumption $\langle \nabla \boldsymbol{F}(\boldsymbol{\theta}_t), \boldsymbol{m}_t \rangle \geq 0$. In SGD, $\epsilon + \sqrt{\omega_{s-} \boldsymbol{v}_{s-}}$ equals to one. In this way, we can estimate the last term as

$$\sum_{s < t \wedge T_\varepsilon \wedge \sigma_1} \left| h(\boldsymbol{\theta}_s) - h(\boldsymbol{\theta}_{s-}) - \sum_{i=1}^d \partial_i h(\boldsymbol{\theta}_{s-}) \Delta \boldsymbol{\theta}_s^i \right|$$

$$\leq \frac{1}{2} \sum_{i,j=1}^d \sum_{s < t \wedge T_\varepsilon \wedge \sigma_1} \left| \int_0^1 (1-v) \partial_i \partial_j h(\boldsymbol{\theta}_{s-} + v\Delta \boldsymbol{\theta}_s) \mathrm{d}v \right| \cdot |\Delta \boldsymbol{\theta}_s^i \Delta \boldsymbol{\theta}_s^j| \leq c_8 \sum_{s \leq t} \|\Delta \boldsymbol{\theta}_s\|^2 = c_8 [\boldsymbol{\theta}]_t^d,$$

holds with some $c_8 > 0$. Furthermore, since $\boldsymbol{v}_t$ and $\boldsymbol{\Sigma}_t$ are assumed to be bounded, then we can upper bound $[\boldsymbol{\theta}]_t^d$ as follows:

$$[\boldsymbol{\theta}]_t^d \leq c_9 [\varepsilon L]_t^d \overset{①}{=} c_9 [\varepsilon \boldsymbol{\xi}]_t^d$$

hold for some constant $c_9$ for all $t \leq \sigma_1$. ① holds since we assume there is no big jump during $t \leq \sigma_1$. Then by combining all the results and letting $\boldsymbol{g}^s = \frac{(\nabla \boldsymbol{F}(\boldsymbol{\theta}_s))^T \boldsymbol{Q}_s^{-1} \boldsymbol{\Sigma}_s}{1 + \boldsymbol{F}(\boldsymbol{\theta}_s)}$ and considering $\boldsymbol{F}(\boldsymbol{\theta}) \leq c_7 \|\boldsymbol{\theta}\|$, we can obtain the following results when $\|\boldsymbol{\theta}\| = \|\boldsymbol{\theta}_0\| \leq \varepsilon^{\bar{\rho}}$ with enough small $\varepsilon$:

$$0 \leq \|\boldsymbol{\theta}_{t \wedge T_\varepsilon \wedge \sigma_1 -}\|^2 \leq \frac{1}{c_6} h(\boldsymbol{\theta}_{t \wedge T_\varepsilon \wedge \sigma_1 -}) \leq c_{10} \left( \varepsilon^{2\bar{\rho}} + \varepsilon \sup_{0 \leq t \leq T_\varepsilon} \left| \int_0^t \boldsymbol{g}^{s-} \mathrm{d}\widehat{\boldsymbol{\xi}}_s \right| + \varepsilon \|\mu_\varepsilon\| T_\varepsilon + \varepsilon^2 [\boldsymbol{\xi}]_{T_\varepsilon}^d \right).$$

where $c_{10}$ is a certain constant. Combining the above results gives

$$\mathbb{P} \left( \sup_{0 \leq t \leq T_\varepsilon \wedge \sigma_1} \|\boldsymbol{\theta}_t\| \geq \varepsilon^{\bar{\rho}} \right) \leq \mathbb{P} \left( \varepsilon^{2\bar{\rho}} \geq \frac{\varepsilon^{\bar{\rho}}}{4c_{10}} \right) + \mathbb{P} \left( \varepsilon \sup_{0 \leq t \leq T_\varepsilon} \left| \int_0^t \boldsymbol{g}^{s-} \mathrm{d}\widehat{\boldsymbol{\xi}}_s \right| \geq \frac{\varepsilon^{\bar{\rho}}}{4c_{10}} \right)$$

$$+ \mathbb{P} \left( \varepsilon \|\mu_\varepsilon\| T_\varepsilon \geq \frac{\varepsilon^{\bar{\rho}}}{4c_{10}} \right) + \mathbb{P} \left( \varepsilon^2 [\boldsymbol{\xi}]_{T_\varepsilon}^d \geq \frac{\varepsilon^{\bar{\rho}}}{4c_{10}} \right).$$

Then by setting $\bar{\rho} < \rho$ and sufficient small $\varepsilon$ such that $\frac{\varepsilon^{\bar{\rho} - \rho}}{4c_{10}} \geq 1$ giving $\frac{\varepsilon^{\bar{\rho}}}{4c_{10}} \geq \varepsilon^\rho$. Then let the results in Lemma 1 and 2 hold simultaneously by setting $\rho_0 = \rho_0(\delta) = \frac{1-\delta}{16} > 0$, $\theta_0 = \theta_0(\delta) = \frac{1-\delta}{3} > 0$, $p_0 = \frac{\rho}{2}$, and small enough $\epsilon$, we have $\|\varepsilon \boldsymbol{\xi}_{T_\varepsilon}\| = \varepsilon \|\mu_\varepsilon\| T_\varepsilon < \varepsilon^{2\rho}$ and $\mathbb{P}\left( [\varepsilon \boldsymbol{\xi}]_{T_\varepsilon}^d \geq \varepsilon^\rho \right) \leq \exp(-\varepsilon^{-p})$ in Lemma 1, and $\mathbb{P}\left( \sup_{0 \leq t \leq T_\varepsilon} \epsilon \left| \sum_{i=1}^d \int_0^t \boldsymbol{g}_{s-}^i \mathrm{d}\widehat{\boldsymbol{\xi}}_s^i \right| \geq \varepsilon^\rho \right) \leq 2 \exp\left(-\varepsilon^{-p}\right)$ in Lemma 2. By using these results, we have

$$\mathbb{P} \left( \sup_{0 \leq t \leq T_\varepsilon \wedge \sigma_1} \|\boldsymbol{\theta}_t\| \geq \varepsilon^{\bar{\rho}} \right) \leq 4 \exp(-\varepsilon^{-p}).$$

for all $p \in (0, p_0]$ and $0 < \varepsilon \leq \varepsilon_0$ with $\epsilon_0 = \varepsilon_0(\rho)$.

**Step 4.** In Steps 1 and 2, we guarantee $\mathbb{P}\left( \sup_{0 \leq t \leq v_\varepsilon \wedge \sigma_1} \|\boldsymbol{\theta}_t - \widehat{\boldsymbol{\theta}}_t\| \geq \varepsilon^{\bar{\rho}} \right) \leq 4 \exp(-\varepsilon^{-p})$. Then after $v_\varepsilon$ time, we have $\|\boldsymbol{\theta}_t\| \leq \varepsilon^{\bar{\rho}}$ for all $t \geq v_\varepsilon$. In this way, the result in Step 4 holds. So in this step, we combine the results in Steps 1, 2 and 3 and extend the initialization in Step 3 to all possible parameter in $\boldsymbol{\theta}_0 \in \boldsymbol{\Omega}$:

$$\mathbb{P} \left( \sup_{0 \leq t \leq v_\varepsilon \wedge \sigma_1} \|\boldsymbol{\theta}_t - \widehat{\boldsymbol{\theta}}_t\| \geq \varepsilon^{\bar{\rho}} \right) \leq 4 \exp(-\varepsilon^{-p}),$$

for all $p \in (0, p_0]$ and $0 < \varepsilon \leq \varepsilon_0$ with $\varepsilon_0 = \varepsilon_0(\rho)$ by setting $\rho_0 = \rho_0(\delta) = \frac{1-\delta}{16} > 0$, $\theta_0 = \theta_0(\delta) = \frac{1-\delta}{3} > 0$, $p_0 = \frac{\rho}{2}$, $\bar{\rho}(1 + c_3 \kappa_1) < \rho$ and small enough $\varepsilon$. Note here we can remove the extra factor $\rho$ by setting $\varepsilon_0 = \varepsilon_0(\bar{\rho})$, $\rho_0 = \rho_0(\delta) = \frac{1-\delta}{16(1 + c_3 \kappa_1)} > 0$, $\theta_0 = \theta_0(\delta) = \frac{1-\delta}{3} > 0$, $p_0 = \frac{\bar{\rho}(1 + c_3 \kappa_1)}{2}$, $p \in (0, p_0)$.

**Step 5.** In this step, we extend the result in Step 4 from the time interval $[0, T_\varepsilon \wedge \sigma_1)$ to the time interval $[0, \sigma_1)$.

Let $\boldsymbol{\theta}_t^{\boldsymbol{\xi}}$ denote the sequence produced by SGD (4) or Adam (5) driven by the process $\boldsymbol{\xi}$. Then it is easy to check that for any $t < \sigma_1$, we have $\boldsymbol{\theta}_t^{\boldsymbol{\xi}} = \boldsymbol{\theta}_t$, since there are no big jumps in $\boldsymbol{\theta}_t$. Then consider any $\boldsymbol{\theta}_0 \in \boldsymbol{\Omega}$ and $k \geq 1$, we have for any $\bar{\rho} > 0$ and $\theta > 0$

$$\mathbb{P}\left(\sup_{0 \leq t < \sigma_1} \|\boldsymbol{\theta}_t - \widehat{\boldsymbol{\theta}}_t\| \geq \varepsilon^{\bar{\rho}}\right)$$

$$\leq \mathbb{P}\left(\sup_{0 \leq t < kT_\varepsilon \wedge \sigma_1} \|\boldsymbol{\theta}_t - \widehat{\boldsymbol{\theta}}_t\| \geq \varepsilon^{\bar{\rho}} \mid kT_\varepsilon < \sigma_1\right) + \mathbb{P}\left(\sup_{0 \leq t < \sigma_1} \|\boldsymbol{\theta}_t - \widehat{\boldsymbol{\theta}}_t\| \geq \varepsilon^{\bar{\rho}} \mid kT_\varepsilon \geq \sigma_1\right)$$

$$\leq \mathbb{P}\left(\sup_{0 \leq t < kT_\varepsilon \wedge \sigma_1} \|\boldsymbol{\theta}_t - \widehat{\boldsymbol{\theta}}_t\| \geq \varepsilon^{\bar{\rho}}\right) + \mathbb{P}\left(kT_\varepsilon \geq \sigma_1\right)$$

$$\leq \mathbb{P}\left(\sup_{0 \leq t < kT_\varepsilon} \|\boldsymbol{\theta}_t - \widehat{\boldsymbol{\theta}}_t\| \geq \varepsilon^{\bar{\rho}}\right) + \mathbb{P}\left(kT_\varepsilon \geq \sigma_1\right).$$

Besides, by using the linear convergence results of $\widehat{\boldsymbol{\theta}}_t$ to the optimum solution $\boldsymbol{\theta}^* = \mathbf{0}$ in the local basin $\boldsymbol{\Omega}$, for enough small $\varepsilon$ we have $\|\widehat{\boldsymbol{\theta}}_{T_\varepsilon}\| \leq \varepsilon^{2\bar{\rho}}$ with initialization $\boldsymbol{\theta}_0 \in \boldsymbol{\Omega}$. Then we let $\widehat{\boldsymbol{\theta}}_t(\boldsymbol{\theta})$ denote the sequence $\widehat{\boldsymbol{\theta}}_t$ but with initialization $\boldsymbol{\theta}$ and define

$$\boldsymbol{E}_i = \left\{\sup_{t \in [iT_\varepsilon, (i+1)T_\varepsilon]} \|\boldsymbol{\theta}_t^{\boldsymbol{\xi}} - \widehat{\boldsymbol{\theta}}_{t-iT_\varepsilon}(\boldsymbol{\theta}_{iT_\varepsilon}^{\boldsymbol{\xi}})\| < \varepsilon^{\bar{\rho}}\right\}, \quad 0 \leq i \leq k-1.$$

Note that the probability of $\boldsymbol{E}_0^c = \left\{\sup_{t \in [0, T_\varepsilon]} \|\boldsymbol{\theta}_t^{\boldsymbol{\xi}} - \widehat{\boldsymbol{\theta}}_t(\boldsymbol{\theta}_0^{\boldsymbol{\xi}})\| \geq \varepsilon^{\bar{\rho}}\right\}$ is given in Step 4 where $\boldsymbol{\theta}_0^{\boldsymbol{\xi}} = \boldsymbol{\theta}_0$. Furthermore for any $k \geq 1$, we have

$$\bigcap_{i=0}^{k-1} \boldsymbol{E}_i \subseteq \left\{\sup_{t \in [0, kT_\varepsilon]} \|\boldsymbol{\theta}_t^{\boldsymbol{\xi}} - \widehat{\boldsymbol{\theta}}_t\| < 2\varepsilon^{\bar{\rho}}\right\}.$$

As a result, we can obtain

$$\mathbb{P}\left(\sup_{t \in [0, kT_\varepsilon]} \|\boldsymbol{\theta}_t^{\boldsymbol{\xi}} - \widehat{\boldsymbol{\theta}}_t\| \geq 2\varepsilon^{\bar{\rho}}\right) \leq \mathbb{P}\left(\bigcup_{i=0}^{k-1} \boldsymbol{E}_i^c\right) = \mathbb{P}\left(\boldsymbol{E}_0^c \bigcup (\boldsymbol{E}_0 \boldsymbol{E}_1^c) \bigcup (\boldsymbol{E}_0 \boldsymbol{E}_1 \boldsymbol{E}_2^c) \bigcup \cdots \left(\bigcup_{i=0}^{k-2} \boldsymbol{E}_i \boldsymbol{E}_{k-1}^c\right)\right)$$

$$\leq \sum_{i=0}^{k-1} \mathbb{P}\left(\boldsymbol{E}_i^c, \boldsymbol{\theta}_{iT_\varepsilon}^{\boldsymbol{\xi}} \in \boldsymbol{\Omega}\right) \leq k \sup_{\boldsymbol{\theta}_0 \in \boldsymbol{\Omega}} \mathbb{P}\left(\boldsymbol{E}_0^c\right).$$

For $k = k_\varepsilon = \varepsilon^{-2r}$ and any $\theta > 0$ we have

$$\mathbb{P}\left(\sigma_1 \geq k_\varepsilon T_\varepsilon\right) = \exp(-k_\varepsilon T_\varepsilon \Theta(\epsilon^{-\delta})) \leq \exp(-\varepsilon^{r\delta - \theta - 2r} \Theta(\varepsilon^{-\delta})) \leq \exp(-\varepsilon^{-p})$$

for all $0 < p \leq (2 - \delta)r$ with enough small $\varepsilon$. On the other hand, we have

$$\mathbb{P}\left(\sup_{t \in [0, kT_\varepsilon]} \|\boldsymbol{\theta}_t^{\boldsymbol{\xi}} - \widehat{\boldsymbol{\theta}}_t\| \geq 2\varepsilon^{\bar{\rho}}\right) \leq k \sup_{\boldsymbol{\theta}_0 \in \boldsymbol{\Omega}} \mathbb{P}\left(\boldsymbol{E}_0^c\right) \leq \varepsilon^{-2r} \exp(-\varepsilon^{-p}) \leq \exp(-\varepsilon^{-p/2})$$

for any $p \leq \frac{2 \log(r \log(\varepsilon)))}{\log(\varepsilon)}$. Therefore, the result in this lemma holds

$$\mathbb{P}\left(\sup_{t \in [0, \sigma_1)} \|\boldsymbol{\theta}_t^{\boldsymbol{\xi}} - \widehat{\boldsymbol{\theta}}_t\| \geq 2\varepsilon^{\bar{\rho}}\right)$$

$$= \mathbb{P}\left(\sup_{t \in [0, \sigma_1)} \|\boldsymbol{\theta}_t^{\boldsymbol{\xi}} - \widehat{\boldsymbol{\theta}}_t\| \geq 2\varepsilon^{\bar{\rho}}, \sigma_1 < kT_\varepsilon\right) + \mathbb{P}\left(\sup_{t \in [0, \sigma_1)} \|\boldsymbol{\theta}_t^{\boldsymbol{\xi}} - \widehat{\boldsymbol{\theta}}_t\| \geq 2\varepsilon^{\bar{\rho}}, \sigma_1 \geq kT_\varepsilon\right)$$

$$\leq \mathbb{P}\left(\sup_{t \in [0, kT_\varepsilon]} \|\boldsymbol{\theta}_t^{\boldsymbol{\xi}} - \widehat{\boldsymbol{\theta}}_t\| \geq 2\varepsilon^{\bar{\rho}}\right) + \mathbb{P}\left(\sigma_1 \geq kT_\varepsilon\right) \leq 2 \exp(-\varepsilon^{-p/2}).$$

for all $p \in (0, p_0]$ and $0 < \varepsilon \leq \varepsilon_0$ with $\varepsilon_0 = \varepsilon_0(\bar{\rho})$ by setting $\rho_0 = \rho_0(\delta) = \frac{1-\delta}{16(1+c_3\kappa_1)} > 0$, $\theta_0 = \theta_0(\delta) = \frac{1-\delta}{3} > 0$, $p_0 = \min(\frac{\bar{\rho}(1+c_3\kappa_1)}{2}, p)$ with $p > 0$ and small enough $\varepsilon$. Besides, we also require $v_\varepsilon = \frac{1}{c_2} \ln\left(\frac{c_1}{\varepsilon^{\bar{\rho}}}\right) = \frac{1}{c_2} \ln\left(\frac{2\Delta}{\mu\varepsilon^{\bar{\rho}}}\right) \leq \varepsilon^{-\theta_0}$ where $c_1 = \frac{2\Delta}{\mu}$ and $c_2 = \frac{2\mu\tau}{\beta_1(v_{\max}+\epsilon)+\mu\tau}\left(\beta_1 - \frac{\beta_2}{4}\right)$ in ADAM, $c_1 = \frac{2\Delta}{\mu}$ and $c_2 = 2\mu$ in SGD. That is, The proof is completed. $\qquad \square$