[Reviews · NeurIPS 2020]

Review 1

Summary and Contributions: This paper analyzes escaping from local minima of SGD and Adam, using their corresponding Levy-driven stochastic differential equations (SDEs). By defining the “flatness” and “sharpness” or local minima based on a Radon measure, the paper theoretically shows that: (1) both SGD and Adam prefer to find flat local minima, i.e., flat minima usually have larger escaping time than sharp ones, (2) Compared to Adam, SGD tends to easily escape from local basins.

Strengths: The theory established in the paper is very solid, and provides a good explanation for the generalization performances of SGD and Adam, and the gap between them. Note that there is always a debate on the connection between generalization performance and flatness of local minima. This proposed definition of flatness based on Radon measure seems can make the debate to an end.

Weaknesses: The experiment mainly verifies the generalization performance gap between SGD and Adam, but does not verify the main theory established in the paper, such as: the relation between Radon measure and the escaping time, the validity of the flatness definition.

Correctness: As far as I checked, I think the claims are correct.

Clarity: The paper is written clearly and easy to read.

Relation to Prior Work: Related works are properly mentioned.

Reproducibility: Yes

Additional Feedback: [After author feedback] The feedback addresses some of my concerns. Hence, I keep my score.


Review 2

Summary and Contributions: This work analyzes the convergence behaviors of SGD and Adam through their Levy-driven stochastic differential equations. The theoretical results show that escaping time from a basin for SGD is smaller than that of Adam. Moreover, comparing to Adam, SGD is more like to finally converge to a flat or asymmetric basin that often have better generalization performance.

Strengths: This work can help us to theoretically better understand why SGD generalizes better than Adam in deep learning.

Weaknesses: 1). Commonly, the optimization algorithm “SGD” in deep learning is SGD with momentum rather than the standard SGD. This work does not discuss SGD with momentum. Compared to the SDE of the standard SGD, the SDE of SGD with momentum is more like that of Adam. Hence, there is a question whether the proposed theory can still interpret that SGD with momentum can generalize better than Adam. 2). One of the contributions is substituting the Gaussian distribution assumption for gradient noise in SGD and Adam with the alpha-stable distribution, but this work does not state how to get the key tail index \alpha. From Figure 1, we can conclude that alpha in practice is usually smaller than 1. This will bring a new problem. According to Theorem 1, the first escaping time \Gamma negatively depends on the learning rate \eta. It means a larger learning rate is beneficial to escaping from a basin. However, in practice it is common that a smaller learning rate is more helpful to escape from a basin. Hence, this theory is somewhat inconsistent with the phenomenon in practice.

Correctness: The theory seems to be inconsistent with the phenomenon in practice.

Clarity: Yes

Relation to Prior Work: Yes

Reproducibility: Yes

Additional Feedback: The responses partly address my concerns, so I raise my score. But the theoretical analysis of SGD-M is still not sufficient, I would like to see more clear justifications for SGD-M in the updated revision.


Review 3

Summary and Contributions: This paper studies the escaping behavior of SGD and Adam through Levy driven SDEs. The theoretical results show that the time for escaping local basic depends on the Radon measure of basin and the heaviness of the gradient noise. This also suggests that Adam has a larger escaping time due to its adaptation. Experiments are also provided to justify their results.

Strengths: The noise assumption considered in this paper is more realistic than gaussian noise assumption made by several previous work. The results suggest a new measurement of “flat” minima, Radon measure of the local basin, which gives another interesting characterization of “flat/sharp” minima.

Weaknesses: The analysis for dynamics of Adam is simplified by Assumption 2. While the intuitive explanation is provided, the results can be strengthened if this can be proved rigorously.

Correctness: The claims and method seem to be reasonable.

Clarity: The paper is well-written and easy to follow.

Relation to Prior Work: The authors discussed related works, and explained several differences between prior works and current paper.

Reproducibility: Yes

Additional Feedback: We know that for over-parameterized neural network, its local minima usually only have an ill-conditioned or even degenerate Hessian. So, I wonder if results would still hold in these situations, such as the non-strongly convex case, i.e., \mu = 0 if Assumption 1. For the right figure of Figure 2(a) and 2(b), I wonder why the loss between two points is decreasing and then increasing. If there is a barrier between these two points, it seems should be first increasing and then decreasing? ============================================================= After rebuttal: Authors feedback addressed my questions.

[Author Response · NeurIPS 2020]

We thank all the reviewers for their insightful and encouraging comments, and will update revision to solve the issues.

**To Reviewer #2.** Our main goal is to theoretically show the stronger escaping ability of SGD over Adam at the same
basin. For the by-product, i.e. relation between Radon measure and escaping time, we construct $f = \min(x^2, a(x-1)^2)$
with a local basin at $x = 1$ shown in right side. By setting $a = 10^5, 500, 150$, we obtain three
basins $A$, $B$ and $C$, where their Radon measures obey $m(A) < m(B) < m(C)$. Then we run SDE
of SGD with initialization $x_0 = 1$ for 2000 iterations, and repeat 1000 times. For $A$, $B$, $C$ with
same Lévy noise, their escaping probabilities are $100\%$, $65.6\%$ and $10.1\%$, and their average
iterations for successful escaping are $122$, $457$ and $1898$. It confirms our theory: the larger
Radon measure of the basin, the harder to escape. We will update it into revision.

The validity of flatness definition should be verified by existing observations. Our definition well explains recent
observation, i.e. good generalization of minima at asymmetric valleys which cannot be explained by existing definitions.

**To Reviewer #3.** 1) Our theory also indicates that SGD with momentum (SGD-M) can generalize better than Adam.
Specifically, as SGD-M does not adapt the geometry, it has the following Lévy SDE with $\boldsymbol{Q}_t = \boldsymbol{I}$:

$$\mathrm{d}\boldsymbol{\theta}_t = -\mu_t \boldsymbol{Q}_t^{-1}\boldsymbol{m}_t + \varepsilon \boldsymbol{Q}_t^{-1}\boldsymbol{\Sigma}_t \mathrm{d}L_t, \quad \mathrm{d}\boldsymbol{m}_t = \beta_1(\nabla \boldsymbol{F}(\boldsymbol{\theta}_t) - \boldsymbol{m}_t), \quad \mathrm{d}\boldsymbol{v}_t = \beta_2([\nabla f_{\mathcal{S}_t}(\boldsymbol{\theta}_t)]^2 - \boldsymbol{v}_t). \tag{9}$$

Then we follow Eqn. (6) in manuscript and obtain escaping set $\mathcal{W} = \{\boldsymbol{y} \in \mathbb{R}^d | \boldsymbol{Q}_{\boldsymbol{\theta}^*}^{-1}\boldsymbol{\Sigma}_{\boldsymbol{\theta}^*}\boldsymbol{y} \notin \Omega^{-\varepsilon^\gamma}\}$ of SGD-M, where
$\boldsymbol{Q}_{\boldsymbol{\theta}^*} = \boldsymbol{I}$ and $\boldsymbol{\Sigma}_{\boldsymbol{\theta}^*} = \lim_{\boldsymbol{\theta}_t \to \boldsymbol{\theta}^*}\boldsymbol{\Sigma}_t$. Since Adam has the same SDE (9) except $\boldsymbol{Q}_t = \mathrm{diag}\left(\sqrt{\omega_t \boldsymbol{v}_t} + \epsilon\right)$ and same escaping set $\mathcal{W}$
except $\boldsymbol{Q}_{\boldsymbol{\theta}^*} = \lim_{\boldsymbol{\theta}_t \to \boldsymbol{\theta}^*}\boldsymbol{Q}_t$, we can directly derive the escaping time $\Gamma = \mathcal{O}\left(\frac{1}{m(\mathcal{W})\Theta(\varepsilon^{-1})}\right)$ of SGD-M with $\Theta(\varepsilon^{-1}) = \frac{2}{\alpha}\varepsilon^\alpha$.

As SGD-M and Adam use the same gradient estimation $\boldsymbol{m}_t$, their gradient noise have the same tail index $\alpha$ and thus the
same factor $\Theta(\varepsilon^{-1})$. For $m(\mathcal{W})$, due to different escaping sets $\mathcal{W}_{\text{SGD-M}}$ of SGD-M and $\mathcal{W}_{\text{Adam}}$ of Adam, $m(\mathcal{W}_{\text{SGD-M}})$ in
SGD-M differs from $m(\mathcal{W}_{\text{Adam}})$ in Adam. By observation, $\mathcal{W}_{\text{SGD-M}}$ is as same as escaping set $\mathcal{W}_{\text{SGD}}$ of SGD in Eqn. (6)
in manuscript, as SGD(-M) have no geometry adaptation. Then Sec. 4.2 proves $\mathcal{W}_{\text{SGD}}$ has much larger volume than
$\mathcal{W}_{\text{Adam}}$. So $m(\mathcal{W}_{\text{SGD-M}})$ is much larger than $m(\mathcal{W}_{\text{Adam}})$. Thus, SGD-M has much smaller escaping time than Adam at the
same basin, and can better escape sharp minima to flat ones for better generalization. We will update this into revision.

2) We follow [20] which analyzes behavior of SGD, and use standard tail index estimation method in [41] as mentioned
in line 334. For learning rate (LR), many works analyze it and conclude: i) an initially large LR helps escape local
minima and accelerates training; ii) decaying LR helps converge to local minima and avoid oscillation. This is testified
by Fig.s 3 in [Jordan, arXiv:1908.01878; R(Kleinberg, arXiv:1802.06175)]. Moreover, Theorem 1 in [R] and analysis in
[Lewkowycz, arXiv:2003.02218] show large LR in SGD help escape. Intuitively, with same basin and gradients, larger
LR gives a larger step and escapes from the basin more easily. These results are consistent with [20] and ours where
$\alpha > 1$ in most cases (see more investigations in [20]). We emphasize that one should focus on the overall variation trend
of $\alpha$ instead of its exact value, as exact value is easily affected by estimation error but variation trend is more robust.

**To Reviewer #4.** 1) Assumption 2 often holds as explained in manuscript but is hard to theoretically prove. $\beta_1 \leq \beta_2 \leq 2\beta_1$
holds under Adam's default setting. There always exist constants $v_{\min}, v_{\max}, \tau_m$ and $\tau$
such that $v_{\min} \leq \sqrt{\boldsymbol{v}_{t,i}} \leq v_{\max}$, $\frac{\|\boldsymbol{m}_t - \widehat{\boldsymbol{m}}_t\|}{\|\int_0^{t^-}(\boldsymbol{m}_s - \widehat{\boldsymbol{m}}_s)\mathrm{d}s\|} \leq \tau_m$ and $\frac{\|\widehat{\boldsymbol{m}}_t\|}{\|\nabla \boldsymbol{F}(\widehat{\boldsymbol{\theta}}_t)\|} \geq \tau$ hold, as i) we allow
$v_{\min} = 0$ due to constant $\epsilon$, ii) $\|\int_0^{t^-}(\boldsymbol{m}_s - \widehat{\boldsymbol{m}}_s)\mathrm{d}s\| \neq 0$ due to their different definitions and
$\|\widehat{\boldsymbol{m}}_t\| \neq 0$ almost sure when non-convergence. $\int_0^\Gamma \langle \frac{\nabla \boldsymbol{F}(\boldsymbol{\theta}_s)}{1+\boldsymbol{F}(\boldsymbol{\theta}_s)}, \mu_s \boldsymbol{Q}_s^{-1}\boldsymbol{m}_s\rangle \mathrm{d}s \geq 0$ generally
holds, as $\nabla \boldsymbol{F}(\boldsymbol{\theta}_t)$ and its exponential average $\boldsymbol{m}_t$ often share similar directions. Right
figure verifies the validity of these assumptions on 4-layered network (width 100).

2) Our results hold for moderately ill-conditioned local basins (ICLBs). Theorem 2 shows that i) after time interval
$v_\varepsilon = \mathcal{O}(\frac{1}{\mu}\ln(\frac{1}{\mu\varepsilon^\delta}))$, noise-free process $\widehat{\boldsymbol{\theta}}_t$ ($\varepsilon = 0$ in SDEs) approaches the minimizer $\boldsymbol{\theta}^*$ of a basin $\Omega$, i.e. $\|\widehat{\boldsymbol{\theta}}_t - \boldsymbol{\theta}^*\| \leq \varepsilon^{-\delta}$;
ii) time interval $\sigma_1$ between two big jumps $\boldsymbol{\zeta}$ (size $\geq \varepsilon^{-\delta}$) is $\sigma_1 = \mathcal{O}(\frac{1}{\varepsilon^{\alpha\delta}})$. Both Theorems 1 and 2 require $v_\varepsilon \leq \sigma_1$ to
guarantee small distance of current solution $\boldsymbol{\theta}_t$ to $\boldsymbol{\theta}^*$ before each big jump. So if $\mu$ of ICLBs is larger than $\mathcal{O}(\varepsilon^{\alpha\delta})$ which
is very small as $\varepsilon$ in SDE is often small to precisely mimic algorithm behaviors, our results still hold. Moreover, to
obtain result i), we assume the optimization trajectory goes along the eigenvector direction corresponding to $\mu$ which is
the worse case and leads to the worst convergence speed. As the measure of one/several eigenvector directions on high
dimension is 0, optimization trajectory cannot always go along the eigenvector direction corresponding to $\mu$. So $v_\varepsilon$ is
actually much larger than $\mathcal{O}(\frac{1}{\mu}\ln(\frac{1}{\mu\varepsilon^\delta}))$, largely improving applicability of our theory. We will update it into revision.

For extremely ICLBs ($\mu \to 0$ or $\mu = 0$), Anandkumar (arXiv:1602.05908v1) proved that first-order algorithms cannot
escape from them, which is also the reasons why recent works (e.g. Jin Chi's works) on escaping saddle points do not
discuss extremely ICLBs. Similarly, our theory also does not hold for this case, which accords with the previous works.
Moreover, $\mu \to 0$ and $\mu = 0$ give asymmetric basins which often generalize well [2,19] and are not needed to escape.

3) Loss around barrier $\cap$ first decreases to the foot of $\cap$, then increases to climb $\cap$ and finally decreases. Fig.2 shows the
first two phases, as final loss is much smaller than the loss around $\cap$, indicating the third phase. We will re-plot it.

[Meta-Review · NeurIPS 2020]

Dear authors, Thank you for your efforts and time. The paper was well-received by the reviewers. Discussions during the rebuttal phase raised the following issue: The theoretical analysis of SGD-M is still not sufficient, and reviewers would like to see more clear justifications for SGD-M in the updated revision. Best AC